# ON THE RELATIONSHIP BETWEEN HETEROPHILY AND ROBUSTNESS OF GRAPH NEURAL NETWORKS

## ABSTRACT

Empirical studies on the robustness of graph neural networks (GNNs) have suggested a relation between the vulnerabilities of GNNs to adversarial attacks and the increased presence of heterophily in perturbed graphs (where edges tend to connect nodes with dissimilar features and labels). In this work, we formalize the relation between heterophily and robustness, bridging two topics previously investigated by separate lines of research. We theoretically and empirically show that for graphs exhibiting homophily (low heterophily), impactful structural attacks always lead to increased levels of heterophily, while for graph with heterophily the change in the homophily level depends on the node degrees. By leveraging these insights, we deduce that a design principle identified to significantly improve predictive performance under heterophily—separate aggregators for ego- and neighbor-embeddings—can also inherently offer increased robustness to GNNs. Our extensive empirical analysis shows that GNNs adopting this design alone can achieve significantly improved *empirical and certifiable* robustness compared to the best-performing unvaccinated model. Furthermore, models with this design can be readily combined with explicit defense mechanisms to yield improved robustness with up to 18.33% increase in performance under attacks compared to the best-performing vaccinated model.

## 1 INTRODUCTION

Graph neural networks (GNNs) aim to translate the enormous empirical success of deep learning to data defined on non-Euclidean domains, such as manifolds or graphs (Bronstein et al., 2017), and have become important tools to solve a variety of learning problems for graph structured and geometrically embedded data. However, recent works show that GNNs—much like their "standard" deep learning counterparts—have a high sensitivity to adversarial attacks: intentionally introduced minor changes in the graph structure can lead to significant changes in performance. This finding, first articulated by Zügner et al. (2018) and Dai et al. (2018), has triggered studies that investigated different attack scenarios (Xu et al., 2019; Wu et al., 2019; Li et al., 2020a; Ma et al., 2020).

A different aspect of GNNs that has been scrutinized recently is that most GNNs do not perform well with many heterophilous datasets. GNNs generally perform well under homophily (or assortativity), i.e., the tendency of nodes with similar features or class labels to connect (Pei et al., 2020; Zhu et al., 2020). Such datasets are thus called *homophilous* (or *assortative*). While homophilous datasets dominate the study of networks, homophily is not a universal principle; certain networks, such as romantic relationship networks or predator-prey networks in ecology, are mostly *heterophilous* (or *disassortative*). Employing a GNN which does not account for heterophily can lead to significant performance loss in heterophilous settings (Abu-El-Haija et al., 2019; Zhu et al., 2020; Bo et al., 2021). Previous works have thus proposed architectures for heterophilous data.

While previous work has focused on naturally-occurring heterophily, heterophilous interactions may also be introduced as adversarial noise: as many GNNs exploit homophilous correlation, they can be sensitive to changes that render the data more heterophilous. A natural follow-up question is if and how this observation manifests itself in previously proposed attacking strategies on GNNs. In this work, we thus investigate the relation between heterophily and robustness of GNNs against adversarial perturbations of graph structure, focusing on semi-supervised node classification tasks.

More specifically, our main contributions are:

- **Formalization:** We formalize the relation between adversarial structural attacks and the change of homophily level in the underlying graphs with theoretical (§3.1) and empirical (§5.1) analysis. Specifically, we show that on homophilous graphs, effective structural attacks lead to increased heterophily, while, on heterophilous graphs, they alter the homophily level contingent on node degrees. To our knowledge, this is the first formal analysis of such kind.
- **Heterophily-inspired Design:** We show how the relation between attacks and heterophily can inspire more robust GNNs by demonstrating that a key architectural feature in handling heterophily, separate aggregators for ego- and neighbor-embeddings, also improves the robustness of GNNs against attacks (§3.2).
- **Extensive Empirical Analysis:** We show the effectiveness of the heterophilous design in improving empirical (§5.2) and certifiable (§5.3) robustness of GNNs with extensive experiments on real-world homophilous and heterophilous datasets. Specifically, we compare GNNs with this design, which we refer to as *heterophily-adjusted* GNNs, to non-adjusted models, including state-of-the-art models designed with robustness in mind. We find that heterophily-adjusted GNNs are up to 5 times more certifiably robust and have stronger performance under attacks by up to 32.92% compared to non-adjusted, standard models. Moreover, this design can be combined with existing vaccination mechanisms, yielding up to 18.33% higher accuracy under attacks than the best non-adjusted vaccinated model.

## 2 NOTATION AND PRELIMINARIES

Let $\mathcal{G} = (\mathcal{V}, \mathcal{E}, \mathbf{X})$ be a simple graph with node set $\mathcal{V}$, edge set $\mathcal{E}$ and node attributes $\mathbf{X}$. The one-hop neighborhood $N(v) = \{u : (u, v) \in \mathcal{E}\}$ of a node $v \in \mathcal{V}$ is the set of all nodes directly adjacent to $v$; the $k$-hop neighborhood of $v \in \mathcal{V}$ is the set of nodes reachable by a shortest path of length $k$. We represent the graph $\mathcal{G}$ algebraically by an adjacency matrix $\mathbf{A} \in \{0, 1\}^{|\mathcal{V}| \times |\mathcal{V}|}$ and node feature matrix $\mathbf{X} \in \mathbb{R}^{|\mathcal{V}| \times F}$. We use $\mathbf{A}_{\mathrm{s}} = \mathbf{A} + \mathbf{I}$ to denote the adjacency matrix with self-loops added, and denote the corresponding row-stochastic matrices as $\bar{\mathbf{A}} = \mathbf{D}^{-1} \mathbf{A}$ and $\bar{\mathbf{A}}_{\mathrm{s}} = \mathbf{D}_{\mathrm{s}}^{-1} \mathbf{A}_{\mathrm{s}}$, respectively, where $\mathbf{D}$ is a diagonal matrix with $\mathbf{D}_{ii} = \sum_j \mathbf{A}_{ij}$ ($\mathbf{D}_{\mathrm{s}}$ is defined analogously). We further assume that there exists a vector $\mathbf{y}$, which contains a unique class label $y_v$ for each node $v$. Given a training set $\mathcal{T}_{\mathcal{V}} = \{(v_1, y_1), (v_2, y_2), ...\}$ of labeled nodes, the goal of semi-supervised node classification is to learn a mapping $\ell : \mathcal{V} \to \mathcal{Y}$ from the nodes to the set $\mathcal{Y}$ of class labels.

**Graph neural networks (GNNs).** Most current GNNs operate according to a message passing paradigm where a representation vector $\mathbf{r}_v$ is assigned to each node $v \in \mathcal{V}$ and continually updated by $K$ layers of learnable transformations. These layers first aggregate representations over neighboring nodes $N(v)$ and then update the current representation via an encoder ENC. For prevailing GNN models like GCN (Kipf & Welling, 2017) and GAT (Veličković et al., 2018), each layer can be formalized as $\mathbf{r}_v^{(k)} = \text{ENC}\left(\text{AGGR}\left(\left\{\mathbf{r}_u^{(k-1)} : u \in N(v) \cup \{v\}\right\}\right)\right)$, where AGGR is the mean function weighted by node degrees (GCN) or an attention mechanism (GAT), and ENC is a learnable (nonlinear) mapping.

**Adversarial attacks on graphs.** Given a graph $\mathcal{G} = (\mathcal{V}, \mathcal{E}, \mathbf{X})$ and a GNN $f$ that processes $\mathcal{G}$, an adversarial attacker tries to create a perturbed graph $\mathcal{G}' = (\mathcal{V}, \mathcal{E}', \mathbf{X})$ with a modified edge-set $\mathcal{E}'$ such that the performance of the GNN $f$ is maximally degraded. The information available to the attacker can vary under different scenarios (Jin et al., 2020a; Sun et al., 2020). Here, we follow the gray-box formalization by Zügner et al. (2018), where the attacker knows the training set $\mathcal{T}_{\mathcal{V}}$, but not the trained GNN $f$. The attacker thus considers a surrogate GNN and picks perturbations that maximize an attack loss $\mathcal{L}_{\mathrm{atk}}$ (Zügner et al., 2018; Zügner & Günnemann, 2019a), assuming that attacks to the surrogate model are transferable to the attacked GNN. For node classification, the attack loss $\mathcal{L}_{\mathrm{atk}}$ quantifies how the predictions $\mathbf{z}_v \in [0, 1]^{|\mathcal{Y}|}$ made by the GNN $f$ differ from the true labels $\mathbf{y}$. For a targeted attack of node $v$ with class label $y_v \in \mathcal{Y}$, we adopt the negative classification margin (**CM-type**) (Zügner et al., 2018; Xu et al., 2019): $\mathcal{L}_{\mathrm{atk}} = -\Delta_c = -(\mathbf{z}_{v,y_v} - \max_{y \neq y_v} \mathbf{z}_{v,y})$. The attacker usually has additional constraints, such as a limit on the size of the perturbations allowed (Zügner et al., 2018; Zügner & Günnemann, 2019a).

**Taxonomy of attacks.** We follow the taxonomy of attacks introduced in (Jin et al., 2020a; Sun et al., 2020). For node classification, the attacker may aim to change the classification of a specific node $v \in \mathcal{V}$ (**targeted attack**), or to decrease the overall classification accuracy (**untargeted attack**). Attacks can also happen at different stages of the training process: we refer to attacks introduced

before training as (pre-training) **poison attacks**, and attacks introduced after the training process (and before potential retraining on perturbed data) as (post-training) **evasion attacks**. While our theoretical analysis (§3) mainly considers targeted evasion attacks, we consider other attacks in our empirical evaluation (§5).

**Characterizing homophily and heterophily in graphs.** Using class labels, we characterize the types of connections in a graph contributing to its overall level of homophily/heterophily as follows:

**Definition 1 (Homo/Heterophilous path and edge)** *A $k$-hop homophilous path from node $w$ to $u$ is a length-$k$ path between endpoint nodes with the same class label $y_w = y_u$. Otherwise, the path is called* heterophilous. *A homophilous or heterophilous edge is a special case with $k = 1$.*

Following (Zhu et al., 2020; Lim et al., 2021), we define the homophily ratio $h$ as:

**Definition 2 (Homophily ratio)** *The* homophily ratio *is the fraction of homophilous edges among all the edges in a graph: $h = |\{(u, v) \in \mathcal{E} | y_u = y_v\}| / |\mathcal{E}|$.*

When the edges in a graph are wired randomly, independent to the node labels, the expectation for $h$ is $h_r = 1/|\mathcal{Y}|$ for balanced classes (Lim et al., 2021). For simplicity, we informally refer to graphs with homophily ratio $h \gg 1/|\mathcal{Y}|$ as **homophilous graphs** (which have been the focus in most prior works), graphs with homophily ratio $h \ll 1/|\mathcal{Y}|$ as **heterophilous graphs**, and graphs with homophily ratio $h \approx 1/|\mathcal{Y}|$ as **weakly heterophilous graphs**.

## 3 RELATION BETWEEN GRAPH HETEROPHILY & MODEL ROBUSTNESS

In this section, we first show theoretical results on the relation between adversarial structural attacks and the change in the homophily level of the underlying graphs. Though empirical analyses from previous works have suggested this relation on homophilous graphs (Wu et al., 2019; Jin et al., 2020a), to our knowledge, we are the first to formalize it with theoretical analysis and address the case of heterophilous graphs. As an implication of the relation, we then discuss how a key design that improves predictive performance of GNNs under heterophily can also help boost their robustness.

### 3.1 HOW DO STRUCTURAL ATTACKS CHANGE HOMOPHILY IN GRAPHS?

**Homophilous Graphs: Structural Attacks are Mostly Heterophilous Attacks.** Our first result shows that, for homophilous data, effective structural attacks on GNNs (as measured by loss $\mathcal{L}_{\text{atk}}$) always result in a reduced level of homophily where either new heterophilous connections are added or existing homophilous connections are removed. It also states that direct perturbations on 1-hop neighbors of the target nodes are more effective than indirect perturbations (influencer attacks (Zügner et al., 2018)) on multi-hop neighbors. For simplicity, akin to previous works (Zügner et al., 2018; Zügner & Günnemann, 2019a) we establish our results for targeted evasion (post-training) attacks in a stylized learning setup with a linear GNN. However, our findings generalize to more general setups on real-world datasets as we show in our experiments (§5.1). In the theorems below, we use the notion of *gambit node*: node $u$ is called a gambit if a perturbation that targets node $v \in \mathcal{V}$ adjusts the connectivity of node $u \in \mathcal{V}$.

**Theorem 1** *Let $\mathcal{G} = (\mathcal{V}, \mathcal{E}, \mathbf{X})$ be a self-loop-free graph with adjacency matrix $\mathbf{A}$ and node features $\mathbf{x}_v = p \cdot \text{onehot}(y_v) + \frac{1-p}{|\mathcal{Y}|} \cdot \mathbf{1}$ for each node $v$, where $\mathbf{1}$ is an all-1 vector, and $p$ is a parameter that regulates the signal to noise ratio. Assume that a fraction $h$ of each node's neighbors belong to the same class, while a fraction $\frac{1-h}{|\mathcal{Y}|-1}$ belongs uniformly to any other class. Consider a 2-layer linear GNN $f_s^{(2)}(\mathbf{A}, \mathbf{X}) = \bar{\mathbf{A}}_s^2 \mathbf{X} \mathbf{W}$ trained on a training set $\mathcal{T}_\mathcal{V} \subseteq \mathcal{D}_\mathcal{V}$, with at least one node from each class $y \in \mathcal{Y}$, and degree $d$ for all nodes with a distance less than 2 to any $v \in \mathcal{D}_\mathcal{V}$. For a unit structural perturbation that involves a target node $v \in \mathcal{D}_\mathcal{V}$, and a correctly classified gambit node with degree $d_a$, the following statements hold if $h \geq \frac{1}{|\mathcal{Y}|}$:*

1. *the attack loss $\mathcal{L}_{\text{atk}}$ (§2) of the target $v$ increases only for actions* increasing heterophily, *i.e., when removing a homophilous edge or path, or adding a heterophilous edge or path to node $v$;*
2. *direct perturbations on edges (or 1-hop paths) incident to the target node $v$ lead to greater increase in $\mathcal{L}_{\text{atk}}$ than indirect perturbations on multi-hop paths to target node $v$.*

We give the proof in App. C.1. Intuitively, the relative inability of existing GNNs to make full use of heterophilous data (Pei et al., 2020; Zhu et al., 2020) can be exploited by inserting heterophilous

connections in graphs where homophilous ones are expected. Though the theorem shows that effective attacks on homophilous graphs *necessarily* reduce the homophily level, the converse is not true: not all perturbations which reduce the homophily level are effective attacks (Ma et al., 2021).

**Heterophilous Graphs: Structural Attacks Can Be Homophilous or Heterophilous, depending on Node Degrees.** When a graph displays heterophily, our analysis shows a more complicated picture on how the level of homophily in the graph is changed by effective structural attacks: in heterophilous case, the direction of change is dependent on the degrees of both the target node $v$ and the gambit node $u$ of the attack. Specifically, if the degree of *either node* is low, attacks increasing the heterophily are still effective; however, if the degrees $d$ and $d_a$ of *both nodes* are high, attacks *decreasing* the heterophily will be effective. Similar to the homophilous case, we formalize our results below for targeted evasion attacks in a stylized learning setup.

**Theorem 2** *Under the setup of Thm. 1, for a unit perturbation that involves a target node $v$ with degree $d$, and a correctly classified gambit node with degree $d_a$, the following statements hold:*

1. **(Low-degree target node)** *if $0 < d \leq |\mathcal{Y}| - 2$, for any $d_a \geq 0$ and $h \in [0, 1]$, the attack loss $\mathcal{L}_{\text{atk}}$ (§2) of $v$ increases only under actions* increasing heterophily *in the graph;*

2. **(High-degree target node)** *if $d > |\mathcal{Y}| - 2$, conditioning on the degree $d_a$ of the gambit node:*

   (a) **(Low-degree gambit node)** *if $d_a < \frac{(d+2)(|\mathcal{Y}|-1)}{d-|\mathcal{Y}|+2}$, for any $h \in [0, 1]$, the attack loss $\mathcal{L}_{\text{atk}}$ (§2) of $v$ increases only under actions* increasing heterophily *in the graph;*

   (b) **(High-degree gambit node)** *if $d_a \geq \frac{(d+2)(|\mathcal{Y}|-1)}{d-|\mathcal{Y}|+2}$, for $0 \leq h < \frac{d_a(d-|\mathcal{Y}|+2)-(d+2)(|\mathcal{Y}|-1)}{(d+1)|\mathcal{Y}|d_a} < \frac{1}{|\mathcal{Y}|}$, $\mathcal{L}_{\text{atk}}$ (§2) of $v$ increases only under actions* reducing heterophily *in the graph.*

*In the statements above, the actions* increasing heterophily *in the graph include removing a homophilous edge or adding a heterophilous edge to node $v$, and the actions* reducing heterophily *in the graph include when adding a homophilous edge or removing a heterophilous edge to node $v$.*

The above theorems cover the situation when the gambit nodes are initially classified correctly (where attacks introducing heterophily can be unambiguously defined using the ground-truth class labels of the nodes involved). However, in §5.1, we show on real-world datasets that a relaxed interpretation of the theorems, where heterophily is instead defined by the *predicted* class labels of GNNs, can explain the behavior of the attacks regardless of the initial correctness of the gambits.

### 3.2 Boosting Robustness with A Simple Heterophilous Design

A natural follow-up question is whether GNNs with better performance under heterophily are also more robust against structural attacks. We deduce that a key design for improving GNN performance for heterophilous data—separate aggregators for ego- and neighbor-embeddings—can also boost the robustness of GNNs by enabling them to better cope with adversarially-introduced changes in heterophily.

**Separate Aggregators for Ego- and Neighbor-embeddings.** This design uses separate GNN aggregators for ego-embedding $\mathbf{r}_v$ and neighbor-embeddings $\{\mathbf{r}_u : u \in N(v)\}$. Formally, the representation learned for node $v$ in the $k$-th layer is:

$$\mathbf{r}_v^{(k)} = \text{ENC}\left(\text{AGGR1}(\mathbf{r}_v^{(k-1)}, \mathbf{r}_v^{(k-2)}, ..., \mathbf{r}_v^{(0)}), \text{AGGR2}(\{\mathbf{r}_u^{(k-1)} : u \in N(v)\})\right), \quad (1)$$

where AGGR1 and AGGR2 are *separate* aggregators, such as averaging functions (GCN), attention mechanisms (GAT), or other pooling mechanisms (Hamilton et al., 2017). The ego-aggregator AGGR1 may also introduce skip connections (Xu et al., 2018) to the ego-embeddings aggregated in previous layers as shown in Eq. (1). This design has been utilized in existing GNN models (see §D.1 for details), and has been shown to significantly boost the representation power of GNNs under natural heterophily (Zhu et al., 2020).

**Intuition.** The key design changes, as compared to the GCN formulation in §2, allow for the ego-embedding $\mathbf{r}_v$ to be aggregated and weighted *separately* from the neighbor-embeddings $\{\mathbf{r}_u : u \in N(v)\}$, as well as for the use of skip connections to ego-embeddings of previous layers. Intuitively, ego-embeddings of feature vectors at the first layer are independent of the graph structure and thus unaffected by adversarial structural perturbations. Hence, a separate aggregator and skip connections can provide better access to unperturbed information, and helps mitigate the effects of the attacks.

**Theoretical Analysis.** We formalize the above intuition that shows how separate aggregators for ego- and neighbor-embeddings enable GNN layers to reduce the attack loss.

**Theorem 3** *Under the setup of Thm. 1, consider two alternative layers from which a two-layer linear GNN is built: **(1)** a layer defined as $f_s(\mathbf{A}, \mathbf{X}) = \bar{\mathbf{A}}_s \mathbf{X} \mathbf{W}$; and **(2)** a layer formulated as $f(\mathbf{A}, \mathbf{X}; \alpha) = \left((1-\alpha)\bar{\mathbf{A}} + \alpha\mathbf{I}\right)\mathbf{X}\mathbf{W}$, which mixes the ego- and neighbor-embedding linearly under a predefined weight $\alpha \in [0, 1]$. Then, for $h > 1/|\mathcal{Y}|$, $\alpha > 1/(1 + d_a)$, and a unit perturbation increasing $\mathcal{L}_{\mathrm{atk}}$ as in Thm. 1, outputs of layer $f$ lead to a strictly smaller increase in $\mathcal{L}_{\mathrm{atk}}$ than $f_s$.*

We provide the proof in App. C.3; note that for $\alpha = 1/(1 + d_a)$, the two layers are the same: $f(\mathbf{A}, \mathbf{X}; \alpha) = f_s(\mathbf{A}, \mathbf{X})$. Theorem 3 shows that an increase to the weights of ego-embedding improves the robustness of the GNN $f$ for a homophily ratio $h > 1/|\mathcal{Y}|$. Though aggregators and encoders are stylized in the theorem, the empirical analysis in §5.2 confirms that GNNs with more advanced aggregators and encoders also benefit from separate aggregators. Specifically, we find that such GNNs outperform methods without this design by up to 33.33% and 48.88% on homophilous and heterophilous graphs, respectively, while performing comparably on clean datasets.

## 4 RELATED WORK

**Adversarial Attacks and Defense Strategies for Graphs** Since NETTACK (Zügner et al., 2018) and RL-S2V (Dai et al., 2018) first demonstrated the vulnerabilities of GNNs against adversarial perturbations, a variety of attack strategies under different scenarios have been proposed, including adversarial attacks on the graph structure (Dai et al., 2018; Xu et al., 2019; Bojchevski & Günnemann, 2019a; Li et al., 2020a; Chang et al., 2020), node features (Takahashi, 2019; Ma et al., 2020), or combinations of both (Zügner et al., 2018; Zügner & Günnemann, 2019a; Wu et al., 2019). On the defense side, various techniques for improving the GNN robustness against adversarial attacks have been proposed, including: adversarial training (Xu et al., 2019; Zügner & Günnemann, 2019a; Bojchevski & Günnemann, 2019b); RGCN (Zhu et al., 2019), which adopts Gaussian-based embeddings and a variance-based attention mechanism; low-rank approximation of graph adjacency (Entezari et al., 2020) against Nettack (Zügner et al., 2018); Pro-GNN (Jin et al., 2020b), which estimates the unperturbed graph structure in training with the assumptions of low-rank, sparsity, and homophily of node features; GCN-Jaccard (Wu et al., 2019) and GNNGuard (Zhang & Zitnik, 2020), which assume homophily of features (or structural embeddings) and train GNN models on a pruned graph with only strong homophilous links; and Soft Medoid (Geisler et al., 2020), an aggregation function with improved robustness. Other recent works have looked into the certification of nodes that are guaranteed to be robust against certain structural and feature perturbations (Zügner & Günnemann, 2019b; Bojchevski & Günnemann, 2019b; Zügner & Günnemann, 2020), including approaches based on model-agnostic randomized smoothing (Cohen et al., 2019; Lee et al., 2019; Bojchevski et al., 2020). Interested readers are referred to the recent surveys (Jin et al., 2020a; Sun et al., 2020) for a comprehensive review of the literature.

**GNNs & Heterophily** Recent works (Pei et al., 2020; Liu et al., 2020; Zhu et al., 2020; Ma et al., 2021) have shown that heterophilous datasets can lead to significant performance loss for popular GNN architectures (e.g., GCN (Kipf & Welling, 2017), GAT (Veličković et al., 2018)). This issue is also known in classical semi-supervised learning (Peel, 2017). To address this issue, several GNN designs for handling heterophilous connections have been proposed (Abu-El-Haija et al., 2019; Pei et al., 2020; Zhu et al., 2020; Dong et al., 2021; Li et al., 2021; Zhu et al., 2021; Bo et al., 2021). Yan et al. (2021) recently discussed the connection between heterophily and oversmoothing for GNNs, and proposed designs to simultaneously address both issues. However, the formal connection between heterophily and robustness of GNNs has received little attention. Here we focus on a simple yet powerful design that significantly improves performance under heterophily (Zhu et al., 2020), and can be readily incorporated into GNNs.

## 5 EMPIRICAL EVALUATION

Our analysis seeks to answer the following questions: (Q1) Does our theoretical analysis on the relations between adversarial attacks and changes in heterophily level generalize to real-world datasets? (Q2) Do heterophily-adjusted GNNs, i.e., models with separate aggregators for ego- and neighbor-embeddings, show improved robustness against state-of-the-art attacks? (Q3) Does the identified design improve the *certifiable* robustness of GNNs?

First, we describe the experimental setup and datasets that we use to answer the above questions.

**Attack Setup.** We consider both targeted and untargeted attacks (§2), generated by NET-TACK (Zügner et al., 2018) and Metattack (Zügner & Günnemann, 2019a), respectively. For each attack method, we consider poison (pre-training) and evasion (post-training) attacks, yielding 4 attack scenarios in total. We focus on robustness against structural perturbations and keep the node features unchanged. We randomly generate 3 sets of perturbations per attack method and dataset, and consistently evaluate each GNN model on them. We provide more details in App. D.2.

**GNN Models.** To show the effectiveness of our identified design, we evaluate four groups of models against adversarial attacks: **(1)** Models with this heterophilous design only: GraphSAGE (Hamilton et al., 2017), $H_2$GCN (Zhu et al., 2020), CPGNN (Zhu et al., 2021), GPR-GNN (Chien et al., 2021) and FAGCN (Bo et al., 2021); we discuss how these models instantiate this design in App. §D.1; **(2)** State-of-the-art "vaccinated" architectures designed with robustness in mind: ProGNN (Jin et al., 2020b), GNNGuard (Zhang & Zitnik, 2020) and GCN-SVD (Entezari et al., 2020); **(3)** Models with both the heterophilous design and explicit robustness-enhancing mechanisms based on low-rank approximation: $H_2$GCN-SVD and GraphSAGE-SVD; **(4)** Models without any vaccination, including some of the most popular methods: GCN (Kipf & Welling, 2017), GAT (Veličković et al., 2018), and the graph-agnostic multilayer perceptron (MLP) which relies only on node features. We discuss the implementations and parameters used for the models in App. D.

**Datasets & Evaluation Setup.** We consider two standard datasets with strong homophily, Cora (McCallum, 2000) and Citeseer (Sen et al., 2008), complemented with one weakly and one strongly heterophilous graph, introduced by Lim et al. (2021): FB100 (Traud et al., 2012) and Snap Patents (Leskovec et al., 2005; Leskovec & Krevl, 2014). We report summary statistics

Table 1: Dataset statistics.

| | Homophilous graphs | | Heterophilous graphs | |
|---|---|---|---|---|
| | **Cora** | **Citeseer** | **FB100** | **Snap** |
| **#Nodes** $|\mathcal{V}|$ | 2,485 | 2,110 | 2,032 | 4,562 |
| **#Edges** $|\mathcal{E}|$ | 5,069 | 3,668 | 78,733 | 12,103 |
| **#Classes** $|\mathcal{Y}|$ | 7 | 6 | 2 | 5 |
| **#Features** $F$ | 1,433 | 3,703 | 1,193 | 269 |
| **Homophily** $h$ | 0.804 | 0.736 | 0.531 | 0.134 |

in Table 1, and provide more details in App. D.5. For computational tractability, we subsample the Snap Patents data via snowball sampling (Goodman, 1961), where we keep 20% of the neighbors for each traversed node (see App. D.5 for details). We follow the evaluation procedure of Zügner et al. (2018) and Jin et al. (2020b), where we split the nodes of each dataset into training (10%), validation (10%) and test (80%) data, and determine the model parameters using the training and validation splits. We report the average performance and standard deviation on the 3 sets of generated perturbations. For targeted attacks with NETTACK, we report the classification accuracy on the target nodes; for untargeted attacks with Metattack, we report it over the whole test data.

**Robustness Certificates.** We adopt randomized smoothing for GNNs (Bojchevski et al., 2020) to evaluate the certifiable robustness, with parameter choices as detailed in App. D.2. We only consider structural perturbations in the randomization scheme. Following Geisler et al. (2020), we measure the certifiable robustness of GNN models with the accumulated certifications (AC) and the average maximum certifiable radii for edge additions ($\bar{r}_a$) and deletions ($\bar{r}_d$) over all correctly predicted nodes. More specifically, AC is defined as $-R(0,0) + \sum_{r_a, r_d \geq 0} R(r_a, r_d)$, where $R(r_a, r_d)$ is the *certifiably correct ratio*, i.e., the ratio of the nodes in the test splits that are *both* predicted correctly by the smoothed classifier *and* certifiably robust at radius $(r_a, r_d)$. In addition, we report the accuracy of each model with randomized smoothing enabled on the test splits of the clean datasets, which is equal to $R(0,0)$. We report the average and standard deviation of each statistic over the 3 different training, validation and test splits.

## 5.1 (Q1) Structural Attacks are Mostly Heterophilous: Empirical Validation

To show that our theoretical analysis in §3.1 generalizes to more complex settings beyond the assumptions we made in the theorems, we look into effective targeted attacks made by NETTACK on real-world homophilous and heterophilous datasets, and present statistics of the attacks in Table 2, with a focus on the ratios of heterophilous attacks. We use a budget of 1 perturbation per target node in this experiment, and the statistics are reported among all effective perturbations targeting nodes that are correctly classified on clean datasets by the surrogate GNN used by NETTACK (i.e., GCN) as described in §5. To help validate the dependency between the degrees of the target/gambit nodes

and the changes of heterophily predicted by Thm. 2, we also show the scatter plots of target and gambit node degrees in Fig. 2 in App. E.5.

**Homophilous Networks.** For the strongly homophilous Cora and Citeseer graphs, all changes in the graph structure that are introduced by effectives attacks follow the conclusion of Thm. 1: they reduce homophily (increase heterophily) by adding heterophilous edges or removing homophilous edges. These results show that despite the simplified analysis, the takeaway of Thm. 1 can be generalized to real-world datasets. In addition, the attacks mostly introduce, rather than prune, edges, suggesting that attacks adding outlier edges to the graph are more powerful

Table 2: Effective targeted attacks by NETTACK: ratios of edge additions, deletions and heterophilous attacks (i.e., attacks increasing heterophily). We consider two heterophily definitions, one based on ground-truth class labels (Label), and the other on predicted class labels by GCN on clean datasets (Pred.). All attacks are direct perturbations on edges incident to the targets.

| | Dataset | Sample Sizes | Attack Type | | Hete. Attacks | |
| --- | --- | --- | --- | --- | --- | --- |
| | | | Add. | Del. | Label | Pred. |
| NETTACK | Cora | 150 | 99.33% | 0.67% | 100.00% | 100.00% |
| | Citeseer | 121 | 100.00% | 0.00% | 100.00% | 100.00% |
| | FB100 | 112 | 100.00% | 0.00% | 50.00% | 100.00% |
| | Snap | 51 | 100.00% | 0.00% | 64.71% | 100.00% |

than attacks removing informative existing edges. These observations in our experiments are consistent with the observations from previous works (Jin et al., 2020a; Geisler et al., 2020).

**Heterophilous Networks.** For heterophilous graphs FB100 ($h \approx 1/|\mathcal{Y}|$) and Snap ($h < 1/|\mathcal{Y}|$), Fig. 2 in App. E.5 shows that almost all attacks leverage gambit nodes with low degrees (1 or 2); no node with degree higher than 5 is leveraged. All attacks leveraging correctly classified gambit nodes are connecting node $u \in \mathcal{V}$ with a different ground-truth class label $y_u \neq y_v$ to the target nodes $v \in \mathcal{V}$; attacks leveraging incorrectly classified gambit nodes are always connecting node $u$ with a different *predicted* class label $\hat{y}_u \neq \hat{y}_v = y_v$ to the target node $v$, even though some gambit nodes have the same *ground-truth* class label $y_u = y_v \neq \hat{y}_u$ as the target nodes. These results validate the conclusion of Thm. 2 on correctly classified gambit nodes, and demonstrate its generalizability under the heterophily definition based on predicted class labels. Note that the predicted class labels $\hat{y}_u$ for each node $u \in \mathcal{V}$ are based on GCN, which is the surrogate GNN used by NETTACK.

## 5.2 (Q2) BENCHMARK STUDY OF GNN MODELS: HETEROPHILOUS DESIGN LEADS TO IMPROVED EMPIRICAL ROBUSTNESS

To answer (Q2) on whether heterophily-adjusted GNN models show improved performance against state-of-the-art attacks, we conduct a comprehensive benchmark study. We consider all four categories of GNN models mentioned in §5, and evaluate their robustness against both targeted and untargeted attacks. We report the hyperparameters for each method in App. D.4. Table 3 shows the performance of each method under poison (pre-training) attacks, and Fig. 1 visualizes the corresponding performance changes relative to the clean datasets. For conciseness, we report further results under evasion (post-training) attacks and on clean (unperturbed) data in the Appendix (Table 7 for NETTACK; Table 8 for Metattack). Also, in App. E.4 we discuss how our simple heterophilous design leads to only minor computational overhead compared to existing vaccination mechanisms.

**Targeted attacks by NETTACK.** ① *Poison attacks*. Under targeted poison attacks, Table 3 (left) shows that GraphSAGE-SVD and H$_2$GCN-SVD, which combine our identified design with a low-rank vaccination approach adopted in GCN-SVD (Entezari et al., 2020) (details in App. D.3), outperform state-of-the-art methods across all datasets by up to 13.34% in homophilous settings and 18.33% in heterophilous settings.

Methods merely employing the identified design also show significantly improved robustness, though there are differences in the amount of robustness improvement due to architectural differences. Specifically, these methods outperform the best unvaccinated method (GAT) on all datasets by up to 32.92% in average, despite having comparable performance on clean datasets. On Citeseer and FB100, methods with the heterophilous design also show comparable or even better robustness than state-of-the-art vaccinated GNNs like ProGNN and GCN-SVD.

② *Evasion attacks*. Under evasion attacks (Table 7), we observe *similar trends as in poison attacks*: GraphSAGE-SVD and H$_2$GCN-SVD are up to 20.55% more accurate than the GCN-SVD baseline, and methods featuring the identified design alone achieve up to 24.45% gain in average performance against the best unvaccinated baseline. We note that GNNGuard and ProGNN are not capable of ad-

Table 3: Benchmark study: mean accuracy against poison attacks (accuracy on clean datasets in paranthesis). Accuracy is reported on target nodes for NETTACK, and on full test splits for Metattack. Best GNN model is highlighted in blue per dataset, and in gray per model group. MLP is immune to structural attacks and not considered as a GNN model. Detailed results, including stdev and accuracy against evasion attacks, are listed in Table 7 and 8 and the setup in §5.

| | Hetero. | Vaccin. | | Homophilous graphs | | Heterophilous graphs | | | Homophilous graphs | | Heterophilous graphs |
| | | | | Cora $h$=0.804 | Citeseer $h$=0.736 | FB100 $h$=0.531 | Snap $h$=0.134 | | Cora $h$=0.804 | Citeseer $h$=0.736 | FB100 $h$=0.531 | Snap $h$=0.134 |
|---|---|---|---|---|---|---|---|---|---|---|---|---|
| H$_2$GCN-SVD | ✓ | ✓ | | 70.00 (74.44) | 65.00 (70.00) | 59.44 (61.67) | 28.89 (30.56) | | 67.87 (76.89) | 70.42 (73.42) | 56.72 (56.81) | 25.60 (27.63) |
| GraphSAGE-SVD | ✓ | ✓ | | 71.67 (77.22) | 67.78 (70.00) | 60.00 (60.00) | 26.67 (27.22) | | 68.86 (77.52) | 69.10 (72.16) | 55.76 (57.38) | 26.58 (26.72) |
| H$_2$GCN | ✓ | | | 38.89 (82.78) | 27.22 (69.44) | 27.78 (60.56) | 12.78 (30.00) | | 57.75 (83.94) | 54.34 (75.34) | 54.84 (56.95) | 25.34 (27.49) |
| GraphSAGE | ✓ | | | 36.67 (82.22) | 31.67 (70.56) | 33.89 (60.00) | 16.67 (24.44) | | 54.68 (82.21) | 59.74 (74.64) | 54.72 (56.60) | 24.14 (27.18) |
| CPGNN | ✓ | | | 47.22 (81.67) | 40.56 (73.33) | 49.44 (66.11) | 21.67 (28.89) | | 74.55 (80.67) | 68.07 (74.92) | 61.58 (60.17) | 26.76 (27.13) |
| GPR-GNN | ✓ | | | 21.67 (82.22) | 24.44 (67.78) | 2.78 (56.67) | 4.44 (27.78) | | 48.29 (81.84) | 35.25 (70.71) | 59.94 (62.40) | 21.06 (26.08) |
| FAGCN | ✓ | | | 26.11 (83.33) | 25.56 (70.56) | 6.11 (58.33) | 8.33 (29.44) | | 60.11 (81.59) | 53.18 (73.99) | 55.97 (59.64) | 24.04 (27.15) |
| GNNGuard | | ✓ | | 58.33 (77.22) | 59.44 (67.78) | 0.56 (67.22) | 9.44 (28.33) | | 74.20 (80.15) | 68.13 (72.61) | 60.89 (65.66) | 23.78 (26.51) |
| ProGNN | | ✓ | | 48.89 (79.44) | 32.78 (67.22) | 33.89 (51.11) | 17.78 (27.22) | | 45.10 (81.32) | 46.58 (71.82) | 53.40 (49.84) | 24.80 (27.49) |
| GCN-SVD | | ✓ | | 53.33 (75.56) | 28.89 (59.44) | 41.67 (50.56) | 25.00 (27.78) | | 47.82 (76.61) | 51.20 (66.90) | 55.00 (55.47) | 25.25 (26.63) |
| GAT | | | | 13.89 (84.44) | 8.89 (70.00) | 0.56 (60.56) | 3.89 (30.56) | | 41.70 (83.72) | 48.40 (73.40) | 50.37 (61.69) | 25.00 (27.30) |
| GCN | | | | 1.67 (82.78) | 4.44 (69.44) | 0.56 (63.33) | 2.22 (33.33) | | 37.46 (84.32) | 45.81 (74.27) | 51.82 (64.86) | 25.03 (27.30) |
| MLP* | | | | 64.44 (64.44) | 70.56 (70.56) | 57.78 (57.78) | 30.00 (30.00) | | 64.55 (64.55) | 67.67 (67.67) | 56.56 (56.56) | 26.25 (26.25) |

dressing evasion attacks. In summary, these observations show that the heterophily-inspired design is orthogonal to existing vaccination mechanisms for improving the robustness of GNNs.

**Untargeted attacks by Metattack.** ① *Poison attacks.* We also test the robustness of each method against untargeted attacks. Table 3 (right) shows the performance under poison attacks. Though our theoretical analysis in §3 focuses on the effect of the heterophilous design under targeted attacks, we observe similar improvements in robustness against untargeted attacks in the poison setup. GNNs with the identified design show mostly improved robustness compared to unvaccinated models, while having similar performance on the clean datasets. Specifically, CPGNN shows exceptional robustness, outperforming the best unvaccinated model by up to 32.85%. Moreover, models combining

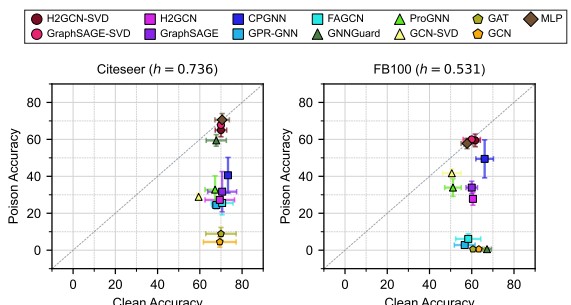

Figure 1: (Best viewed in color.) Classification accuracy on clean data and against poison attacks for target nodes attacked by NETTACK. Error bars show standard deviation across different sets of experiments. Detailed results are listed in Table 7. As expected, MLP is not influenced by the adversarial structural attacks.

the identified design with low-rank approximation show more than 10% improvement accuracy compared to GCN-SVD, which uses only low-rank approximation, and ProGNN. We note that GNNGuard, which uses a similarity-based defense strategy, also shows more competitive robustness under untargeted attacks. This is in line with existing works showing that low-rank based approach does not adapt well to untargeted attacks (Jin et al., 2020b). Nevertheless, methods combining a heterophilous design and low-rank approximations still outperform GNNGuard on two datasets.

② *Evasion attacks.* We present the performance under evasion attacks and on clean datasets in Table 8 in the Appendix. *Unlike the poison attacks*, the evasion setup only leads to a slight decrease in average accuracy of less than 2% for most models. Moreover, there appears to be no clearly increased robustness for vaccinated models (with the identified design or other vaccination mechanisms) compared to unvaccinated models. This can be attributed to the reduced effectiveness of evasion vs. poison attacks (as in NETTACK), and the increased challenges of untargeted attacks.

## 5.3 (Q3) HETEROPHILY-ADJUSTED GNNS ARE CERTIFIABLY MORE ROBUST

It is worth noting that robustness against specific attacks such as NETTACK and Metattack does not guarantee robustness towards other possible attacks. To overcome this limitation, *robustness certificates* provide guarantees (in some cases probabilistically) that attacks within a certain radius cannot change a model's predictions. Complementary to our evaluation on empirical robustness, we further demonstrate that heterophily-adjusted GNNs featuring our identified design are certifiably

Table 4: Accumulated certifications (AC), average certifiable radii ($\bar{r}_a$ and $\bar{r}_d$) and accuracy of GNNs with randomized smoothing enabled (i.e., $f(\phi(\mathbf{s}))$) on the test splits of the clean datasets, with a ramdomization scheme $\phi$ allowing both addition and deletion (i.e., $p_+ = 0.001, p_- = 0.4$). For each statistic, we report the mean and stdev across 3 runs. Best results highlighted in blue per dataset, and in gray per model group. For results with other randomization schemes, see Table 9.

| | Hete. | | Addition & Deletion | | | | | Addition & Deletion | | | |
|---|---|---|---|---|---|---|---|---|---|---|---|
| | | | AC | $\bar{r}_a$ | $\bar{r}_d$ | Acc. % | | AC | $\bar{r}_a$ | $\bar{r}_d$ | Acc. % |
| **$H_2$GCN** | ✓ | **Cora** | 3.91±0.31 | 0.46±0.08 | 3.88±0.29 | 79.14±2.01 | **Citeseer** | 2.98±0.88 | 0.34±0.13 | 3.29±0.67 | 71.43±3.92 |
| **GraphSAGE** | ✓ | | 2.12±0.07 | 0.12±0.00 | 2.41±0.04 | 79.43±1.43 | | 2.25±0.15 | 0.20±0.01 | 2.59±0.10 | 73.34±2.66 |
| **CPGNN** | ✓ | | 1.87±0.27 | 0.14±0.05 | 2.24±0.30 | 75.37±1.65 | | 2.03±0.17 | 0.11±0.01 | 2.52±0.20 | 73.48±0.61 |
| **GPR-GNN** | ✓ | | 4.42±0.43 | 0.63±0.06 | 4.35±0.22 | 74.90±2.34 | | 4.63±0.27 | 0.81±0.07 | 4.92±0.24 | 66.33±0.20 |
| **FAGCN** | ✓ | | 4.30±0.07 | 0.57±0.02 | 4.25±0.04 | 76.49±1.73 | | 4.07±0.15 | 0.58±0.02 | 4.23±0.09 | 71.82±0.73 |
| **GAT** | | | 1.60±0.10 | 0.07±0.01 | 1.83±0.05 | 79.88±2.49 | | 1.30±0.06 | 0.08±0.02 | 1.62±0.06 | 72.87±0.80 |
| **GCN** | | | 1.73±0.09 | 0.09±0.01 | 1.99±0.03 | 79.39±3.72 | | 1.77±0.08 | 0.14±0.02 | 2.09±0.07 | 73.48±0.53 |
| **$H_2$GCN** | ✓ | **FB100** | 8.12±0.10 | 1.76±0.02 | 8.14±0.06 | 57.38±0.17 | **Snap** | 1.44±0.18 | 0.59±0.10 | 3.79±0.40 | 26.97±0.10 |
| **GraphSAGE** | ✓ | | 6.98±0.06 | 1.50±0.04 | 7.32±0.13 | 56.72±1.56 | | 0.70±0.21 | 0.19±0.11 | 2.16±0.54 | 26.84±0.47 |
| **CPGNN** | ✓ | | 6.80±0.19 | 1.41±0.21 | 7.05±0.70 | 59.00±5.71 | | 1.45±0.23 | 0.61±0.14 | 3.89±0.51 | 26.71±0.25 |
| **GPR-GNN** | ✓ | | 5.81±0.16 | 1.11±0.02 | 5.95±0.10 | 61.99±0.44 | | 0.52±0.06 | 0.11±0.01 | 1.70±0.14 | 26.31±1.03 |
| **FAGCN** | ✓ | | 7.45±0.21 | 1.53±0.02 | 7.40±0.06 | 59.76±1.47 | | 1.41±0.10 | 0.56±0.06 | 3.81±0.22 | 27.07±0.16 |
| **GAT** | | | 4.30±0.26 | 0.77±0.04 | 4.72±0.19 | 61.56±0.78 | | 0.28±0.09 | 0.04±0.01 | 0.95±0.33 | 27.12±0.52 |
| **GCN** | | | 3.93±0.09 | 0.64±0.02 | 4.24±0.10 | 65.54±0.43 | | 0.33±0.06 | 0.03±0.00 | 1.17±0.25 | 26.79±0.39 |

more robust than methods without it, thus answering (Q3). For GNN models, we include $H_2$GCN, GraphSAGE, CPGNN, GPR-GNN, FAGCN, GAT and GCN in this analysis. We exclude other models that either learn to rewrite the graph structure through the training process or require a recalculation of the low-rank approximation for every randomized perturbation as sampling on these models is infeasible. We use the same hyperparameters as in the benchmark study in §5.2.

Table 4 shows multiple metrics of certifiable robustness of each GNN model under a randomization scheme allowing for both addition and deletion of edges; we additionally report results under randomization schemes allowing only addition or deletion in Table 9. For the scheme allowing both addition and deletion, we observe that all heterophily-adjusted methods have better certifiable robustness compared to methods without the design. Specifically, on homophilous datasets (Cora and Citeseer), methods with the identified design achieve an up to 1.6 times relative improvement in accumulated certification. On heterophilous datasets (FB100 and Snap), they outperform the baselines by a factor of 4.4. In the more challenging case with the addition only scheme, methods with the design also show up to 1.2 times relative increase in AC on the homophilous datasets and 5.0 times relative increase in AC on the heterophilous datasets compared to the baselines. For the deletion only scheme, unvaccinated models like GCN already have decent certifiable robustness in this scenario. This is commensurate with our discussions in §5.1 that deletions create less severe perturbations. Overall, our results show that models featuring our identified design achieve significantly improved *certifiable robustness* compared to models lacking this design. However, like in our empirical robustness evaluation, architectural differences lead to some variability of robustness; the results also show tradeoffs between accuracy and robustness. We also observe that the robustness rankings for methods under certifiable and empirical robustness are different, as in the previous results from Geisler et al. (2020); we discuss possible reasons in App. E.3.

# 6 Conclusion

We formalized the relation between heterophily and adversarial structural attacks, and showed theoretically and empirically that effective attacks gravitate towards increasing heterophily in both homophilous and heterophilous graphs by leveraging low-degree (gambit) nodes. Using these insights, we showed that a key design addressing heterophily, namely separate aggregators for ego- and neighbor-embeddings, can lead to competitive improvement on empirical and certifiable robustness, with only small influence on clean performance. Finally, we compared the design with state-of-the-art vaccination mechanisms under different attack scenarios for various datasets, and illustrated that they are complementary and that their combination can lead to more robust GNN models. We note that while we focus on the structural attacks, GNNs are also vulnerable to other types of attacks such as feature perturbations. In addition, the graph-agnostic MLP, which is immune to structural attacks, on some datasets outperforms all GNNs against attacks; this calls for further works in understanding of the nature of the attacks and also for effective defense strategies upon our discoveries.

REPRODUCIBILITY STATEMENT

For reproducibility, we describe the experimental setups in §5 with additional details in App. D.2, including the implementation and the hyperparameter settings used for each method, and the hardware specifications. We provide anonymized code and datasets in the supplementary material with instructions for replicating the experiments, and will make them publicly available upon acceptance.

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

## A  SOCIETAL IMPACTS

A large number of popular GNN models are inherently based on homophily. Our work shows that such models may be less robust to adversarial perturbations, and thus when employed for decision-making these models may lead to undesirable, erroneous or biased results. For example, an inherently homophilous GNN model may lead to the so-called "filter bubble" phenomenon in a recommendation system in which existing beliefs or preferences are reinforced. Similarly, as homophily-based GNNs typically average over node neighborhoods, this may result in less visibility of minority groups in the network, thus reinforcing disparities. Heterophilous network design may improve some of these aspects, and also benefit from some additional robustness as shown in this work.

However, it should be noted that heterophilous GNNs on their own cannot fully solve the aforementioned issues. In particular, while heterophilous design can improve robustness, it is not (in general) constructed to ensure other important aspects such as fairness. Better understanding of GNNs and tailored auditing tools are necessary in order to deploy these learning algorithms in the context of decision-making affecting humans.

## B  NOMENCLATURE

We summarize the main symbols used in this work and their definitions below:

Table 5: Major symbols and definitions.

| Symbols | Definitions |
|---|---|
| $\mathcal{G} = (\mathcal{V}, \mathcal{E}, \mathbf{X})$ | graph $\mathcal{G}$ with nodeset $\mathcal{V}$, edgeset $\mathcal{E}$, and $|\mathcal{V}| \times F$ node feature matrix $\mathbf{X}$ |
| $\mathbf{X}$ | $|\mathcal{V}| \times F$ node feature matrix of $\mathcal{G}$, $\mathbf{X} \in \mathbb{R}^{|\mathcal{V}| \times F}$ |
| $\mathbf{A}$ | $|\mathcal{V}| \times |\mathcal{V}|$ adjacency matrix of $\mathcal{G}$, $\mathbf{A} \in \{0, 1\}^{|\mathcal{V}| \times |\mathcal{V}|}$ |
| $\mathbf{A}_s$ | adjacency matrix with self-loops added, $\mathbf{A}_s = \mathbf{A} + \mathbf{I}$ |
| $\mathbf{D}$ | diagonal matrix of degrees, with $\mathbf{D}_{ii} = \sum_j \mathbf{A}_{ij}$ |
| $\bar{\mathbf{A}}$ | row-stochastic matrix for $\mathbf{A}$, $\bar{\mathbf{A}} = \mathbf{D}^{-1}\mathbf{A}$ |
| $\bar{\mathbf{A}}_s$ | row-stochastic matrix for $\mathbf{A}_s$, $\bar{\mathbf{A}}_s = \mathbf{D}_s^{-1}\mathbf{A}_s$ |
| $\tilde{\mathbf{A}}$ | low-rank approximation of the adjacency matrix $\mathbf{A}$ |
| $\mathbf{W}$ | learnable weight matrix for GNN models |
| $\mathbf{x}_v$ | $F$-dimensional feature vector for node $v$ |
| $N(v)$ | direct (1-hop) neighbors of node $v$ in $\mathcal{G}$ *without self-loops* (i.e., excluding $v$) |
| $\mathcal{Y}$ | set of class labels |
| $y_v$ | class label for node $v \in \mathcal{V}$ |
| $\mathbf{y}$ | $|\mathcal{V}|$-dimensional vector of class labels (for all the nodes) |
| $\mathcal{T}_{\mathcal{V}} = \{(v_1, y_1), (v_2, y_2), ...\}$ | training data for semi-supervised node classification |
| $\mathcal{G}' = (\mathcal{V}, \mathcal{E}', \mathbf{X})$ | graph $\mathcal{G}'$ with modified edgeset $\mathcal{E}'$ |
| $f$ | a certain GNN model that processes $\mathcal{G}$ |
| $K$ | the number of layers of GNN $f$ |
| $\mathbf{r}_v^{(k)}$ | node representations learned by GNN $f$ at round / layer $k$ |
| AGGR | function that aggregates node feature representations within a neighborhood |
| ENC | learnable (nonlinear) mapping that generates latent representation |
| $\mathbf{z}_v$ | label prediction by GNN $f$ for node $v \in \mathcal{V}$, $\mathbf{z}_v \in [0, 1]^{|\mathcal{Y}|}$ |
| $\mathcal{L}_{\text{atk}}$ | attack loss that quantifies the mismatch between $\mathbf{z}$ and the true labels $\mathbf{y}$ |
| $\mathcal{L}_{\text{atk}}^{\text{CM}}$ | attack loss defined with negative classification margin $\mathcal{L}_{\text{atk}}^{\text{CM}} = -\Delta_c = -(\mathbf{z}_{v,y_v} - \max_{y \neq y_v} \mathbf{z}_{v,y})$ |
| $d$ | the degree of the nodes which have a distance less than 2 to any $v \in \mathcal{D}_{\mathcal{V}}$ |
| $d_a$ | the degree of a gambit node leveraged by an attack |
| $p$ | parameter regulating the signal strength of one-hot class label vs. uniform noise |
| $\alpha$ | a predefined weight scalar in Theorem 3, $\alpha \in [0, 1]$ |
| $h$ | homophily ratio defined on node class labels, $h = |\{(u, v) \in \mathcal{E} | y_u = y_v\}| / |\mathcal{E}|$ |

## C  PROOFS AND DISCUSSIONS OF THEOREMS

### C.1  DETAILED ANALYSIS OF THEOREM 1

**Proof 1 (for Thm. 1)** *We give the proof in three parts: first, we analyze the training process of the GNN $f_s^{(2)}(\mathbf{A}, \mathbf{X}) = \bar{\mathbf{A}}_s^2 \mathbf{X} \mathbf{W}$ on clean data and analytically derive the optimal weight matrix $\mathbf{W}_*$ in a stylized learning setup; then, we construct a targeted evasion attack and calculate the attack loss for a unit structural attack; last, we summarize and validate the statements in the theorem.*

***Stylized learning on clean data.*** *Given the 2-layer linearized GNN $f_s^{(2)}(\mathbf{A}, \mathbf{X}) = \bar{\mathbf{A}}_s^2 \mathbf{X} \mathbf{W}$ and the training set $\mathcal{T}_{\mathcal{V}} \subseteq \mathcal{D}_{\mathcal{V}}$, the goal of the training process is to optimize the weight matrix $\mathbf{W}$ to minimize the cross-entropy loss function $\mathcal{L}([\mathbf{z}]_{\mathcal{T}_{\mathcal{V}},:}, [\mathbf{Y}]_{\mathcal{T}_{\mathcal{V}},:})$, where predictions $[\mathbf{z}]_{\mathcal{T}_{\mathcal{V}},:} = [\bar{\mathbf{A}}_s^2 \mathbf{X}]_{\mathcal{T}_{\mathcal{V}},:} \mathbf{W}$ correspond to the predicted class label distributions for each node $v$ in the training set $\mathcal{T}_{\mathcal{V}}$, and $[\mathbf{Y}]_{\mathcal{T}_{\mathcal{V}},:}$ is the one-hot encoding of class labels provided in the training set.*

*Without loss of generality, we reorder $\mathcal{T}_{\mathcal{V}}$ accordingly such that the one-hot encoding of labels for nodes in the training set $[\mathbf{Y}]_{\mathcal{T}_{\mathcal{V}},:}$ is in increasing order of the class label $y_v$:*

$$[\mathbf{Y}]_{\mathcal{T}_{\mathcal{V}},:} = \left[\begin{array}{ccccc} 1 & 0 & 0 & \cdots & 0 \\ \vdots & \vdots & \vdots & \ddots & \vdots \\ 1 & 0 & 0 & \cdots & 0 \\ \hdashline 0 & 1 & 0 & \cdots & 0 \\ \vdots & \vdots & \vdots & \ddots & \vdots \\ 0 & 1 & 0 & \cdots & 0 \\ \hdashline \vdots & \vdots & \vdots & \ddots & \vdots \\ \hdashline 0 & 0 & 0 & \cdots & 1 \\ \vdots & \vdots & \vdots & \ddots & \vdots \\ 0 & 0 & 0 & \cdots & 1 \end{array}\right]_{|\mathcal{T}_{\mathcal{V}}| \times |\mathcal{Y}|} \tag{2}$$

*Now we look at the term $[\bar{\mathbf{A}}_s^2 \mathbf{X}]_{\mathcal{T}_{\mathcal{V}},:}$ in $[\mathbf{z}]_{\mathcal{T}_{\mathcal{V}},:} = [\bar{\mathbf{A}}_s^2 \mathbf{X}]_{\mathcal{T}_{\mathcal{V}},:} \mathbf{W}$, which are the feature vectors aggregated by the two GNN layers for nodes $v$ in the training set $\mathcal{T}_{\mathcal{V}}$. As stated in the theorem, we assume $\mathcal{T}_{\mathcal{V}} \subseteq \mathcal{D}_{\mathcal{V}}$, where node $u \in \mathcal{D}_{\mathcal{V}}$ have degree $d$; proportion $h$ of their neighbors belong to the same class, while proportion $\frac{1-h}{|\mathcal{Y}|-1}$ of them belong to any other class uniformly, and for each node $v \in \mathcal{V}$ the node features are given as $\mathbf{x}_v = p \cdot \mathrm{onehot}(y_v) + \frac{1-p}{|\mathcal{Y}|} \cdot \mathbf{1}$ for each node $v \in \mathcal{V}$. Then, after the first layer, we have:*

$$[\bar{\mathbf{A}}_s \mathbf{X}]_{\mathcal{T}_{\mathcal{V}},:} = \frac{1}{d+1} \left[\begin{array}{ccccc} (hd+1)p & \frac{1-h}{|\mathcal{Y}|-1}dp & \frac{1-h}{|\mathcal{Y}|-1}dp & \cdots & \frac{1-h}{|\mathcal{Y}|-1}dp \\ \vdots & \vdots & \vdots & \ddots & \vdots \\ (hd+1)p & \frac{1-h}{|\mathcal{Y}|-1}dp & \frac{1-h}{|\mathcal{Y}|-1}dp & \cdots & \frac{1-h}{|\mathcal{Y}|-1}dp \\ \hdashline \frac{1-h}{|\mathcal{Y}|-1}dp & (hd+1)p & \frac{1-h}{|\mathcal{Y}|-1}dp & \cdots & \frac{1-h}{|\mathcal{Y}|-1}dp \\ \vdots & \vdots & \vdots & \ddots & \vdots \\ \frac{1-h}{|\mathcal{Y}|-1}dp & (hd+1)p & \frac{1-h}{|\mathcal{Y}|-1}dp & \cdots & \frac{1-h}{|\mathcal{Y}|-1}dp \\ \hdashline \vdots & \vdots & \vdots & \ddots & \vdots \\ \hdashline \frac{1-h}{|\mathcal{Y}|-1}dp & \frac{1-h}{|\mathcal{Y}|-1}dp & \frac{1-h}{|\mathcal{Y}|-1}dp & \cdots & (hd+1)p \\ \vdots & \vdots & \vdots & \ddots & \vdots \\ \frac{1-h}{|\mathcal{Y}|-1}dp & \frac{1-h}{|\mathcal{Y}|-1}dp & \frac{1-h}{|\mathcal{Y}|-1}dp & \cdots & (hd+1)p \end{array}\right]_{|\mathcal{T}_{\mathcal{V}}| \times |\mathcal{Y}|} + \frac{1-p}{|\mathcal{Y}|} \tag{3}$$

*and after the second layer*

$$[\bar{\mathbf{A}}_{\mathrm{s}}^2 \mathbf{X}]_{\mathcal{T}_\mathcal{V},:} = \frac{1}{(d+1)^2 |\mathcal{Y}|(|\mathcal{Y}|-1)} \begin{bmatrix} S_1 & T_1 & T_1 & \cdots & T_1 \\ \vdots & \vdots & \vdots & \ddots & \vdots \\ S_1 & T_1 & T_1 & \cdots & T_1 \\ \hline T_1 & S_1 & T_1 & \cdots & T_1 \\ \vdots & \vdots & \vdots & \ddots & \vdots \\ T_1 & S_1 & T_1 & \cdots & T_1 \\ \hline \vdots & \vdots & \vdots & \ddots & \vdots \\ \hline T_1 & T_1 & T_1 & \cdots & S_1 \\ \vdots & \vdots & \vdots & \ddots & \vdots \\ T_1 & T_1 & T_1 & \cdots & S_1 \end{bmatrix}_{|\mathcal{T}_\mathcal{V}| \times |\mathcal{Y}|} + \frac{1}{|\mathcal{Y}|}, \qquad (4)$$

*where $S_1 = ((h|\mathcal{Y}|-1)d + |\mathcal{Y}| - 1)^2 p$, and $T_1 = \frac{((h|\mathcal{Y}|-1)d+|\mathcal{Y}|-1)^2 p}{|\mathcal{Y}|-1}$.*

*For $[\mathbf{Y}]_{\mathcal{T}_\mathcal{V},:}$ and $[\bar{\mathbf{A}}_{\mathrm{s}}^2 \mathbf{X}]_{\mathcal{T}_\mathcal{V},:}$ which we derived in Eq. (2) and (4), we can find the optimal weight matrix $\mathbf{W}_*$ such that $[\bar{\mathbf{A}}_{\mathrm{s}}^2 \mathbf{X}]_{\mathcal{T}_\mathcal{V},:} \mathbf{W}_* = [\mathbf{Y}]_{\mathcal{T}_\mathcal{V},:}$, making the cross-entropy loss $\mathcal{L}([\mathbf{z}]_{\mathcal{T}_\mathcal{V},:}, [\mathbf{Y}]_{\mathcal{T}_\mathcal{V},:}) = 0$.*

*To find $\mathbf{W}_*$, we can proceed as follows. First, sample one node from each class to form a smaller set $\mathcal{T}_S \subset \mathcal{T}_\mathcal{V}$. Therefore, we have:*

$$[\mathbf{Y}]_{\mathcal{T}_S,:} = \begin{bmatrix} 1 & 0 & 0 & \cdots & 0 \\ 0 & 1 & 0 & \cdots & 0 \\ \vdots & \vdots & \vdots & \ddots & \vdots \\ 0 & 0 & 0 & \cdots & 1 \end{bmatrix}_{|\mathcal{Y}| \times |\mathcal{Y}|} = \mathbf{I}_{|\mathcal{Y}| \times |\mathcal{Y}|}$$

*and*

$$[\bar{\mathbf{A}}_{\mathrm{s}}^2 \mathbf{X}]_{\mathcal{T}_S,:} = \frac{1}{(d+1)^2 |\mathcal{Y}|(|\mathcal{Y}|-1)} \begin{bmatrix} S_1 & T_1 & T_1 & \cdots & T_1 \\ T_1 & S_1 & T_1 & \cdots & T_1 \\ \vdots & \vdots & \vdots & \ddots & \vdots \\ T_1 & T_1 & T_1 & \cdots & S_1 \end{bmatrix}_{|\mathcal{Y}| \times |\mathcal{Y}|} + \frac{1}{|\mathcal{Y}|}.$$

*Note that $[\bar{\mathbf{A}}_{\mathrm{s}}^2 \mathbf{X}]_{\mathcal{T}_S,:}$ is a specific circulant matrix, and therefore its inverse exists. Using the Sherman-Morrison formula, we can find its inverse as:*

$$\left([\bar{\mathbf{A}}_{\mathrm{s}}^2 \mathbf{X}]_{\mathcal{T}_S,:}\right)^{-1} = \frac{(d+1)^2(|\mathcal{Y}|-1)^2}{p(d(h|\mathcal{Y}|-1)+|\mathcal{Y}|-1)^2 |\mathcal{Y}|} \cdot \begin{bmatrix} |\mathcal{Y}|-1 & -1 & \cdots & -1 \\ -1 & |\mathcal{Y}|-1 & \cdots & -1 \\ \vdots & \vdots & \ddots & \vdots \\ -1 & -1 & \cdots & |\mathcal{Y}|-1 \end{bmatrix} + \frac{1}{|\mathcal{Y}|}. \qquad (5)$$

*Now, let $\mathbf{W}_* = \left([\bar{\mathbf{A}}_{\mathrm{s}}^2 \mathbf{X}]_{\mathcal{T}_S,:}\right)^{-1}$, then we have $[\mathbf{z}]_{\mathcal{T}_S,:} = [\bar{\mathbf{A}}_{\mathrm{s}}^2 \mathbf{X}]_{\mathcal{T}_S,:} \mathbf{W}_* = [\mathbf{Y}]_{\mathcal{T}_S,:} = \mathbf{I}_{|\mathcal{Y}| \times |\mathcal{Y}|}$. It is also easy to verify that $[\mathbf{z}]_{\mathcal{T}_\mathcal{V},:} = [\bar{\mathbf{A}}_{\mathrm{s}}^2 \mathbf{X}]_{\mathcal{T}_\mathcal{V},:} \mathbf{W}_* = [\mathbf{Y}]_{\mathcal{T}_\mathcal{V},:}$. Since $\mathbf{W}_*$ satisfies $\mathcal{L}([\mathbf{z}]_{\mathcal{T}_\mathcal{V},:}, [\mathbf{Y}]_{\mathcal{T}_\mathcal{V},:}) = 0$, we know $\mathbf{W}_* = \left([\bar{\mathbf{A}}_{\mathrm{s}}^2 \mathbf{X}]_{\mathcal{T}_S,:}\right)^{-1}$ is the optimal weight matrix that we can learn under $\mathcal{T}_\mathcal{V}$.*

***Attack loss under evasion attacks.*** *Now consider an arbitrary target node $v \in \mathcal{D}_\mathcal{V}$ with class label $y_v \in \mathcal{Y}$, and a unit structural perturbation leveraging gambit node $u \in \mathcal{V}$ with degree $d_a$ that affects the predictions $\mathbf{z}_v$ of node $v$ made by GNN $f_s^{(2)}$. Without loss of generality, we assume node $v$ has $y_v = 1$. As $f_s^{(2)}$ contains 2 GNN layers with each layer aggregating feature vectors within neighborhood $N(v)$ of each node $v$, the perturbation must take place in the direct (1-hop) neighborhood $N(v)$ or 2-hop neighborhood $N_2(v)$ to affect the predictions $\mathbf{z}_v$. For the unit perturbation, the*

*attacker can add or remove a homophilous edge or path between nodes $u$ and $v$, which we denote as $\delta_1$ ($\delta_1 = 1$ for addition and $\delta_1 = -1$ for removal); alternatively, the attacker can add or remove a heterophilous edge or path between nodes $u$ and $v$, which we denote as $\delta_2 = \pm 1$ analogously. We denote the perturbed graph adjacency matrix as $\bar{\mathbf{A}}'_s$, and $\mathbf{z}'_v = [\bar{\mathbf{A}}'^2_s \mathbf{X}]_{v,:} \mathbf{W}_*$*

①  *Unit perturbation in direct neighborhood $N(v)$. We first consider a unit perturbation in the direct (1-hop) neighborhood $N(v)$ of node $v$. For simplicity of derivation, we assume that the perturbation does not change the row-stochastic normalization of $\bar{\mathbf{A}}_s$, and only affects the aggregated feature vectors of the target node $v$.*

*In the case of $\delta_1 = \pm 1$ and $\delta_2 = 0$, we have*

$$[\bar{\mathbf{A}}'_s\mathbf{X}]_{v,:} - [\bar{\mathbf{A}}_s\mathbf{X}]_{v,:} = \frac{1}{d_a+1}\left[\ \delta_1\left(\frac{1-p}{|\mathcal{Y}|}+p\right) \quad \delta_1\left(\frac{1-p}{|\mathcal{Y}|}\right) \quad \cdots \quad \delta_1\left(\frac{1-p}{|\mathcal{Y}|}\right)\ \right]$$

*and*

$$[\bar{\mathbf{A}}'^2_s\mathbf{X}]_{v,:} - [\bar{\mathbf{A}}^2_s\mathbf{X}]_{v,:} = \frac{1}{(d_a+1)^2(d+1)|\mathcal{Y}|}\left[\ S_2 \quad \frac{T_2}{|\mathcal{Y}|-1} \quad \cdots \quad \frac{T_2}{|\mathcal{Y}|-1}\ \right],$$

*where $S_2 = \delta_1\left(d_a(p(d(h|\mathcal{Y}|-1)+h|\mathcal{Y}|+|\mathcal{Y}|-2)+d+2)+(d+2)(|\mathcal{Y}|-1)p+d+2\right)$ and $T_2 = \delta_1\left(-\left(d_a(p(d(h|\mathcal{Y}|-1)+h|\mathcal{Y}|+|\mathcal{Y}|-2)+(-d-2)(|\mathcal{Y}|-1))+(d+2)(|\mathcal{Y}|-1)(p-1)\right)\right)$. By Multiplying $[\bar{\mathbf{A}}'^2_s\mathbf{X}]_{v,:}$ by $\mathbf{W}_*$, we can get the predictions $\mathbf{z}'_v$ after perturbations; we omit the analytical expression of $\mathbf{z}'_v$ here due to its complexity.*

*On the perturbed graph, the CM-type attack loss is calculated as*

$$\mathcal{L}^{\mathrm{CM}}_{\mathrm{atk}}(\mathbf{z}'_v) = -(\mathbf{z}'_{v,y_v} - \max_{y \neq y_v} \mathbf{z}'_{v,y})$$

*Since on clean data, $\mathcal{L}^{\mathrm{CM}}_{\mathrm{atk}}(\mathbf{z}_v) = -1$, the change in attack loss before and after attack is*

$$\Delta\mathcal{L}^{\mathrm{CM}}_{\mathrm{atk}} = \mathcal{L}^{\mathrm{CM}}_{\mathrm{atk}}(\mathbf{z}'_v) - \mathcal{L}^{\mathrm{CM}}_{\mathrm{atk}}(\mathbf{z}_v) = \mathcal{L}^{\mathrm{CM}}_{\mathrm{atk}}(\mathbf{z}'_v) + 1 \tag{6}$$

$$= -\frac{(d+1)\delta_1(|\mathcal{Y}|-1)\left(d_a(d(h|\mathcal{Y}|-1)+h|\mathcal{Y}|+|\mathcal{Y}|-2)+(d+2)(|\mathcal{Y}|-1)\right)}{(d_a+1)^2(d(h|\mathcal{Y}|-1)+|\mathcal{Y}|-1)^2}. \tag{7}$$

*Solving following system of inequalities for $\delta_1$,*

$$\begin{cases} \Delta\mathcal{L}^{\mathrm{CM}}_{\mathrm{atk}} > 0 \\ h \in [0,1] \\ |\mathcal{Y}| \geq 2 \\ d, d_a, |\mathcal{Y}| \in \mathbb{Z}^+ \end{cases} \tag{8}$$

*we get the valid range of $\delta_1$ as*

$$\begin{cases} \delta_1 < 0, \text{ when } 0 < d \leq |\mathcal{Y}| - 2 \\ \delta_1 < 0, \text{ when } d > |\mathcal{Y}| - 2 \text{ and } d_a < \frac{(d+2)(|\mathcal{Y}|-1)}{d-|\mathcal{Y}|+2} \\ \delta_1 < 0, \text{ when } d > |\mathcal{Y}| - 2 \text{ and } d_a \geq \frac{(d+2)(|\mathcal{Y}|-1)}{d-|\mathcal{Y}|+2} \text{ and } 1 \geq h > \frac{d_a(d-|\mathcal{Y}|+2)-(d+2)(|\mathcal{Y}|-1)}{(d+1)|\mathcal{Y}|d_a} \\ \delta_1 > 0, \text{ when } d > |\mathcal{Y}| - 2 \text{ and } d_a \geq \frac{(d+2)(|\mathcal{Y}|-1)}{d-|\mathcal{Y}|+2} \text{ and } 0 \leq h < \frac{d_a(d-|\mathcal{Y}|+2)-(d+2)(|\mathcal{Y}|-1)}{(d+1)|\mathcal{Y}|d_a} \end{cases} . \tag{9}$$

*Note that the above solution is not applicable when $h = \frac{d-|\mathcal{Y}|+1}{d|\mathcal{Y}|}$, in which case $d(h|\mathcal{Y}|-1)+|\mathcal{Y}|-1 = 0$ and the solution of optimal weight matrix $\mathbf{W}_* = \left([\bar{\mathbf{A}}^2_s\mathbf{X}]_{\mathcal{T}_S,:}\right)^{-1}$ is undefined.*

*In the case of $\delta_1 = 0$ and $\delta_2 = \pm 1$, we have*

$$[\bar{\mathbf{A}}'_s\mathbf{X}]_{v,:} - [\bar{\mathbf{A}}_s\mathbf{X}]_{v,:} = \frac{1}{d_a+1}\left[\ \delta_2\left(\frac{1-p}{|\mathcal{Y}|}\right) \quad \delta_2\left(\frac{1-p}{|\mathcal{Y}|}+p\right) \quad \cdots \quad \delta_2\left(\frac{1-p}{|\mathcal{Y}|}+p\right)\ \right]$$

*and*

$$[\bar{\mathbf{A}}'^2_s\mathbf{X}]_{v,:} - [\bar{\mathbf{A}}^2_s\mathbf{X}]_{v,:} = \frac{1}{(d_a+1)^2(d+1)|\mathcal{Y}|}\left[\ \frac{S_4}{|\mathcal{Y}|-1} \quad T_4 \quad \cdots \quad T_4\ \right],$$

*where $S_4 = \delta_2\left(-\left(d_a(p(d(h|\mathcal{Y}|-1)+h|\mathcal{Y}|+|\mathcal{Y}|-2)+(-d-2)(|\mathcal{Y}|-1))+(d+2)(|\mathcal{Y}|-1)(p-1)\right)\right)$ and $T_4 = \delta_2\left(d_a(p(d(h|\mathcal{Y}|-1)+h|\mathcal{Y}|+|\mathcal{Y}|-2)+d+2)+(d+2)(|\mathcal{Y}|-1)p+d+2\right)$. By*

multiplying $[\bar{\mathbf{A}}_s'^2\mathbf{X}]_{v,:}$ with $\mathbf{W}_*$, we can get the predictions $\mathbf{z}_v'$ after the perturbations. Following a similar derivation to that in the previous case, we can compute the change in the CM-type attack loss before and after attack as

$$\Delta\mathcal{L}_{\text{atk}}^{\text{CM}} = \frac{(d+1)(|\mathcal{Y}|-1)\left(d_a(d(h|\mathcal{Y}|-1)+h|\mathcal{Y}|+|\mathcal{Y}|-2)+(d+2)(|\mathcal{Y}|-1)\right)\delta_2}{(d_a+1)^2(d(h|\mathcal{Y}|-1)+|\mathcal{Y}|-1)^2}. \quad (10)$$

Solving the same system of inequalities as Eq. (8) for $\delta_2$, we obtain the valid range of $\delta_2$ as

$$\begin{cases} \delta_2 > 0, \text{ when } 0 < d \le |\mathcal{Y}|-2 \\ \delta_2 > 0, \text{ when } d > |\mathcal{Y}|-2 \text{ and } d_a < \frac{(d+2)(|\mathcal{Y}|-1)}{d-|\mathcal{Y}|+2} \\ \delta_2 > 0, \text{ when } d > |\mathcal{Y}|-2 \text{ and } d_a \ge \frac{(d+2)(|\mathcal{Y}|-1)}{d-|\mathcal{Y}|+2} \text{ and } 1 \ge h > \frac{d_a(d-|\mathcal{Y}|+2)-(d+2)(|\mathcal{Y}|-1)}{(d+1)|\mathcal{Y}|d_a} \\ \delta_2 < 0, \text{ when } d > |\mathcal{Y}|-2 \text{ and } d_a \ge \frac{(d+2)(|\mathcal{Y}|-1)}{d-|\mathcal{Y}|+2} \text{ and } 0 \le h < \frac{d_a(d-|\mathcal{Y}|+2)-(d+2)(|\mathcal{Y}|-1)}{(d+1)|\mathcal{Y}|d_a} \end{cases}. \quad (11)$$

Note that the above solution is not applicable when $h = \frac{d-|\mathcal{Y}|+1}{d|\mathcal{Y}|}$, in which case $d(h|\mathcal{Y}|-1)+|\mathcal{Y}|-1 = 0$ and the solution of optimal weight matrix $\mathbf{W}_* = \left([\bar{\mathbf{A}}_s^2\mathbf{X}]_{\mathcal{T}_S,:}\right)^{-1}$ is undefined. We also note that we always have $\frac{d_a(d-|\mathcal{Y}|+2)-(d+2)(|\mathcal{Y}|-1)}{(d+1)|\mathcal{Y}|d_a} - \frac{1}{|\mathcal{Y}|} = -\frac{(|\mathcal{Y}|-1)(d_a+d+2)}{(d+1)|\mathcal{Y}|d_a} < 0$ when $|\mathcal{Y}| \ge 2$.

② Unit perturbation in 2-hop neighborhood $N_2(v)$. We now consider a unit perturbation in the 2-hop neighborhood $N(v)$ of node $v$. In this case we will have $[\bar{\mathbf{A}}_s'\mathbf{X}]_{v,:} = [\bar{\mathbf{A}}_s\mathbf{X}]_{v,:}$.

In the case of $\delta_1 = \pm 1$ and $\delta_2 = 0$, we have

$$[\bar{\mathbf{A}}_s'^2\mathbf{X}]_{v,:} - [\bar{\mathbf{A}}_s^2\mathbf{X}]_{v,:} = \frac{1}{(d_a+1)^2|\mathcal{Y}|} \left[ \begin{array}{cccc} S_5 & \frac{T_5}{|\mathcal{Y}|-1} & \cdots & \frac{T_5}{|\mathcal{Y}|-1} \end{array} \right],$$

where

$$S_5 = (d_a(p(h|\mathcal{Y}|-1)+1)+(|\mathcal{Y}|-1)p+1)\delta_1$$

and

$$T_5 = (d_a(-h|\mathcal{Y}|p+|\mathcal{Y}|+p-1)+|\mathcal{Y}|(-p)+|\mathcal{Y}|+p-1)\delta_1.$$

By multiplying $[\bar{\mathbf{A}}_s'^2\mathbf{X}]_{v,:}$ with $\mathbf{W}_*$, we can get the predictions $\mathbf{z}_v'$ after perturbations. Following a similar derivation as before, we can get the change in the CM-type attack loss before and after attack as

$$\Delta\mathcal{L}_{\text{atk}}^{\text{CM}} = -\frac{(d+1)^2(|\mathcal{Y}|-1)\left(d_a(h|\mathcal{Y}|-1)+|\mathcal{Y}|-1\right)\delta_1}{(d_a+1)^2(d(h|\mathcal{Y}|-1)+|\mathcal{Y}|-1)^2}. \quad (12)$$

Solving the same system of inequalities as Eq. (8) for $\delta_1$, we get the valid range of $\delta_1$ as

$$\begin{cases} \delta_1 < 0, \text{ when } d_a < |\mathcal{Y}|-1 \\ \delta_1 < 0, \text{ when } d_a \ge |\mathcal{Y}|-1 \text{ and } \frac{d_a-|\mathcal{Y}|+1}{|\mathcal{Y}|d_a} < h \le 1 \\ \delta_1 > 0, \text{ when } d_a > |\mathcal{Y}|-1 \text{ and } 0 \le h < \frac{d_a-|\mathcal{Y}|+1}{|\mathcal{Y}|d_a} \end{cases}. \quad (13)$$

Note that the above solution is not applicable when $h = \frac{d-|\mathcal{Y}|+1}{d|\mathcal{Y}|}$, in which case $d(h|\mathcal{Y}|-1)+|\mathcal{Y}|-1 = 0$ and the solution of optimal weight matrix $\mathbf{W}_* = \left([\bar{\mathbf{A}}_s^2\mathbf{X}]_{\mathcal{T}_S,:}\right)^{-1}$ is undefined.

For the case $\delta_1 = 0$ and $\delta_2 = \pm 1$, we have

$$[\bar{\mathbf{A}}_s'^2\mathbf{X}]_{v,:} - [\bar{\mathbf{A}}_s^2\mathbf{X}]_{v,:} = \frac{1}{(d_a+1)^2|\mathcal{Y}|} \left[ \begin{array}{cccc} \frac{S_6}{|\mathcal{Y}|-1} & T_6 & \cdots & T_6 \end{array} \right]$$

where

$$S_6 = \delta_2(d_a(-h|\mathcal{Y}|p+|\mathcal{Y}|+p-1)+|\mathcal{Y}|(-p)+|\mathcal{Y}|+p-1)$$

and

$$T_6 = \delta_2(d_a(p(h|\mathcal{Y}|-1)+1)+(|\mathcal{Y}|-1)p+1).$$

Multiplying $[\bar{\mathbf{A}}_s'^2\mathbf{X}]_{v,:}$ with $\mathbf{W}_*$, we can get the predictions $\mathbf{z}_v'$ after perturbations. As before we can compute the change in the CM-type attack loss before and after attack as

$$\Delta\mathcal{L}_{\text{atk}}^{\text{CM}} = \frac{(d+1)^2(|\mathcal{Y}|-1)\left(d_a(h|\mathcal{Y}|-1)+|\mathcal{Y}|-1\right)\delta_2}{(d_a+1)^2(d(h|\mathcal{Y}|-1)+|\mathcal{Y}|-1)^2} \quad (14)$$

*Finally, solving the same system of inequalities as Eq. (8) for $\delta_2$, we get the valid range of $\delta_2$ as*

$$
\begin{cases}
\delta_2 > 0, & \text{when } d_a < |\mathcal{Y}| - 1 \\
\delta_2 > 0, & \text{when } d_a \geq |\mathcal{Y}| - 1 \text{ and } \frac{d_a - |\mathcal{Y}| + 1}{|\mathcal{Y}| d_a} < h \leq 1 \\
\delta_2 < 0, & \text{when } d_a > |\mathcal{Y}| - 1 \text{ and } 0 \leq h < \frac{d_a - |\mathcal{Y}| + 1}{|\mathcal{Y}| d_a}
\end{cases}
. \tag{15}
$$

*Note that the above solution is not applicable when $h = \frac{d - |\mathcal{Y}| + 1}{d|\mathcal{Y}|}$, in which case $d(h|\mathcal{Y}| - 1) + |\mathcal{Y}| - 1 = 0$ and the solution of optimal weight matrix $\mathbf{W}_* = \left( [\bar{\mathbf{A}}_s^2 \mathbf{X}]_{\mathcal{T}_S,:} \right)^{-1}$ is undefined. We also note that we always have $\frac{d_a - |\mathcal{Y}| + 1}{|\mathcal{Y}| d_a} - \frac{1}{|\mathcal{Y}|} = \frac{1 - |\mathcal{Y}|}{|\mathcal{Y}| d_a} < 0$ when $|\mathcal{Y}| \geq 2$.*

***Summary and validation of theorem statements.*** *Based on our derivations, we summarize and validate our statements in the theorem next.*

① The attack losses $\mathcal{L}_{\text{atk}}$ (CM-type, §2) increase *only* by removing a homophilous edge or path, or adding a heterophilous edge or path to node $v$. *From Eq. (9), (11), (13), (15), we observe that for both direct attacks in 1-hop neighborhood $N(v)$ and indirect attacks in 2-hop neighborhood $N_2(v)$, when $h \geq \frac{1}{|\mathcal{Y}|}$, the attack loss increases only if $\delta_1 < 0$, which represents removal of a homophilous edge or path to node $v$, or if $\delta_2 > 0$, which represents addition of a heterophilous edge or path to node $v$.*

② Direct perturbations on edges (or 1-hop paths) of the target node $v$ lead to greater increase in $\mathcal{L}_{\text{atk}}$ than indirect perturbations on multi-hop paths to target node $v$.

*From Eq. (7) and Eq. (10), the change in the CM-type attack loss $\Delta \mathcal{L}_{\text{atk}}^{\text{CM}}(\mathbf{z}'_v)$ for direct perturbations on 1-hop neighborhood $N(v)$ of the target node $v$ considering both $\delta_1$ and $\delta_2$ can be written as*

$$
\Delta \mathcal{L}_{\text{atk}}^{\text{CM,direct}} = \frac{(d+1)(|\mathcal{Y}| - 1)\left(d_a(d(h|\mathcal{Y}| - 1) + h|\mathcal{Y}| + |\mathcal{Y}| - 2) + (d+2)(|\mathcal{Y}| - 1)\right)(\delta_2 - \delta_1)}{(d_a + 1)^2 (d(h|\mathcal{Y}| - 1) + |\mathcal{Y}| - 1)^2}
\tag{16}
$$

*From Eq. (12) and Eq. (14), the change in the CM-type attack loss $\Delta \mathcal{L}_{\text{atk}}^{\text{CM}}(\mathbf{z}'_v)$ for indirect perturbations on 2-hop neighborhood $N_2(v)$ of the target node $v$ considering both $\delta_1$ and $\delta_2$ is*

$$
\Delta \mathcal{L}_{\text{atk}}^{\text{CM,indirect}} = \frac{(d+1)^2 (|\mathcal{Y}| - 1)\left(d_a(h|\mathcal{Y}| - 1) + |\mathcal{Y}| - 1\right)(\delta_2 - \delta_1)}{(d_a + 1)^2 (d(h|\mathcal{Y}| - 1) + |\mathcal{Y}| - 1)^2}
\tag{17}
$$

*Note that when $h \geq \frac{1}{|\mathcal{Y}|}$, we have $h|\mathcal{Y}| \geq 1$, and*

$$
\frac{dh|\mathcal{Y}| - d + 2|\mathcal{Y}| - 2}{d(h|\mathcal{Y}| - 1) + |\mathcal{Y}| - 1} = 1 + \frac{d_a(d(h|\mathcal{Y}| - 1) + |\mathcal{Y}| - 1) + (d+1)(|\mathcal{Y}| - 1)}{d_a(h|\mathcal{Y}| - 1) + |\mathcal{Y}| - 1} > 1
\tag{18}
$$

*Therefore we will always have $\Delta \mathcal{L}_{\text{atk}}^{\text{CM,direct}} > \Delta \mathcal{L}_{\text{atk}}^{\text{CM,indirect}}$ for an effective unit perturbation that increases attack loss $\mathcal{L}_{\text{atk}}^{\text{CM}}$ (i.e., $\delta_1 = -1$ and $\delta_2 = 0$, or $\delta_1 = 0$ and $\delta_2 = 1$) when $h \geq \frac{1}{|\mathcal{Y}|}$.* ∎

## C.2 DETAILED ANALYSIS OF THEOREM 2

**Proof 2 (for Thm. 2)** *For a direct unit perturbation in the 1-hop neighborhood $N(v)$ of the target node $v$, from Eq. (9) and (11) of Proof 1, we observe that the signs of $\delta_1$ and $\delta_2$ which increase the attack loss are contingent on the degree of the target node $d$, the degree of the gambit node $d_a$ and the homophily ratio $h$ of the graph:*

① *if $0 < d \leq |\mathcal{Y}| - 2$ (i.e., when degree $d$ of the target node is low), regardless of $d_a$ and $h$, the attack loss increases only if $\delta_1 < 0$, which represents removal of a homophilous edge to node $v$, or if $\delta_2 > 0$, which represents addition of a heterophilous edge to $v$;*

② *if $d > |\mathcal{Y}| - 2$ (i.e., when degree $d$ of the target node is high), the increase of the attack loss will be dependent to the degree of the gambit node $d_a$ and the homophily ratio $h$ of the graph:*

(a) when $d_a < \frac{(d+2)(|\mathcal{Y}|-1)}{d-|\mathcal{Y}|+2}$ (i.e., when degree $d_a$ of the gambit node is low), regardless of $h$, the attack loss increases only if $\delta_1 < 0$, which represents removal of a homophilous edge to node $v$, or if $\delta_2 > 0$, which represents addition of a heterophilous edge to $v$;

(b) when $d_a \geq \frac{(d+2)(|\mathcal{Y}|-1)}{d-|\mathcal{Y}|+2}$ (i.e., when degree $d_a$ of the gambit node is high), for $0 \leq h < \frac{d_a(d-|\mathcal{Y}|+2)-(d+2)(|\mathcal{Y}|-1)}{(d+1)|\mathcal{Y}|d_a} < \frac{1}{|\mathcal{Y}|}$, the attack loss increases only if $\delta_1 < 0$, which represents removal of a homophilous edge to node $v$, or if $\delta_2 > 0$, which represents addition of a heterophilous edge to $v$.

We note that the above conclusions are not applicable when $h = \frac{d-|\mathcal{Y}|+1}{d|\mathcal{Y}|}$, in which case $d(h|\mathcal{Y}| - 1) + |\mathcal{Y}| - 1 = 0$ and the solution of optimal weight matrix $\mathbf{W}_* = \left([\bar{\mathbf{A}}_s^2\mathbf{X}]_{\mathcal{T}_S,:}\right)^{-1}$ is undefined.

∎

### C.3   Detailed Analysis of Theorem 3

**Proof 3 (for Thm. 3)** *In this proof, we mainly focus on analyzing the increase in $\mathcal{L}_{\mathrm{atk}}$ for the GNN layer defined as $f(\mathbf{A}, \mathbf{X}; \alpha) = \left((1-\alpha)\bar{\mathbf{A}} + \alpha\mathbf{I}\right)\mathbf{X}\mathbf{W}$. We follow a similar process as in Proof 1, since the layer defined as $f_s(\mathbf{A}, \mathbf{X}) = \bar{\mathbf{A}}_s\mathbf{X}\mathbf{W}$ is a special case of the previous formulation when $\alpha = \frac{1}{1+d}$.*

***Layer** $f(\mathbf{A}, \mathbf{X}; \alpha) = \left((1-\alpha)\bar{\mathbf{A}} + \alpha\mathbf{I}\right)\mathbf{X}\mathbf{W}$. We first derive the optimal weight matrix $\mathbf{W}_*$ in a stylized learning setup as in Proof 1. Following a similar process, for this GNN layer we have*

$$\left[\left((1-\alpha)\bar{\mathbf{A}} + \alpha\mathbf{I}\right)\mathbf{X}\right]_{\mathcal{T}_S,:} = \frac{1-p}{|\mathcal{Y}|} + \begin{bmatrix} S_7 & T_7 & T_7 & \cdots & T_7 \\ T_7 & S_7 & T_7 & \cdots & T_7 \\ \vdots & \vdots & \vdots & \ddots & \vdots \\ T_7 & T_7 & T_7 & \cdots & S_7 \end{bmatrix}_{|\mathcal{Y}|\times|\mathcal{Y}|}$$

*where $S_7 = p(\alpha + h - \alpha h)$, and $T_7 = \frac{(\alpha-1)(h-1)p}{|\mathcal{Y}|-1}$, and*

$$[\mathbf{Y}]_{\mathcal{T}_S,:} = \begin{bmatrix} 1 & 0 & 0 & \cdots & 0 \\ 0 & 1 & 0 & \cdots & 0 \\ \vdots & \vdots & \vdots & \ddots & \vdots \\ 0 & 0 & 0 & \cdots & 1 \end{bmatrix}_{|\mathcal{Y}|\times|\mathcal{Y}|} = \mathbf{I}_{|\mathcal{Y}|\times|\mathcal{Y}|}$$

*Using the Sherman-Morrison formula, we find its inverse:*

$$\left(\left[\left((1-\alpha)\bar{\mathbf{A}} + \alpha\mathbf{I}\right)\mathbf{X}\right]_{\mathcal{T}_S,:}\right)^{-1} = \frac{|\mathcal{Y}|-1}{p(a(h-1)|\mathcal{Y}| - h|\mathcal{Y}| + 1)|\mathcal{Y}|} \cdot$$
$$\begin{bmatrix} 1-|\mathcal{Y}| & 1 & 1 & \cdots & 1 \\ 1 & 1-|\mathcal{Y}| & 1 & \cdots & 1 \\ \vdots & \vdots & \vdots & \ddots & \vdots \\ 1 & 1 & 1 & \cdots & 1-|\mathcal{Y}| \end{bmatrix} + \frac{1}{|\mathcal{Y}|}$$

*Assuming $\mathbf{W}_* = \left(\left[\left((1-\alpha)\bar{\mathbf{A}} + \alpha\mathbf{I}\right)\mathbf{X}\right]_{\mathcal{T}_S,:}\right)^{-1}$, we obtain*

$$[\mathbf{z}]_{\mathcal{T}_S,:} = \left[\left((1-\alpha)\bar{\mathbf{A}} + \alpha\mathbf{I}\right)\mathbf{X}\right]_{\mathcal{T}_S,:}\mathbf{W}_* = [\mathbf{Y}]_{\mathcal{T}_S,:} = \mathbf{I}_{|\mathcal{Y}|\times|\mathcal{Y}|}.$$

*Since $\mathbf{W}_*$ satisfies $\mathcal{L}([\mathbf{z}]_{\mathcal{T}_\mathcal{V},:}, [\mathbf{Y}]_{\mathcal{T}_\mathcal{V},:}) = 0$, we know $\mathbf{W}_*$ is the optimal weight matrix that we can learn under $\mathcal{T}_\mathcal{V}$.*

*Now consider an arbitrary target node $v \in \mathcal{D}_\mathcal{V}$ with class label $y_v \in \mathcal{Y}$, and a unit structural perturbation leveraging gambit node $u \in \mathcal{V}$ with degree $d_a$ that affects the predictions $\mathbf{z}_v$ of node $v$ made by GNN $f_s^{(2)}$. Without loss of generality, we assume node $v$ has $y_v = 1$. Note that we will only discuss the case of direct structural perturbation to the 1-hop neighborhood $N(v)$ of target node*

$v$, as indirect perturbations do not affect the predictions $\mathbf{z}_v$ for node $v$ produced by a single GNN layer. Denote $\Delta\bar{\mathbf{A}} = \bar{\mathbf{A}}' - \bar{\mathbf{A}}$ as the change in the adjacency matrix $\bar{\mathbf{A}}$ before and after the attack. Similar to Proof 1, for simplicity of derivation, we assume that the perturbation does not change the row-stochastic normalization of $\bar{\mathbf{A}}$, and we use $\delta_1$ to denote addition ($\delta_1 = 1$) or removal ($\delta_1 = -1$) of a homophilous edge to node $v$, and use $\delta_2$ to denote addition or removal of a heterophilous edge to node $v$.

In the case of $\delta_1 = \pm 1$ and $\delta_2 = 0$, we have

$$\left[ \left( (1-\alpha)\Delta\bar{\mathbf{A}} + \alpha\mathbf{I} \right) \mathbf{X} \right]_{v,:} = \frac{(1-\alpha)\delta_1}{d_a|\mathcal{Y}|} \left[ \ ((|\mathcal{Y}|-1)p+1) \quad (1-p) \quad \cdots \quad (1-p) \ \right]$$

and the change in the CM-type attack loss $\mathcal{L}_{\text{atk}}^{\text{CM}}$ before and after the perturbation can be derived as

$$\Delta\mathcal{L}_{\text{atk}}^{\text{CM}} = \frac{((1-\alpha)|\mathcal{Y}| + \alpha - 1)\delta_1}{d_a(\alpha(h-1)-h)|\mathcal{Y}| + d_a}. \tag{19}$$

In the case of $\delta_1 = 0$ and $\delta_2 = \pm 1$, we have

$$\left[ \left( (1-\alpha)\Delta\bar{\mathbf{A}} + \alpha\mathbf{I} \right) \mathbf{X} \right]_{v,:} = \frac{(1-\alpha)\delta_2}{d_a|\mathcal{Y}|} \left[ \ (1-p) \quad (|\mathcal{Y}|-1)p+1 \quad \cdots \quad (|\mathcal{Y}|-1)p+1 \ \right]$$

and the change in the CM-type attack loss $\mathcal{L}_{\text{atk}}^{\text{CM}}$ before and after the perturbation can be derived as

$$\Delta\mathcal{L}_{\text{atk}}^{\text{CM}} = \frac{(\alpha - 1)(|\mathcal{Y}| - 1)\delta_2}{d_a(\alpha(h-1)-h)|\mathcal{Y}| + d_a}. \tag{20}$$

From Eq. (19) and Eq. (20), the change in the CM-type attack loss $\Delta\mathcal{L}_{\text{atk}}^{\text{CM}}$ for GNN layer $f(\mathbf{A}, \mathbf{X}; \alpha)$ considering both $\delta_1$ and $\delta_2$ can be written as

$$\Delta\mathcal{L}_{\text{atk}}^{\text{CM,f}} = \frac{((1-\alpha)|\mathcal{Y}| + \alpha - 1)\delta_1}{d_a(\alpha(h-1)-h)|\mathcal{Y}| + d_a} + \frac{(\alpha - 1)(|\mathcal{Y}| - 1)\delta_2}{d_a(\alpha(h-1)-h)|\mathcal{Y}| + d_a}. \tag{21}$$

**Layer** $f_s(\mathbf{A}, \mathbf{X}) = \bar{\mathbf{A}}_s\mathbf{X}\mathbf{W}$. This formulation is a special case of the previously discussed $f(\mathbf{A}, \mathbf{X}; \alpha)$ formulation when $\alpha = \frac{1}{1+d_a}$.

In the case of $\delta_1 = \pm 1$ and $\delta_2 = 0$, from Eq. (19), we have the change in the CM-type attack loss $\mathcal{L}_{\text{atk}}^{\text{CM}}$ before and after the perturbation as

$$\Delta\mathcal{L}_{\text{atk}}^{\text{CM}} = -\frac{(|\mathcal{Y}| - 1)\delta_1}{d_a(h|\mathcal{Y}| - 1) + |\mathcal{Y}| - 1}. \tag{22}$$

In the case of $\delta_1 = 0$ and $\delta_2 = \pm 1$, from Eq. (20), we have the change in the CM-type attack loss $\mathcal{L}_{\text{atk}}^{\text{CM}}$ before and after the perturbation as

$$\Delta\mathcal{L}_{\text{atk}}^{\text{CM}} = \frac{(|\mathcal{Y}| - 1)\delta_2}{d_a(h|\mathcal{Y}| - 1) + |\mathcal{Y}| - 1}. \tag{23}$$

From Eq. (22) and Eq. (23), the change in the CM-type attack loss $\Delta\mathcal{L}_{\text{atk}}^{\text{CM}}$ for GNN layer $f_s(\mathbf{A}, \mathbf{X})$ considering both $\delta_1$ and $\delta_2$ can be written as

$$\Delta\mathcal{L}_{\text{atk}}^{\text{CM,fs}} = -\frac{(|\mathcal{Y}| - 1)\delta_1}{d_a(h|\mathcal{Y}| - 1) + |\mathcal{Y}| - 1} + \frac{(|\mathcal{Y}| - 1)\delta_2}{d_a(h|\mathcal{Y}| - 1) + |\mathcal{Y}| - 1} \tag{24}$$

**Comparison of increase in attack loss** $\Delta\mathcal{L}_{\text{atk}}^{\text{CM}}$.

Solving the following system of inequalities for variable $\alpha$

$$\begin{cases} \Delta\mathcal{L}_{\text{atk}}^{\text{CM,fs}} > \Delta\mathcal{L}_{\text{atk}}^{\text{CM,f}} > 0 \\ \alpha, h \in [0, 1] \\ |\mathcal{Y}| \geq 2 \\ d_a, |\mathcal{Y}| \in \mathbb{Z}^+ \\ \delta_1, \delta_2 \in \{-1, 0, 1\} \end{cases} \tag{25}$$

*we get the valid range of $\alpha$ as*

$$
\begin{cases}
\frac{1}{d_a+1} < \alpha < 1, & \text{when } 0 \le h < \frac{1}{|\mathcal{Y}|} \text{ and } 0 < d_a < \frac{1-|\mathcal{Y}|}{h|\mathcal{Y}|-1} \text{ and } \delta_1 < \delta_2 \\
0 \le \alpha < \frac{1}{d_a+1}, & \text{when } 0 \le h < \frac{1}{|\mathcal{Y}|} \text{ and } d_a > \frac{1-|\mathcal{Y}|}{h|\mathcal{Y}|-1} \text{ and } \delta_1 > \delta_2 \\
\frac{1}{d_a+1} < \alpha < 1, & \text{when } \frac{1}{|\mathcal{Y}|} \le h \le 1 \text{ and } \delta_1 < \delta_2
\end{cases}
. \tag{26}
$$

*From the solution in Eq. (26), we observe that when $h > \frac{1}{|\mathcal{Y}|}$, a unit perturbation increasing $\mathcal{L}_{\text{atk}}$ as discussed in Theorem 1 (i.e. $\delta_1 = -1$ and $\delta_2 = 0$, or $\delta_1 = 0$ and $\delta_2 = 1$) will satisfy the condition $\delta_1 < \delta_2$, and thus lead to a strictly smaller increase $\Delta\mathcal{L}_{\text{atk}}^{\text{CM,f}}$ in the attack loss for layer $f(\mathbf{A}, \mathbf{X}; \alpha)$ than the increase $\Delta\mathcal{L}_{\text{atk}}^{\text{CM,fs}}$ for layer $f_s(\mathbf{A}, \mathbf{X})$ if $\alpha > \frac{1}{d_a+1}$.* ∎

# D    DETAILED EXPERIMENTAL SETUPS AND HYPERPARAMETERS

## D.1    INSTANTIATIONS OF DESIGN ON GNNS

We explicitly demonstrate how the heterophilous design outlined in Eq. (1) of §3.2 are instantiated in various GNN models used in our experiments. In particular, we highlight how these GNN architectures allow separate aggregations of the ego- and neighbor-embeddings.

- In **$H_2$GCN**, a final hidden representation is computed for each node $v \in \mathcal{V}$ through $\mathbf{r}_v^{(\text{final})} = \text{CONCAT}(\mathbf{r}_v^{(0)}, \mathbf{r}_v^{(1)}, ..., \mathbf{r}_v^{(K)})$, where $\mathbf{r}_v^{(0)}$ is the non-linear ego-embedding of node features and $\mathbf{r}_v^{(k)}$ are the intermediate representations aggregated in the $k$-th layer, where $k \in (1, ..., K)$. By interpreting the update rule's CONCAT as the ENC operation, AGGR1 as the skip connection to the ego-embedding of node features, and the concatenation of the intermediate representations as AGGR2, the ego- and neighbor-embeddings are separately aggregated and the identified heterophilous design is recovered.

- Similarly, **GraphSAGE** (with mean aggregator) utilizes a concatenation-based encoding scheme through their update of

$$\mathbf{r}_v^{(k)} = \sigma \left( \text{CONCAT} \left( \mathbf{r}_v^{(k-1)}, \text{ MEAN} \left( \{\mathbf{r}_u^{(k-1)}, \forall u \in N(i)\} \right) \right) \cdot \mathbf{W} \right)$$

where $\text{AGGR1}(\cdot) = \mathbf{r}_u^{(k-1)}$, AGGR2 is the mean function and $\text{ENC}(\mathbf{x}_1, \mathbf{x}_2) = \sigma(\text{CONCAT}(\mathbf{x}_1, \mathbf{x}_2) \cdot \mathbf{W})$.

- **GPR-GNN** embeds each node feature vector separately with a fully connected layer to compute $\mathbf{r}_v^{(0)}$ (or $\mathbf{H}_{v:}^{(0)}$ as in the original paper), similar to $H_2$GCN, and then updates each node's hidden representations through a weighted sum of all $k$-th hop layers around the ego-node, where $k \in (0, 1, ..., K)$. By interpreting the summation as the ENC operation as well as $\text{AGGR1}(\cdot) = \boldsymbol{\gamma}_0 \mathbf{H}^{(0)}$ and $\text{AGGR2}(\cdot) = \sum_{k=1}^{K} \gamma_k \tilde{\mathbf{A}}_{\text{sym}}^k \mathbf{H}^{(k-1)}$, where $\boldsymbol{\gamma}$ is a vector denoting the weights associated with each $k$-hop ego network, the aggregation of the ego- and neighbor-embeddings can be decoupled and thus GPR-GNN also satisfies the heterophilous design.

- **FAGCN** follows a similar update function to GPR-GNN with

$$\mathbf{h}_i^{(l)} = \varepsilon \mathbf{h}_i^{(0)} + \sum_{j \in N(i)} \frac{\alpha_{ij}^G}{\sqrt{d_i d_j}} \mathbf{h}_j^{(l-1)}$$

where $\mathbf{h}_i^{(0)}$, equivalent to $\mathbf{r}_i^{(0)}$ in Eq. (1), represents the non-linear ego-embedding and $\alpha_{ij}^G$ is a proportionality constant measuring the ratio of low and high frequency components. The relationship between FAGCN and the proposed heterophilous design can similarly be inferred by interpreting the sum as the ENC operation, $\text{AGGR1}(\cdot) = \varepsilon \mathbf{h}_i^{(0)}$ as a weighted skip connection to the ego-embedding of feature vectors, and the weighted sum of embeddings within the neighborhood $N(i)$ of node $i \in \mathcal{V}$ as $\text{AGGR2}(\cdot)$.

- **CPGNN** formulates the update function of belief vectors $\bar{\mathbf{B}}^{(k)}$ after the $k$-th propagation layer as $\bar{\mathbf{B}}^{(k)} = \bar{\mathbf{B}}^{(0)} + \mathbf{A}\bar{\mathbf{B}}^{(k-1)}\bar{\mathbf{H}}$, where $\bar{\mathbf{B}}^{(0)}$ consists of prior belief vectors for each node (which can be seen as the ego-embedding $\mathbf{r}_i^{(0)}$ in Eq. (1)), and $\bar{\mathbf{H}}$ is the linearized compatibility matrix. The heterophilous design can be recovered by letting $\text{AGGR1}(\cdot) = \bar{\mathbf{B}}^{(0)}$ as a skip connection to the prior belief, $\text{AGGR2}(\cdot) = \mathbf{A}\bar{\mathbf{B}}^{(k-1)}\bar{\mathbf{H}}$, and the ENC operation as the summation.

### D.2 MORE DETAILS ON THE EXPERIMENTAL SETUP

**Benchmark Implementations.** Our empirical framework is built on DeepRobust (Li et al., 2020b), Python Fire[1] and signac[2]. We incorporated the following implementations of GNN models in our framework. For GNNGuard and GCN-SVD, there are some implementation ambiguities, which we discuss in the next paragraph.

| | |
|---|---|
| **$H_2$GCN** (Zhu et al., 2020) | `https://github.com/GemsLab/H2GCN` |
| **GraphSAGE** (Hamilton et al., 2017) | Implemented on top of `https://github.com/GemsLab/H2GCN` |
| **CPGNN** (Zhu et al., 2021) | `https://github.com/GemsLab/CPGNN` |
| **GNNGuard** (Zhang & Zitnik, 2020) | `https://github.com/mims-harvard/GNNGuard` |
| **ProGNN** (Jin et al., 2020b) | `https://github.com/ChandlerBang/Pro-GNN` |
| **GCN-SVD** (Entezari et al., 2020) | `https://github.com/DSE-MSU/DeepRobust/blob/master/examples/graph/test_gcn_svd.py` |
| **GAT** (Veličković et al., 2018) | `https://github.com/DSE-MSU/DeepRobust/blob/master/deeprobust/graph/defense/gat.py` |
| **GCN** (Kipf & Welling, 2017) | `https://github.com/DSE-MSU/DeepRobust/blob/master/deeprobust/graph/defense/gcn.py` |

**Notes on the GNNGuard and GCN-SVD Implementations.** We note that there are ambiguities in the implementations of GNNGuard (Zhang & Zitnik, 2020) and GCN-SVD (Entezari et al., 2020), which can lead to different variants with different performance and robustness, as we show in Table 6.

Table 6: Performance comparison between variants of GNNGuard and GCN-SVD: mean accuracy ± stdev over multiple sets of experiments.

| | | Nettack | | | | Metattack | | | |
|---|---|---|---|---|---|---|---|---|---|
| | | Homophilous graphs | | Heterophilous graphs | | Homophilous graphs | | Heterophilous graphs | |
| | | **Cora** $h$=0.804 | **Citeseer** $h$=0.736 | **FB100** $h$=0.531 | **Snap** $h$=0.134 | **Cora** $h$=0.804 | **Citeseer** $h$=0.736 | **FB100** $h$=0.531 | **Snap** $h$=0.134 |
| **Clean Datasets** | **GNNGuard ( I )** | 75.56±5.15 | 70.00±6.24 | 68.89±2.08 | 31.67±0.00 | 79.58±0.97 | 71.68±1.10 | 65.31±1.48 | 26.37±0.70 |
| | **GNNGuard (II)** | 77.22±6.29 | 67.78±4.78 | 67.22±2.08 | 28.33±3.60 | 80.15±0.55 | 72.61±0.28 | 65.66±0.60 | 26.51±0.98 |
| | **GCN-SVD ( I )** $k=5$ | 66.67±8.16 | 63.89±5.50 | 51.67±6.24 | 29.44±0.79 | 69.43±0.99 | 68.31±0.34 | 52.95±0.13 | 27.66±0.05 |
| | **GCN-SVD (II)** $k=5$ | 52.78±5.50 | 35.00±1.36 | 50.56±4.37 | 25.00±5.93 | 55.05±1.77 | 41.47±0.72 | 52.40±0.18 | 25.84±0.07 |
| | **GCN-SVD ( I )** $k=10$ | 66.11±6.71 | 65.00±3.60 | 52.78±4.16 | 30.56±2.08 | 71.08±0.46 | 69.19±1.13 | 54.47±0.32 | 27.57±0.18 |
| | **GCN-SVD (II)** $k=10$ | 66.11±4.78 | 45.00±3.60 | 51.11±2.08 | 22.22±4.78 | 64.79±1.56 | 52.17±0.39 | 51.19±0.41 | 25.45±0.21 |
| | **GCN-SVD ( I )** $k=15$ | 72.78±6.98 | 63.89±7.74 | 57.78±2.83 | 28.89±2.08 | 72.74±0.29 | 66.51±1.53 | 57.67±0.36 | 27.61±0.55 |
| | **GCN-SVD (II)** $k=15$ | 69.44±2.08 | 46.67±6.24 | 52.78±5.15 | 21.67±1.36 | 65.61±0.19 | 60.55±0.73 | 53.24±0.45 | 26.63±0.25 |
| | **GCN-SVD ( I )** $k=50$ | 78.89±6.29 | 66.67±3.60 | 65.56±2.83 | 31.11±0.79 | 77.98±0.43 | 68.25±0.86 | 63.41±0.45 | 27.81±0.39 |
| | **GCN-SVD (II)** $k=50$ | 75.56±4.16 | 59.44±0.79 | 55.00±1.36 | 27.78±6.71 | 76.61±0.31 | 66.90±0.16 | 55.47±0.23 | 25.62±0.12 |
| **Poison Attacks** | **GNNGuard ( I )** | 57.22±2.08 | 60.00±3.60 | 0.56±0.79 | 11.11±0.79 | 73.68±0.99 | 67.89±0.92 | 60.82±0.45 | 23.98±0.71 |
| | **GNNGuard (II)** | 58.33±1.36 | 59.44±3.14 | 0.56±0.79 | 9.44±1.57 | 74.20±0.55 | 68.13±0.74 | 60.89±0.48 | 23.78±0.67 |
| | **GCN-SVD ( I )** $k=5$ | 64.44±9.06 | 60.00±4.71 | 41.67±6.24 | 27.78±6.29 | 64.65±2.57 | 66.35±1.48 | 53.14±0.43 | 25.64±0.47 |
| | **GCN-SVD (II)** $k=5$ | 44.44±2.83 | 33.33±2.72 | 41.67±2.36 | 25.00±6.80 | 42.19±5.33 | 40.17±1.57 | 51.87±0.38 | 24.82±0.43 |
| | **GCN-SVD ( I )** $k=10$ | 67.78±5.50 | 57.78±1.57 | 35.56±1.57 | 31.67±5.93 | 65.54±1.28 | 65.46±0.92 | 55.68±0.15 | 25.93±0.75 |
| | **GCN-SVD (II)** $k=10$ | 48.89±3.14 | 31.67±2.36 | 34.44±0.79 | 26.11±6.85 | 49.92±5.88 | 47.16±3.93 | 53.16±0.45 | 25.30±0.28 |
| | **GCN-SVD ( I )** $k=15$ | 64.44±3.93 | 52.78±4.78 | 23.89±6.29 | 29.44±3.93 | 65.46±2.33 | 61.04±1.04 | 58.06±0.05 | 25.83±0.69 |
| | **GCN-SVD (II)** $k=15$ | 51.11±3.42 | 33.89±3.93 | 30.56±2.83 | 26.11±6.14 | 50.30±3.80 | 47.87±1.31 | 54.20±0.36 | 25.25±0.91 |
| | **GCN-SVD ( I )** $k=50$ | 61.67±4.71 | 48.33±7.07 | 16.67±4.08 | 30.56±7.97 | 60.06±5.43 | 49.31±4.52 | 62.07±0.69 | 26.05±0.63 |
| | **GCN-SVD (II)** $k=50$ | 53.33±4.91 | 28.89±2.08 | 23.33±2.72 | 25.00±5.44 | 47.82±7.59 | 51.20±1.78 | 55.00±2.06 | 25.18±0.98 |
| **Evasion Attacks** | **GNNGuard ( I )** | - | - | - | - | - | - | - | - |
| | **GNNGuard (II)** | - | - | - | - | - | - | - | - |
| | **GCN-SVD ( I )** $k=5$ | 64.44±8.20 | 59.44±3.93 | 41.11±7.97 | 31.67±3.60 | 68.18±1.13 | 67.54±0.97 | 52.91±0.28 | 27.40±0.29 |
| | **GCN-SVD (II)** $k=5$ | 46.67±4.08 | 32.22±4.37 | 45.56±3.93 | 26.11±6.14 | 52.01±2.45 | 30.69±1.01 | 52.32±0.10 | 25.87±0.27 |
| | **GCN-SVD ( I )** $k=10$ | 65.56±7.49 | 57.22±3.42 | 36.11±1.57 | 31.11±0.79 | 68.36±1.33 | 67.85±0.72 | 54.51±0.68 | 27.30±0.30 |
| | **GCN-SVD (II)** $k=10$ | 57.22±6.14 | 37.78±3.93 | 36.67±3.60 | 30.56±8.85 | 58.70±3.00 | 45.62±2.52 | 52.58±0.20 | 24.60±0.26 |
| | **GCN-SVD ( I )** $k=15$ | 67.22±6.14 | 54.44±6.14 | 24.44±5.50 | 30.00±1.36 | 69.32±1.21 | 65.26±0.97 | 57.79±0.38 | 28.06±0.23 |
| | **GCN-SVD (II)** $k=15$ | 65.56±6.14 | 38.33±5.93 | 23.89±6.98 | 27.22±7.97 | 64.02±1.30 | 54.09±2.25 | 53.81±0.35 | 25.29±0.41 |
| | **GCN-SVD ( I )** $k=50$ | 65.00±6.24 | 50.56±6.43 | 18.33±2.36 | 25.56±1.57 | 75.30±0.62 | 64.49±1.58 | 63.53±0.26 | 27.74±0.61 |
| | **GCN-SVD (II)** $k=50$ | 60.00±6.24 | 47.78±4.37 | 25.00±4.71 | 30.56±9.56 | 73.21±1.68 | 59.34±3.42 | 56.95±0.33 | 25.80±0.67 |

For **GNNGuard**, the ambiguity comes from different interpretations of Eq. (4) in the original paper (Zhang & Zitnik, 2020): we consider the authors' original implementation as variant (I), and the model described in the original paper as variant (II), which we implement by building on the authors' implementation.

[1] `https://github.com/google/python-fire`

[2] `https://signac.io`

Table 6 shows that the differences in accuracy between the two variants are in most cases less than 2%, while in many cases variant (II) shows better accuracy compared to variant (I), especially in experiments against Metattack. Thus, we use variant (II) as the default implementation for the empirical evaluations in §5.

For **GCN-SVD**, the ambiguity comes from the order of applying the preprocessing and low-rank approximation for the adjacency matrix $\mathbf{A}$, which is not discussed in the original paper (Entezari et al., 2020).

- **Variant (I)**: Since the original authors' implementation is not publicly available, we consider the implementation provided in DeepRobust (Li et al., 2020b) as variant (I): it first calculates the rank-$k$ approximation $\tilde{\mathbf{A}}$ of $\mathbf{A}$, and then generates the preprocessed adjacency matrix $\hat{\mathbf{A}}_{\mathrm{s}} = \tilde{\mathbf{D}}_{\mathrm{s}}^{-1/2}(\tilde{\mathbf{A}} + \mathbf{I})\tilde{\mathbf{D}}_{\mathrm{s}}^{-1/2} = \tilde{\mathbf{D}}_{\mathrm{s}}^{-1/2}\tilde{\mathbf{A}}_{\mathrm{s}}\tilde{\mathbf{D}}_{\mathrm{s}}^{-1/2}$, which is then processed by a GCN (Kipf & Welling, 2017). However, as the identity matrix $\mathbf{I}$ is added into $\tilde{\mathbf{A}}$ after the low-rank approximation, the diagonal elements of the resulting $\hat{\mathbf{A}}_{\mathrm{s}}$ matrix (i.e., the weights for the self-loop edges in the graph) can become significantly larger than the off-diagonal elements, especially when the rank $k$ is low. As a result, this order of applying the preprocessing and low-rank approximation inadvertently adopts Design 1 which we identified; we have shown in Theorem 3 that even merely increasing the weights $\alpha$ for the ego-embedding in the linear combination `ENC` in Eq. (1) can lead to reduced attack loss $\mathcal{L}_{\mathrm{atk}}$ under structural perturbations.
- **Variant (II)**: In variant (II), we consider the opposite order where we first add the identity matrix $\mathbf{I}$ (self-loops) into the original adjacency matrix $\mathbf{A}$, then we perform the low-rank approximation, and finally we symmetrically normalize the low-rank matrix $\tilde{\mathbf{A}}_{\mathrm{s}}$ to generate the preprocessed $\hat{\mathbf{A}}_{\mathrm{s}}$ used by a GCN model. This order allows the diagonal elements to be more on par in magnitude with the off-diagonal elements. As an example, on Citeseer, when using variant (I) with rank $k = 5$, the average magnitude of the diagonal elements of the resulting $\hat{\mathbf{A}}_{\mathrm{s}}$ can be 22.3 times the average magnitude of the off-diagonal elements; when using variant (II) instead, the average magnitude of the diagonal elements is only 9.0 times that of the off-diagonal elements.

In Table 6, we report the performance of the two variants of GCN-SVD under the experimental settings considered in §5 with rank $k \in \{5, 10, 15, 50\}$: Variant (I), with our first design implicitly built-in, has in most cases significantly higher performance than variant (II), especially on homophilous datasets and when rank $k$ is low. These results further demonstrate the effectiveness of Design 1 that we identified. To enable a clear perspective of the performance and robustness improvement brought by Design 1, in our empirical analysis in §5, on top of the low-rank approximation vaccination we adopt variant (II) as the default implementation.

**Attack Implementations.** We incorporate the following implementations of attacks from `DeepRobust` (Li et al., 2020b) to our empirical framework.

| | |
|---|---|
| NETTACK (Zügner et al., 2018) | https://github.com/DSE-MSU/DeepRobust/blob/master/deeprobust/graph/targeted_attack/nettack.py |
| **Metattack** (Zügner & Günnemann, 2019a) | https://github.com/DSE-MSU/DeepRobust/blob/master/deeprobust/graph/global_attack/mettack.py |

**More Details on the Attack Setup.** For NETTACK, we randomly select 60 nodes from the graph as the target nodes for each set of perturbations, instead of the GCN-based target selection approach as in (Zügner et al., 2018): the approach in (Zügner et al., 2018) only selects nodes that are correctly classified by GCN (Kipf & Welling, 2017) on clean data, thus introducing unfair advantages towards GCN, especially on heterophilous datasets where GCN can exhibit significantly inferior accuracy to models like GraphSAGE (Zhu et al., 2020). For the experiments in §5.1, we use a budget of 1 perturbation per target node to match the setups of our theorems; for the benchmark study in §5.2, we use an attack budget equal to a node's degree and allow direct attacks on target nodes. For Metattack, we budget the attack as 20% of the number of edges in each dataset, and we use the Meta-Self variant as it shows the most destructiveness (Zügner & Günnemann, 2019a).

**More Details on Randomized Smoothing Setup.** Following Bojchevski et al. (2020), we similarly set the significance level $\alpha = 0.01$ (i.e., the certificates hold with probability $1 - \alpha = 0.99$), using $10^3$ samples to estimate the predictions of the smoothed classifier $f(\phi(\mathbf{s}))$ for input $\mathbf{s}$, and another $10^6$ samples to obtain multi-class certificates. For the randomization scheme $\phi$, we only consider structural perturbations where with probability $p_+$ an new edge is added, and with probability $p_-$ an existing edge is removed. We consider multiple sets of $(p_+, p_-)$ in our experiments for a finer-

grained evaluation: (1) $p_+ = 0.001, p_- = 0.4$, where both addition and deletion are allowed; (2) $p_+ = 0.001, p_- = 0$, where only addition is allowed; and (3) $p_+ = 0, p_- = 0.4$, where only deletion is allowed.

**Hardware Specifications.** We use a workstation with a 12-core AMD Ryzen 9 3900X CPU, 64GB RAM, and a Quadro P6000 GPU with 24 GB GPU Memory.

### D.3 COMBINING HETEROPHILOUS DESIGN WITH LOW-RANK APPROXIMATION

In this section, we provide more details on how we incorporate the low-rank approximation vaccination into the formulations of H$_2$GCN (Zhu et al., 2020) and GraphSAGE (Hamilton et al., 2017) in order to form the hybrid methods, H$_2$GCN-SVD and GraphSAGE-SVD.

**H$_2$GCN-SVD.** From (Zhu et al., 2020), each layer in the neighborhood aggregation stage of H$_2$GCN can be algebraically formulated as

$$\mathbf{R}^{(k)} = \text{CONCAT}\left(\hat{\mathbf{A}}_2 \mathbf{R}^{(k-1)}, \hat{\mathbf{A}} \mathbf{R}^{(k-1)}, \mathbf{R}^{(k-1)}\right), \tag{27}$$

where $\hat{\mathbf{A}} = \mathbf{D}^{-1/2} \mathbf{A} \mathbf{D}^{-1/2}$ is the symmetrically normalized adjacency matrix without self-loops; $\hat{\mathbf{A}}_2 = \mathbf{D}_2^{-1/2} \mathbf{A}_2 \mathbf{D}_2^{-1/2}$ is the symmetrically normalized 2-hop graph adjacency matrix $\mathbf{A}_2 \in \{0, 1\}^{|\mathcal{V}| \times |\mathcal{V}|}$, with $[\mathbf{A}_2]_{u,v} = 1$ if $v$ is in the 2-hop neighborhood $N_2(u)$ of node $u$; $\mathbf{R}^{(k)}$ are the node representations after the $k$-th layer, and CONCAT is the column-wise concatenation function. For H$_2$GCN-SVD, we replace $\hat{\mathbf{A}}$ and $\hat{\mathbf{A}}_2$ in Eq. (27) respectively with the low-rank approximations of $\tilde{\mathbf{A}}$ and $\tilde{\mathbf{A}}_2$, which are both postprocessed to be symmetrically normalized.

**GraphSAGE-SVD.** From (Hamilton et al., 2017), each layer in GraphSAGE can be algebraically formulated as

$$\mathbf{R}^{(k)} = \sigma\left(\text{CONCAT}\left(\bar{\mathbf{A}} \mathbf{R}^{(k-1)}, \mathbf{R}^{(k-1)}\right) \cdot \mathbf{W}^{(k-1)}\right), \tag{28}$$

where $\bar{\mathbf{A}}$ is the row-stochastic graph adjacency matrix without self-loops; $\mathbf{R}^{(k)}$ are the node representations after the $k$-th layer; CONCAT is the column-wise concatenation function; $\mathbf{W}^{(k)}$ is the learnable weight matrix for the $k$-th layer, and $\sigma$ is the non-linear activation function (ReLU). For GraphSAGE-SVD, we replace $\bar{\mathbf{A}}$ in Eq. (28) with the low-rank approximation of the adjacency matrix $\tilde{\mathbf{A}}$, postprocessed by row-stochastic normalization. Note that we do not enable the neighborhood sampling function for the GraphSAGE and GraphSAGE-SVD models tested in this work, as noted in Appendix D.4.

### D.4 HYPERPARAMETERS

- **H$_2$GCN-SVD**

Initialization Parameters:

– `adj_svd_rank`:
  best k chosen from $\{5, 50\}$ for each dataset

Training Parameters:

– `early_stopping`: Yes
– `train_iters`: 200
– `patience`: 100

- **GraphSAGE-SVD**

Initialization Parameters:

– `adj_nhood`: ['1']
– `network_setup`:
  `I-T1-G-V-C1-M64-R-T2-G-V-C2-MO-R`
– `adj_norm_type`: rw
– `adj_svd_rank`:
  best k chosen from $\{5, 50\}$ for each dataset

Training Parameters:

– `early_stopping`: Yes
– `train_iters`: 200
– `patience`: 100

- **H$_2$GCN**

  Initialization Parameters:
  (default parameters)

  Training Parameters:
  - `early_stopping`: Yes
  - `train_iters`: 200
  - `patience`: 100
  - `lr`: 0.01

- **GraphSAGE**

  Initialization Parameters:
  - `adj_nhood`: ['1']
  - `network_setup`:
    `I-T1-G-V-C1-M64-R-T2-G-V-C2-MO-R`
  - `adj_norm_type`: rw

  Training Parameters:
  - `early_stopping`: Yes
  - `train_iters`: 200
  - `patience`: 100
  - `lr`: 0.01

- **CPGNN**

  Initialization Parameters:
  - `network_setup`:
    `M64-R-MO-E-BP2`

  Training Parameters:
  - `early_stopping`: Yes
  - `train_iters`: 400
  - `patience`: 100
  - `lr`: 0.01

- **GPR-GNN**

  Initialization Parameters:
  - `nhid`: 64
  - `alpha`: 0.9, which is chosen from the best $\alpha \in \{0.1, 0.2, 0.5, 0.9\}$ on all datasets

  Training Parameters:
  - `train_iters`: 200
  - `lr`: 0.01

- **FAGCN**

  Initialization Parameters:
  - `nhid`: 64
  - `alpha`: 0.9, which is chosen from the best $\alpha \in \{0.1, 0.2, 0.5, 0.9\}$ on all datasets
  - `dropout`: 0.5

  Training Parameters:
  - `early_stopping`: Yes
  - `lr`: 0.01

- **GNNGuard**

  Initialization Parameters:
  - `nhid`: 64
  - `dropout`: 0.5
  - `base_model`: GCN for variant (I);
    GCN-fixed for variant (II) (default).

  Training Parameters:
  - `train_iters`: 81
  - `lr`: 0.01

- **ProGNN**

  Initialization Parameters:

  – `nhid`: 64
  – `dropout`: 0.5

  Training Parameters:

  – `epochs`: 400
  – `lr`: 0.01
  – `lr_adj`: 0.01
  – `weight_decay`: 5e-4
  – `alpha`: 5e-4
  – `beta`: 1.5
  – `gamma`: 1
  – `lambda_`: 0
  – `phi`: 0
  – `outer_steps`: 1
  – `innter_steps`: 2

- **GCN-SVD**

  Initialization Parameters:

  – `nhid`: 64
  – `k`: best k chosen from $\{5, 10, 15, 50\}$ for each dataset
  – `dropout`: 0.5
  – `svd_solver`:
    `eye-svd` (for variant (II) only)

  Training Parameters:

  – `train_iters`: 200
  – `weight_decay`: 5e-4
  – `lr`: 0.01

- **GCN**

  Initialization Parameters
  (in `class MultiLayerGCN`):

  – `nhid`: 64
  – `nlayer`: 2

  Training Parameters:

  – `train_iters`: 200
  – `lr`: 0.01
  – `weight_decay`: 5e-4

- **GAT**

  Initialization Parameters

  – `nhid`: 8
  – `heads`: 8
  – `dropout`: 0.5

  Training Parameters:

  – `early_stopping`: Yes
  – `train_iters`: 1000
  – `patience`: 100
  – `lr`: 0.01
  – `weight_decay`: 5e-4

- **MLP**

  Initialization Parameters:
  (in `class H2GCN`):

  – `network_setup`:
    `M64-R-D0.5-MO`

  Training Parameters:

  – `early_stopping`: Yes
  – `train_iters`: 200
  – `patience`: 100
  – `lr`: 0.01

### D.5 DATASETS

**Dataset and Unidentifiability.**

- **Heterophilous Datasets:** FB100 (Traud et al., 2012) is a set of 100 university friendship network snapshots from Facebook in 2005 (Lim et al., 2021), from which we use one network. Each node is labeled with the reported gender, and the features encode education and accommodation. Data is sent to the original authors (Lim et al., 2021) in an anonymized form. Though the dataset contains limited demographic (categorical) information volunteered by users on their individual Facebook pages, we manually inspect the dataset and confirm that the anonymized dataset is not recoverable and thus not identifiable. Also, no offensive content is found within the data.

  Snap Patents (Leskovec et al., 2005; Leskovec & Krevl, 2014) is a utility patent citation network. Node labels reflect the time the patent was granted, and the features are derived from the patent's metadata. The dataset is maintained by the National Bureau of Economic Research, and is freely available for download[3]. Neither personally identifiable information nor offensive content is identified when we manually inspect the dataset.

- **Homophilous Datasets:** Cora (McCallum, 2000) and Citeseer (Sen et al., 2008) datasets are scientific publication citation networks, whose labels categorize the research field, and features indicate the absence or presence of the corresponding word from the dictionary. No personally identifiable information or offensive content is identified when we manually inspect both datasets.

**Downsampling.** For better computational tractability, we sample a subset of the Snap Patents data using a snowball sampling approach (Goodman, 1961), where a random 20% of the neighbors for each traversed node are kept. We provide the pseudocode for the downsampling process in Algorithm 1.

---

**Algorithm 1:** Downsampling Algorithm For Snap Patents

**Input:** Graph to sample $G$
        Number of nodes to sample $N$
        Sampling ratio $p$
**Output:** Downsampled graph $G'$

1  initialization
    /* Initialize a queue $bfs_{quene}$ for Breadth First Search,
      and a list $nodes_{sampled}$ for storing sampled nodes           */
2  $bfs_{quene} \leftarrow$ QUEUE()
3  $nodes_{sampled} \leftarrow$ LIST()
    /* Start BFS with a random node from the largest connected component
      in G;
      RANDOM(array, n) returns n elements from an array with equal
      probability without replacement                   */
4  $node_{starting} \leftarrow$ RANDOM(LARGESTCONNECTEDCOMPONENT($G$), 1)
5  push $node_{starting}$ into $bfs_{quene}$
6  **while** LENGTH(*nodes_{sampled}*) ¡ $N$ **do**
7      node $\leftarrow bfs_{quene}$.pop()
8      neighbors $\leftarrow$ one hop neighbors of node
9      $neighbors_{drawn} \leftarrow$ RANDOM(neighbors, $p\times$ LENGTH(neighbors))
10     **for** *neighbor* $\in$ *neighbors_{drawn}* **do**
11        **if** *neighbor* $\notin$ *nodes_{sampled}* **then**
12           append neighbor to $nodes_{sampled}$
13           push neighbor into $bfs_{quene}$
14        **end**
15     **end**
16  **end**
17  $G' \leftarrow$ subgraph induced by $nodes_{sampled}$
18  **return** $G'$

---

[3] https://www.nber.org/research/data/us-patents

# E DETAILED EXPERIMENT RESULTS

## E.1 DETAILED RESULTS FOR EVALUATION ON EMPIRICAL ROBUSTNESS

Table 7: Detailed classification accuracy (and standard deviation) of each method for the target nodes attacked by NETTACK, calculated across different sets of perturbation.

| | Homophilous graphs | | Heterophilous graphs | |
|---|---|---|---|---|
| | **Cora** $h$=0.804 | **Citeseer** $h$=0.736 | **FB100** $h$=0.531 | **Snap** $h$=0.134 |
| | **Clean** | | | |
| **H$_2$GCN-SVD** | $74.44_{\pm3.42}$ | $70.00_{\pm2.72}$ | $61.67_{\pm2.36}$ | $30.56_{\pm2.08}$ |
| **GraphSAGE-SVD** | $77.22_{\pm4.78}$ | $70.00_{\pm1.36}$ | $60.00_{\pm4.08}$ | $27.22_{\pm5.50}$ |
| **H$_2$GCN** | $82.78_{\pm8.31}$ | $69.44_{\pm6.98}$ | $60.56_{\pm1.57}$ | $30.00_{\pm2.72}$ |
| **GraphSAGE** | $82.22_{\pm9.56}$ | $70.56_{\pm6.85}$ | $60.00_{\pm2.72}$ | $24.44_{\pm4.16}$ |
| **CPGNN** | $81.67_{\pm8.28}$ | $73.33_{\pm1.36}$ | $66.11_{\pm4.16}$ | $28.89_{\pm5.50}$ |
| **GPRGNN** | $82.22_{\pm7.49}$ | $67.78_{\pm2.08}$ | $56.67_{\pm4.91}$ | $27.78_{\pm3.42}$ |
| **FAGCN** | $83.33_{\pm8.16}$ | $70.56_{\pm5.15}$ | $58.33_{\pm5.93}$ | $29.44_{\pm0.79}$ |
| **GNNGuard** | $77.22_{\pm6.29}$ | $67.78_{\pm4.78}$ | $67.22_{\pm2.08}$ | $28.33_{\pm3.60}$ |
| **ProGNN** | $79.44_{\pm3.42}$ | $67.22_{\pm4.78}$ | $51.11_{\pm3.93}$ | $27.22_{\pm5.50}$ |
| **GCN-SVD** | $75.56_{\pm4.16}$ | $59.44_{\pm0.79}$ | $50.56_{\pm4.37}$ | $27.78_{\pm6.71}$ |
| **GAT** | $84.44_{\pm3.42}$ | $70.00_{\pm7.20}$ | $60.56_{\pm0.79}$ | $30.56_{\pm2.83}$ |
| **GCN** | $82.78_{\pm4.78}$ | $69.44_{\pm7.74}$ | $63.33_{\pm2.72}$ | $33.33_{\pm2.72}$ |
| **MLP** | $64.44_{\pm3.42}$ | $70.56_{\pm3.42}$ | $57.78_{\pm2.83}$ | $30.00_{\pm2.72}$ |
| | **Poison** (Pre-training) | | | |
| **H$_2$GCN-SVD** | $70.00_{\pm2.72}$ | $65.00_{\pm3.60}$ | $59.44_{\pm3.42}$ | $28.89_{\pm3.42}$ |
| **GraphSAGE-SVD** | $71.67_{\pm2.36}$ | $67.78_{\pm3.42}$ | $60.00_{\pm1.36}$ | $26.67_{\pm6.80}$ |
| **H$_2$GCN** | $38.89_{\pm5.50}$ | $27.22_{\pm1.57}$ | $27.78_{\pm3.42}$ | $12.78_{\pm2.83}$ |
| **GraphSAGE** | $36.67_{\pm2.72}$ | $31.67_{\pm10.89}$ | $33.89_{\pm3.42}$ | $16.67_{\pm7.07}$ |
| **CPGNN** | $47.22_{\pm6.14}$ | $40.56_{\pm9.65}$ | $49.44_{\pm10.30}$ | $21.67_{\pm2.72}$ |
| **GPRGNN** | $21.67_{\pm2.72}$ | $24.44_{\pm2.08}$ | $2.78_{\pm0.79}$ | $4.44_{\pm2.08}$ |
| **FAGCN** | $26.11_{\pm6.14}$ | $25.56_{\pm6.43}$ | $6.11_{\pm2.83}$ | $8.33_{\pm3.60}$ |
| **GNNGuard** | $58.33_{\pm1.36}$ | $59.44_{\pm3.14}$ | $0.56_{\pm0.79}$ | $9.44_{\pm1.57}$ |
| **ProGNN** | $48.89_{\pm7.97}$ | $32.78_{\pm7.49}$ | $33.89_{\pm4.78}$ | $17.78_{\pm9.26}$ |
| **GCN-SVD** | $53.33_{\pm4.91}$ | $28.89_{\pm2.08}$ | $41.67_{\pm2.36}$ | $25.00_{\pm5.44}$ |
| **GAT** | $13.89_{\pm0.79}$ | $8.89_{\pm3.42}$ | $0.56_{\pm0.79}$ | $3.89_{\pm4.37}$ |
| **GCN** | $1.67_{\pm0.00}$ | $4.44_{\pm2.83}$ | $0.56_{\pm0.79}$ | $2.22_{\pm2.08}$ |
| **MLP** | $64.44_{\pm3.42}$ | $70.56_{\pm3.42}$ | $57.78_{\pm2.83}$ | $30.00_{\pm2.72}$ |
| | **Evasion** (Post-training) | | | |
| **H$_2$GCN-SVD** | $70.56_{\pm3.42}$ | $66.11_{\pm4.78}$ | $60.00_{\pm2.72}$ | $28.89_{\pm3.14}$ |
| **GraphSAGE-SVD** | $70.56_{\pm2.08}$ | $68.33_{\pm3.60}$ | $59.44_{\pm2.83}$ | $26.11_{\pm6.85}$ |
| **H$_2$GCN** | $45.56_{\pm3.42}$ | $33.89_{\pm1.57}$ | $32.78_{\pm0.79}$ | $12.78_{\pm2.83}$ |
| **GraphSAGE** | $44.44_{\pm3.14}$ | $35.00_{\pm8.92}$ | $42.22_{\pm2.83}$ | $15.56_{\pm6.71}$ |
| **CPGNN** | $52.22_{\pm6.98}$ | $46.67_{\pm6.80}$ | $15.56_{\pm2.83}$ | $22.78_{\pm4.78}$ |
| **GPRGNN** | $29.44_{\pm3.14}$ | $32.22_{\pm0.79}$ | $9.44_{\pm2.83}$ | $3.33_{\pm1.36}$ |
| **FAGCN** | $38.89_{\pm6.71}$ | $37.78_{\pm4.78}$ | $12.78_{\pm2.83}$ | $10.00_{\pm2.36}$ |
| **GNNGuard** | - | - | - | - |
| **ProGNN** | - | - | - | - |
| **GCN-SVD** | $60.00_{\pm6.24}$ | $47.78_{\pm4.37}$ | $45.56_{\pm3.93}$ | $30.56_{\pm9.56}$ |
| **GAT** | $12.22_{\pm4.16}$ | $23.33_{\pm5.44}$ | $1.67_{\pm1.36}$ | $2.22_{\pm3.14}$ |
| **GCN** | $5.56_{\pm2.08}$ | $8.89_{\pm4.16}$ | $0.56_{\pm0.79}$ | $0.56_{\pm0.79}$ |
| **MLP** | $64.44_{\pm3.42}$ | $70.56_{\pm3.42}$ | $57.78_{\pm2.83}$ | $30.00_{\pm2.72}$ |

Table 8: Detailed classification accuracy (and standard deviation) for the unlabeled nodes of each method attacked by Metattack with budget as 20% of the total number of edges of each graph, calculated across different sets of perturbation.

| | Homophilous graphs | | Heterophilous graphs | |
|---|---|---|---|---|
| | **Cora** $h$=0.804 | **Citeseer** $h$=0.736 | **FB100** $h$=0.531 | **Snap** $h$=0.134 |
| **Clean** | | | | |
| **H$_2$GCN-SVD** | $76.89_{\pm0.37}$ | $73.42_{\pm1.03}$ | $56.81_{\pm0.77}$ | $27.63_{\pm0.26}$ |
| **GraphSAGE-SVD** | $77.52_{\pm0.29}$ | $72.16_{\pm0.17}$ | $57.38_{\pm0.86}$ | $26.72_{\pm0.70}$ |
| **H$_2$GCN** | $83.94_{\pm0.97}$ | $75.34_{\pm0.90}$ | $56.95_{\pm0.13}$ | $27.49_{\pm0.05}$ |
| **GraphSAGE** | $82.21_{\pm0.63}$ | $74.64_{\pm0.93}$ | $56.60_{\pm1.40}$ | $27.18_{\pm0.84}$ |
| **CPGNN** | $80.67_{\pm0.51}$ | $74.92_{\pm0.62}$ | $60.17_{\pm7.09}$ | $27.13_{\pm0.63}$ |
| **GPRGNN** | $81.84_{\pm1.75}$ | $70.71_{\pm0.46}$ | $62.40_{\pm0.83}$ | $26.08_{\pm0.31}$ |
| **FAGCN** | $81.59_{\pm0.82}$ | $73.99_{\pm0.63}$ | $59.64_{\pm1.38}$ | $27.15_{\pm0.23}$ |
| **GNNGuard** | $80.15_{\pm0.55}$ | $72.61_{\pm0.28}$ | $65.66_{\pm0.60}$ | $26.51_{\pm0.98}$ |
| **ProGNN** | $81.32_{\pm0.43}$ | $71.82_{\pm1.12}$ | $49.84_{\pm0.03}$ | $27.49_{\pm0.66}$ |
| **GCN-SVD** | $76.61_{\pm0.31}$ | $66.90_{\pm0.16}$ | $55.47_{\pm0.23}$ | $26.63_{\pm0.25}$ |
| **GAT** | $83.72_{\pm0.24}$ | $73.40_{\pm1.00}$ | $61.69_{\pm0.92}$ | $27.30_{\pm0.03}$ |
| **GCN** | $84.32_{\pm0.32}$ | $74.27_{\pm0.15}$ | $64.86_{\pm0.79}$ | $27.30_{\pm0.43}$ |
| **MLP** | $64.55_{\pm1.58}$ | $67.67_{\pm0.11}$ | $56.56_{\pm0.58}$ | $26.25_{\pm1.05}$ |
| **Poison** (Pre-training) | | | | |
| **H$_2$GCN-SVD** | $67.87_{\pm0.47}$ | $70.42_{\pm0.46}$ | $56.72_{\pm0.08}$ | $25.60_{\pm0.14}$ |
| **GraphSAGE-SVD** | $68.86_{\pm1.32}$ | $69.10_{\pm0.52}$ | $55.76_{\pm0.33}$ | $26.58_{\pm0.30}$ |
| **H$_2$GCN** | $57.75_{\pm6.61}$ | $54.34_{\pm0.82}$ | $54.84_{\pm0.76}$ | $25.34_{\pm0.59}$ |
| **GraphSAGE** | $54.68_{\pm2.56}$ | $59.74_{\pm1.74}$ | $54.72_{\pm0.83}$ | $24.14_{\pm0.76}$ |
| **CPGNN** | $74.55_{\pm1.23}$ | $68.07_{\pm1.93}$ | $61.58_{\pm1.50}$ | $26.76_{\pm0.41}$ |
| **GPRGNN** | $48.29_{\pm5.23}$ | $35.25_{\pm2.77}$ | $59.94_{\pm0.60}$ | $21.06_{\pm1.29}$ |
| **FAGCN** | $60.11_{\pm4.82}$ | $53.18_{\pm6.00}$ | $55.97_{\pm1.81}$ | $24.04_{\pm0.62}$ |
| **GNNGuard** | $74.20_{\pm0.55}$ | $68.13_{\pm0.74}$ | $60.89_{\pm0.48}$ | $23.78_{\pm0.67}$ |
| **ProGNN** | $45.10_{\pm6.20}$ | $46.58_{\pm1.02}$ | $53.40_{\pm1.19}$ | $24.80_{\pm1.09}$ |
| **GCN-SVD** | $47.82_{\pm7.59}$ | $51.20_{\pm1.78}$ | $55.00_{\pm2.06}$ | $25.25_{\pm0.91}$ |
| **GAT** | $41.70_{\pm3.60}$ | $48.40_{\pm2.17}$ | $50.37_{\pm0.66}$ | $25.00_{\pm0.73}$ |
| **GCN** | $37.46_{\pm3.35}$ | $45.81_{\pm2.99}$ | $51.82_{\pm1.41}$ | $25.03_{\pm0.68}$ |
| **MLP** | $64.55_{\pm1.58}$ | $67.67_{\pm0.11}$ | $56.56_{\pm0.58}$ | $26.25_{\pm1.05}$ |
| **Evasion** (Post-training) | | | | |
| **H$_2$GCN-SVD** | $74.01_{\pm0.35}$ | $71.54_{\pm1.92}$ | $56.58_{\pm0.63}$ | $27.26_{\pm0.17}$ |
| **GraphSAGE-SVD** | $74.31_{\pm0.40}$ | $70.22_{\pm0.90}$ | $57.38_{\pm0.88}$ | $26.77_{\pm1.01}$ |
| **H$_2$GCN** | $82.86_{\pm1.01}$ | $73.20_{\pm2.04}$ | $57.05_{\pm0.20}$ | $27.10_{\pm0.06}$ |
| **GraphSAGE** | $80.57_{\pm0.55}$ | $72.89_{\pm1.72}$ | $56.91_{\pm1.61}$ | $27.16_{\pm0.78}$ |
| **CPGNN** | $79.06_{\pm1.18}$ | $73.44_{\pm0.98}$ | $60.19_{\pm7.20}$ | $27.02_{\pm0.75}$ |
| **GPRGNN** | $80.80_{\pm1.67}$ | $69.77_{\pm0.42}$ | $61.91_{\pm0.74}$ | $26.16_{\pm0.25}$ |
| **FAGCN** | $80.70_{\pm0.99}$ | $73.14_{\pm1.02}$ | $59.39_{\pm1.36}$ | $27.25_{\pm0.30}$ |
| **GNNGuard** | - | - | - | - |
| **ProGNN** | - | - | - | - |
| **GCN-SVD** | $73.21_{\pm1.68}$ | $59.34_{\pm3.42}$ | $56.95_{\pm0.33}$ | $25.29_{\pm0.41}$ |
| **GAT** | $81.96_{\pm0.31}$ | $70.70_{\pm0.69}$ | $61.44_{\pm0.94}$ | $27.45_{\pm0.13}$ |
| **GCN** | $83.03_{\pm0.69}$ | $73.06_{\pm0.58}$ | $64.86_{\pm0.47}$ | $27.15_{\pm0.39}$ |
| **MLP** | $64.55_{\pm1.58}$ | $67.67_{\pm0.11}$ | $56.56_{\pm0.58}$ | $26.25_{\pm1.05}$ |

### E.2 DETAILED RESULTS FOR EVALUATION ON CERTIFIABLE ROBUSTNESS

Table 9: Accumulated certifications (AC), average certifiable radii ($\bar{r}_a$ and $\bar{r}_d$) and accuracy of GNNs with randomized smoothing enabled (i.e., $f(\phi(\mathbf{s}))$) on all nodes of the clean datasets, with a ramdomization scheme $\phi$ allowing addition only (i.e., $p_+ = 0.001, p_- = 0$) or deletion only (i.e., $p_+ = 0, p_- = 0.4$). For each statistic, we report the mean and stdev across 3 runs. Best results are highlighted in blue per dataset, and in gray per model group. For results with randomization scheme allowing both addition and deletion, see Table 4.

| | Hete. | | Addition Only | | | | Deletion Only | | | |
|---|---|---|---|---|---|---|---|---|---|---|
| | | | AC | $\bar{r}_a$ | $\bar{r}_d$ | Acc. % | AC | $\bar{r}_a$ | $\bar{r}_d$ | Acc. % |
| $H_2$GCN | ✓ | | $0.42_{\pm0.01}$ | $0.52_{\pm0.03}$ | - | $80.85_{\pm1.98}$ | $5.41_{\pm0.23}$ | - | $6.44_{\pm0.20}$ | $84.04_{\pm1.55}$ |
| GraphSAGE | ✓ | | $0.28_{\pm0.03}$ | $0.34_{\pm0.03}$ | - | $81.05_{\pm1.34}$ | $5.05_{\pm0.12}$ | - | $6.12_{\pm0.06}$ | $82.49_{\pm1.06}$ |
| CPGNN | ✓ | Cora | $0.17_{\pm0.02}$ | $0.21_{\pm0.03}$ | - | $78.34_{\pm1.26}$ | $4.92_{\pm0.33}$ | - | $6.17_{\pm0.42}$ | $79.69_{\pm0.81}$ |
| GPR-GNN | ✓ | | $0.43_{\pm0.03}$ | $0.55_{\pm0.03}$ | - | $76.96_{\pm2.18}$ | $5.37_{\pm0.14}$ | - | $6.58_{\pm0.05}$ | $81.56_{\pm1.59}$ |
| FAGCN | ✓ | | $0.43_{\pm0.01}$ | $0.54_{\pm0.01}$ | - | $79.04_{\pm0.68}$ | $5.74_{\pm0.06}$ | - | $7.03_{\pm0.05}$ | $81.56_{\pm0.80}$ |
| GAT | | | $0.19_{\pm0.04}$ | $0.23_{\pm0.04}$ | - | $81.62_{\pm2.01}$ | $5.56_{\pm0.12}$ | - | $6.58_{\pm0.06}$ | $84.46_{\pm1.08}$ |
| GCN | | | $0.19_{\pm0.02}$ | $0.24_{\pm0.02}$ | - | $81.52_{\pm3.10}$ | $5.71_{\pm0.07}$ | - | $6.76_{\pm0.03}$ | $84.49_{\pm0.84}$ |
| $H_2$GCN | ✓ | | $0.29_{\pm0.05}$ | $0.40_{\pm0.06}$ | - | $72.93_{\pm2.30}$ | $5.55_{\pm0.10}$ | - | $7.29_{\pm0.18}$ | $76.17_{\pm0.48}$ |
| GraphSAGE | ✓ | | $0.33_{\pm0.01}$ | $0.45_{\pm0.01}$ | - | $74.66_{\pm1.19}$ | $5.43_{\pm0.07}$ | - | $7.15_{\pm0.12}$ | $75.97_{\pm0.47}$ |
| CPGNN | ✓ | Citeseer | $0.15_{\pm0.02}$ | $0.20_{\pm0.02}$ | - | $74.62_{\pm0.30}$ | $5.59_{\pm0.22}$ | - | $7.54_{\pm0.22}$ | $74.05_{\pm0.84}$ |
| GPR-GNN | ✓ | | $0.40_{\pm0.01}$ | $0.59_{\pm0.02}$ | - | $67.52_{\pm0.49}$ | $5.05_{\pm0.05}$ | - | $7.21_{\pm0.06}$ | $70.10_{\pm0.15}$ |
| FAGCN | ✓ | | $0.38_{\pm0.02}$ | $0.53_{\pm0.02}$ | - | $72.41_{\pm1.03}$ | $5.46_{\pm0.09}$ | - | $7.42_{\pm0.12}$ | $73.56_{\pm0.18}$ |
| GAT | | | $0.09_{\pm0.01}$ | $0.13_{\pm0.02}$ | - | $74.07_{\pm0.44}$ | $5.39_{\pm0.16}$ | - | $7.25_{\pm0.12}$ | $74.33_{\pm1.43}$ |
| GCN | | | $0.18_{\pm0.01}$ | $0.25_{\pm0.00}$ | - | $74.07_{\pm1.65}$ | $5.55_{\pm0.10}$ | - | $7.42_{\pm0.16}$ | $74.76_{\pm0.72}$ |
| $H_2$GCN | ✓ | | $0.54_{\pm0.00}$ | $0.94_{\pm0.00}$ | - | $57.11_{\pm0.10}$ | $4.75_{\pm0.03}$ | - | $8.32_{\pm0.03}$ | $57.15_{\pm0.23}$ |
| GraphSAGE | ✓ | | $0.52_{\pm0.01}$ | $0.92_{\pm0.01}$ | - | $56.70_{\pm1.41}$ | $4.28_{\pm0.05}$ | - | $7.56_{\pm0.10}$ | $56.58_{\pm1.32}$ |
| CPGNN | ✓ | FB100 | $0.54_{\pm0.04}$ | $0.90_{\pm0.04}$ | - | $60.39_{\pm7.26}$ | $4.22_{\pm0.05}$ | - | $6.66_{\pm0.11}$ | $63.30_{\pm0.29}$ |
| GPR-GNN | ✓ | | $0.46_{\pm0.01}$ | $0.73_{\pm0.02}$ | - | $62.26_{\pm0.26}$ | $4.04_{\pm0.08}$ | - | $6.51_{\pm0.10}$ | $62.05_{\pm0.31}$ |
| FAGCN | ✓ | | $0.55_{\pm0.00}$ | $0.90_{\pm0.01}$ | - | $60.60_{\pm0.36}$ | $4.58_{\pm0.10}$ | - | $7.71_{\pm0.03}$ | $59.45_{\pm1.26}$ |
| GAT | | | $0.46_{\pm0.03}$ | $0.74_{\pm0.04}$ | - | $61.97_{\pm1.41}$ | $3.33_{\pm0.09}$ | - | $5.37_{\pm0.11}$ | $62.01_{\pm1.01}$ |
| GCN | | | $0.42_{\pm0.01}$ | $0.65_{\pm0.01}$ | - | $64.70_{\pm0.55}$ | $3.28_{\pm0.04}$ | - | $5.01_{\pm0.11}$ | $65.50_{\pm0.56}$ |
| $H_2$GCN | ✓ | | $0.11_{\pm0.01}$ | $0.42_{\pm0.05}$ | - | $26.74_{\pm0.18}$ | $1.41_{\pm0.04}$ | - | $5.16_{\pm0.19}$ | $27.28_{\pm0.21}$ |
| GraphSAGE | ✓ | | $0.06_{\pm0.02}$ | $0.24_{\pm0.08}$ | - | $27.00_{\pm0.63}$ | $1.10_{\pm0.11}$ | - | $4.04_{\pm0.36}$ | $27.21_{\pm0.99}$ |
| CPGNN | ✓ | Snap | $0.12_{\pm0.02}$ | $0.43_{\pm0.08}$ | - | $27.00_{\pm0.41}$ | $1.69_{\pm0.06}$ | - | $6.39_{\pm0.13}$ | $26.45_{\pm0.50}$ |
| GPR-GNN | ✓ | | $0.03_{\pm0.01}$ | $0.11_{\pm0.02}$ | - | $26.14_{\pm0.73}$ | $1.10_{\pm0.04}$ | - | $4.19_{\pm0.17}$ | $26.24_{\pm0.43}$ |
| FAGCN | ✓ | | $0.10_{\pm0.01}$ | $0.36_{\pm0.03}$ | - | $27.13_{\pm0.16}$ | $1.77_{\pm0.05}$ | - | $6.48_{\pm0.20}$ | $27.25_{\pm0.18}$ |
| GAT | | | $0.02_{\pm0.00}$ | $0.08_{\pm0.02}$ | - | $27.00_{\pm0.59}$ | $1.10_{\pm0.03}$ | - | $4.06_{\pm0.11}$ | $27.18_{\pm0.04}$ |
| GCN | | | $0.02_{\pm0.00}$ | $0.07_{\pm0.01}$ | - | $27.45_{\pm0.82}$ | $1.38_{\pm0.05}$ | - | $5.01_{\pm0.16}$ | $27.57_{\pm0.27}$ |

### E.3 COMPARISON BETWEEN CERTIFIABLE AND EMPIRICAL ROBUSTNESS

While the evaluations on both empirical and certifiable robustness have shown that methods featuring the design have shown largely improved robustness compared to the best-performing unvaccinated method, we find that the robustness rankings for methods under certifiable and empirical robustness are different; previous results from Geisler et al. (2020) have also shown discrepancy in certifiable and empirical robustness rankings.

We think this discrepancy may be attributed to multiple factors. Firstly, the radius on which certificates can be issued with randomized smoothing may not cover the radius of perturbations we allowed for NETTACK and Metattack in §5.2, which are more than tens of edges for Metattack and high-degree nodes in NETTACK (where we use an attack budget equal to the degree of the target node). Furthermore, even for low-degree nodes, attacks are much more inclined to introduce new edges instead of removing existing ones, as we have shown in §5.1, which can make it much harder to obtain certificates (Bojchevski et al., 2020). In our case, the methods we evaluated generally have $\bar{r}_a < 1$, meaning that for most nodes there are no certificates to cover even the addition of a single edge. Thus, methods which display higher certifiable robustness under smaller perturbations may not keep their robustness under larger perturbations by empirical attacks. Moreover, while we are evaluating on randomized smoothed models $f(\phi(\mathbf{s}))$ to measure certifiable robustness, in empirical

robustness we are evaluating on the robustness of the base models $f(s)$, which may differ from the robustness of the randomized smoothed models. Lastly, it is worth noting that the lack of certification for a model within certain radius does not imply a vulnerability to all adversarial attacks within that radius; the model may still be robust against many attacks within that radius. Similarly, it is also possible for existing certification approaches to miss some certifiable cases. Taking all these factors into account, we believe that while evaluations on certifiable robustness provide complementary perspectives to evaluations empirical robustness, at the current stage it cannot replace the evaluations on empirical robustness, and the relation between certifiable and empirical robustness remains as a question for future works.

### E.4 COMPLEXITY AND RUNTIME OF HETEROPHILOUS VACCINATION

Another benefit of adopting heterophily-inspired design for boosting robustness of GNNs is their smaller computational overhead compared to existing vaccination mechanisms, especially vaccinations based on low-rank approximation. As our identified design can be applied as simple architectural changes on top of an existing GNN, they usually maintain the same order of computational complexity as the base model. For example, adding the heterophilous design to GCN (Kipf & Welling, 2017) results in an architecture similar to GraphSAGE (Hamilton et al., 2017); both have the same order of computational complexity as $O(|\mathcal{V}| + |\mathcal{E}|)$ by leveraging the sparse connectivity of most real-world graphs. Low-rank approximation-based vaccination, on the other hand, approximates the adjacency matrix of a graph by an SVD, resulting in an adjusted low-rank adjacency matrix $\tilde{\mathbf{A}}$ based on which the GNN runs. However, not only is computing an SVD potentially costly ($O(|\mathcal{V}|^3)$ in general), but in most cases it also results in a dense $\tilde{\mathbf{A}}$ (in contrast to the sparse original adjacency matrix), thus increasing the complexity of each iteration of the GNN.

Table 10: Runtime (in seconds) of 200 training iterations on Cora. See App. D for the implementation used for each method.

| GCN | GAT | GNNGuard | ProGNN | GCN-SVD | $H_2$GCN |
|---|---|---|---|---|---|
| 2.17 | 2.98 | 39.63 | 220.30 | 134.81 | 16.54 |

| GraphSAGE | FAGCN | GRP-GNN | CPGNN | $H_2$GCN-SVD | GraphSAGE-SVD |
|---|---|---|---|---|---|
| 17.24 | 1.91 | 2.66 | 24.08 | 62.33 | 55.45 |

Table 10 shows the runtime of 200 training iterations of each model. We observe that models with the heterophilous design have the smallest runtime among all vaccinated models. Even for methods based on the same implementation, $H_2$GCN and GraphSAGE are still 3-4 times faster than the corresponding $H_2$GCN-SVD and GraphSAGE-SVD methods. For fair runtime measurements, we measure the runtime of each model on an Amazon EC2 instance with instance type as `p3.2xlarge`, which features an 8-core CPU, 61 GB Memory, and a Tesla V100 GPU with 16 GB GPU Memory.

### E.5 ADDITIONAL RESULTS ON STRUCTURAL ATTACKS ON HETEROPHILOUS GRAPHS

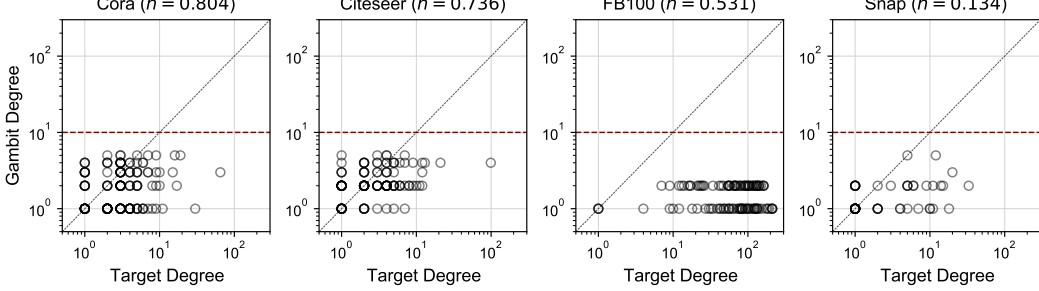

Figure 2: Scatter plots of the degrees of the target nodes (x-axis) and gambit nodes (y-axis) involved in the targeted attacks studied in §5.1. Attacks tend to leverage gambit nodes with low degrees, especially on heterophious graphs, which makes attacks increasing heterophily effective following the conclusions of Thm. 2.

