# OpenReview forum: "On the Relationship between Heterophily and Robustness of Graph Neural Networks"
_ICLR.cc/2022/Conference — ICLR 2022 Submitted_

### Official Review · Reviewer_kkG8 · 2021-10-31

**Correctness:** 4
**Technical Novelty And Significance:** 2
**Empirical Novelty And Significance:** 2
**Recommendation:** 5
**Confidence:** 4

**Main Review:**

Pros:
- The writing of this paper is very clean and easy to read.
- The paper focuses on a quite interesting problem, i.e., the connection between the adversarial attack on the graph and the change of heterophily.
Cons:
- The major concern is the novelty and significance of this paper:
    1. The main observation comes from previous work (Wu et al. 2019; Jin et al. 2020a) as mentioned in the paper. And the main contribution from the theoretical view, as the authors claimed, is the first formalization of the connection. However, the theory is rather weak as both the graph structure and the used model are simplified, i.e., regular graph with very simple node features plus linear GNN.
    2. The design principle mentioned in the paper has already been proposed by previous works, e.g., GraphSAGE, and the connection between this principle and the aforementioned theory results is vague.



**Summary Of The Paper:**

This paper investigates the change of the degree of homophily under adversarial perturbation of graphs from both theoretical and empirical points of view. In theory, this paper proves under simplified settings, for homophily graphs, the attack will increase the level of heterophily, and for heterophily graphs, this change will depend on the degree. In practice, this paper advocates a design principle of GNNs that separate aggregators for ego- and neighbor-embeddings.

**Summary Of The Review:**

This paper aims to investigate an interesting question about the relationship between robustness and heterophily of graphs, however, the novelty and significance of both theory and empirical results are not clear.

---

> ### Author Response · Authors · 2021-11-23
> **Reply to Reviewer kkG8 [1/2]**
>
> We appreciate your review on our work, and we would like to address your specific comments as follows:
>
> ## Novelty and significance of the paper
>
> As we mention in the meta-response, rather than proposing a well-rounded new method, the main focus of our work is on ***understanding and formalizing the relation* between heterophily and adversarial structural attacks, and *demonstrating the implication* of this insight on improving the robustness of GNNs against adversarial perturbations.**
>
> - Firstly, we *formally* show that adversarial structural attacks on homophilous graphs mostly lead to increase of the heterophily level. **Though former works have utilized this relation as heuristic in designing GNNs with improved robustness, they do not formally propose this research question, nor provide any formal theoretical analysis.**
>     - Though derived under specific assumptions, our theoretical analysis (Theorem 1) helps reveal *why* effective attacks on commonly seen homophilous graphs need to exploit increased level of heterophily.
>     - Our empirical observations (Table 2 in Section 5.1) confirm our takeaway on more generalized settings for real-world datasets.
> - Moreover, to our knowledge, **we are the first to show on *heterophilous* graphs how adversarial attacks change the level of homophily with formal theoretical and empirical analysis**. With our updated analysis, we reveal that in heterophilous graphs, the effectiveness of heterophilous attacks is dependent on low-degree nodes, which is supported by our empirical analysis on real-world datasets.
> - Finally, we demonstrate how the implication of the relation between heterophily and adversarial attacks can help defend against the attacks by showing the effectiveness of the heterophily-inspired designs in improving the robustness of GNNs. While some of these designs are used in existing methods, **their effectiveness in improving the robustness of GNNs with formal theoretical justifications, and the extensive evaluation on both *empirical* and *certifiable* *robustness* is novel, which has been largely unknown before this work*.*** Though we do not propose a concrete new defense method, we believe our observations can inspire future works in leveraging the relations between heterophily and adversarial attacks.
>
> ## "The theory is rather weak as both the graph structure and the used model are simplified."
>
> First, **we believe that the assumption of no non-linearity in GNN models is an acceptable simplification as it is commonly adopted in theoretical analysis of many prior works**. For example, Nettack (Zügner et al., 2018) and Metattack (Zügner & Günnemann, 2019a), two state-of-the-art attack methods which we studied in this work, both uses linearized GCN as the surrogate model when optimizing the adversarial perturbations; Wu et al. (2019b) has also shown that SGC, a simplified version of GCN with non-linearity removed, is able to achieve very similar performance to GCN in many cases.
>
> Moreover, as we mentioned in the meta-response, **we have relaxed the assumptions that we made in our theoretical analysis in the updated version, in particular the d-regular graph assumption.** The updated theorems further reveals that in heterophilous graphs, the effectiveness of heterophilous attacks is dependent on low-degree nodes, which is a novel insight to the field.
>
> Last but not least, **despite the theorems being derived under specific assumptions, our experiments in Sec. 5.1 have confirmed their generalizability to multiple real-world graphs that go beyond these assumptions**, where the behavior of the attacks can be well explained by our theorems. We also validated the effectiveness of the heterophily-inspired design (Sec. 5.2 and 5.3) under generalized settings on real-world datasets, including on both *targeted* (by Nettack) and *untargeted* attacks (by Metattack), GNNs with *non-linear* activation functions, and both *empirical* and *certifiable* robustness.

---

> > ### Author Response · Authors · 2021-11-23
> > **Reply to Reviewer kkG8 [2/2]**
> >
> > ## "The connection between the heterophilous design principle and the aforementioned theory results is vague."
> >
> > Theorem 1 and Theorem 2 discuss the effective types of attacks (homophilous or heterophilous) on homophilous and heterophilous graphs, which we directly validate in Sec. 5.1. Following the reviewers' feedback, we have further revised the theoretical results and experiments to make the connections even more apparent.
> >
> > Theorem 3 shows theoretically how a *simple* instantiation of the heterophily-adjusted design can lead to better robustness compared to the vanilla GCN without this design, while in practice, the instantiation of the design can be more comprehensive (like the models which we use in the benchmark study, which we elaborate in Appendix D.1). Our empirical analysis in Sec. 5.2 and 5.3 has also clearly show that models featuring the heterophily-adjusted design has better *empirical* and *certifiable* robustness to vanilla models without the design. In the response to Reviewer aMbV, we also discuss the experiments results on APPNP, which formulates its GNN layer similar to Theorem 3 and further supports our analysis.
> >
> > ### Additional References
> >
> > Wu, Felix, et al. "Simplifying graph convolutional networks." *International conference on machine learning*. PMLR, 2019.

---

> > > ### Comment · Reviewer_kkG8 · 2021-11-29
> > > **Thanks for the response!**
> > >
> > > Thank you for the updates which partially address my concern about the theoretical part and also the additional experimental results. I will increase my score from 3 to 5. I like the observation about the connection between the adversarial attack on the graph and the change of heterophily. But the technical contribution and significance are borderline to the acceptance bar.

---

> > > > ### Author Response · Authors · 2021-11-30
> > > > **Thanks for your reply! Could you update your score in the original review?**
> > > >
> > > > Thank you for taking the time to read our response. We are glad to hear that we addressed the majority of your concerns and that you are willing to increase your score — however, we still see your previous score above in your original review; we would appreciate it if you could take some time to update the original score above as well, and thank you in advance!

---

### Official Review · Reviewer_Vyu4 · 2021-11-01

**Correctness:** 4
**Technical Novelty And Significance:** 2
**Empirical Novelty And Significance:** 4
**Recommendation:** 8
**Confidence:** 4

**Main Review:**

### Strengths
* The paper has a thorough and well-designed experimental study.
* The paper studies empirical *and* certifiable robustness, which are both interesting and important.
* The authors compare a broad range of different models.

### Weaknesses
* The paper does not propose a new method.
* The authors only combine the SVD preprocessing with two of the models studied in this work.
* The paper references [Geisler et al. 2020] but does not compare to their proposed model.

### Detailed comments
Some remarks and questions:

**SVD preprocessing**
* Why are only two GNNs combined with SVD? CPGNN is the most robust non-vaccinated GNN, so adding SVD to it might lead to more robustness than e.g., GraphSage-SVD.
* The SVD preprocessing effectively acts as a low-pass filter and could "destroy" some of the high-frequency heterophily information in FB100 and Snap. Could this be a reason why the SVD-augmentation adds much less robustness on these datasets?
* On a related note, the SVD preprocessing does in general not preserve sparsity of the graph. Thus, the 1- and 2-hop neighborhoods of the nodes might become very large and could lead to oversmoothing. Is this an issue in the present study? An alternative could be to use GDC [Klicpera et al. 2019] preprocessing, which is also a low-pass filter but preserves sparsity.

**Robustness in homophily-based GNNs.** While the authors' arguments for increasing robustness via heterophily-adjusted GNNs makes sense to me, [Geisler et al. 2020] achieve comparable or larger certified performance (e.g., as measured by AC) on Cora and Citeseer, while not adhering to the heterophily design proposed in this work. Instead, the method "doubles down" on the homophily design by using GDC [Klicpera et al. 2019] preprocessing and an aggregation function designed to be not affected by outliers. Are the strategies in the present work and [Geisler et al. 2020] alternative solutions to increase robustness in homophily networks? A discussion on this would be very interesting in my view.

**Benchmark results.** In my view the benchmark results section would benefit from also showing the clean performance of the models for the untargeted attacks. For example by having an analogous to Fig. 1 for untargeted attacks or by swapping the error bars in Table 3 with the change in accuracy compared to the clean performance (the error bars should then of course be provided in the appendix).

**Heterophilic structure attacks (Q1) (side remark).** The results for the homophilic datasets in Table 2 seem to corroborate the findings presented in [[Zügner et al. 2020](https://dl.acm.org/doi/10.1145/3394520)]  and [[Zügner and Günnemann 2019](https://openreview.net/pdf?id=Bylnx209YX)] (which is good to see), and the results on heterophilic datasets are a valuable and interesting addition.

**Summary Of The Paper:**

The authors present a study on the robustness on GNNs on homophilic and heterophilic datasets. The authors first show in a stylized theoretical setting that GNNs with a *heterophilic design* (separate aggregation of ego- and neighborhood representations) should show increased robustness against structure perturbations. Then, they follow up with an experimental study where they compare homophily-based GNNs with GNNs following a heterophilic design on both homophilic and heterophilic datasets. In addition, the authors augment three GNN models with the SVD preprocessing by [Entezari et al. 2020] in order to improve robustness. The authors study both empirical as well as certifiable robustness (via randomized smoothing). In general, GNNs with heterophilic design show higher empirical and certifiable robustness across datasets.

**Summary Of The Review:**

In summary, I appreciate the thorough experimental study in the paper and believe that – even though the paper does not propose a new method – the findings have relevant implications for GNN design choices. There are some additional experiments and discussions I would be interested to see (see main review). Overall, I'm not opposed acceptance currently.

After reading the authors' response to my review I increase my score.

---

> ### Author Response · Authors · 2021-11-23
> **Reply to Reviewer Vyu4 [1/3]**
>
> We really appreciate your thoughtful and constructive feedback. We are very glad that you find our experimental study thorough, well-designed and significant. In the response below, we would like to address your specific questions and comments as follows:
>
> ## Questions on SVD preprocessing
>
> - **"Why are only two GNNs combined with SVD?"**
>
>     While it is possible to combine SVD with some additional GNN models, the two models combined with SVD (H2GCN-SVD & GraphSAGE-SVD) that we studied in our experiments in comparison with GCN-SVD are used as representative examples to support our claim that the identified design can be applied in addition to existing vaccination mechanisms. Though CPGNN shows stronger robustness than H2GCN and GraphSAGE, its approach is not compatible with the SVD preprocessing. CPGNN models in its compatibility matrix the likelihood of nodes belonging to each pair of classes to connect in the original graph, while applying SVD will change the graph structure and "destroy" the compatibility information used by CPGNN.
>
> - **"Why the SVD-augmentation adds much less robustness on heterophilous datasets?"**
>
>     We think this question is referring to the robustness improvement by SVD on heterophilous graphs under the untargeted attack setup (which we show in Table 8), as we do observe significant robustness improvement brought by SVD on heterophilous graphs under the targeted attack setup (which we show in Table 7). As discussed in a previous work (Jin et al., 2020b), low-rank based approach does not adapt well against untargeted attacks, which we think explains the generally lower robustness improvement brought by SVD on Metattack compared to improvement on Netattack in our experiments. Comparing the robustness against Metattack, we agree that the low-pass filtering property by SVD can explain in particular the lower robustness improvement by SVD on heterophilous graphs to the improvement on homophilous graphs.
>
> - **"Does SVD preprocessing lead to over-smoothing issue?"**
>
>     We indeed observe that the adjacency matrices preprocessed by SVD are much denser than the original; the loss of sparsity can lead to over-smoothing issues, which helps explain the performance loss of models with SVD on clean datasets compared to the vanilla versions, which we show in Table 3 in the updated paper (and Table 7 and Table 8 in Appendix E.1). We appreciate the suggestion of using GDC preprocessing as an alternative to SVD: our preliminary experiments on GNNs with GDC preprocessing has indeed showed improved accuracy on clean datasets compared to models with SVD-preprocessing, though SVD-preprocessing still shows better robustness against targeted attacks. We discuss the results of the experiments and more details in the next section.

---

> > ### Author Response · Authors · 2021-11-23
> > **Reply to Reviewer Vyu4 [2/3]**
> >
> > ## Comparison to GNNs with Soft Medoid Aggregator (Geisler et al., 2020) and GDC preprocessing (Klicpera et al., 2019)
> >
> > - **"Geisler et al. (2020) achieve comparable or larger certified performance while not adhering to the heterophily design proposed in this work... Are the strategies in the present work and (Geisler et al., 2020) alternative solutions to increase robustness in homophily networks?"**
> >
> >     This is an interesting point, but this observation does not contradict our claim, as Soft Medoid aggregator and GDC takes complementary approaches to our identified design for improving the robustness. Intuitively, our identified design tries to reduce the effects of potentially disruptive connections, while Soft Medoid and GDC tries to aggregate more useful information by taking into account additional helpful (homophilous) edges (that are filtered by the more robust Soft Medoid aggregator, which also limits the influence of "outliers").
> >
> >     We further validate the above intuition by conducting additional empirical analysis to demonstrate the effectiveness of our identified design when used together with the Soft Medoid aggregator and GDC. In these preliminary experiments, we compare the performance on clean datasets and against *targeted* *poison* attacks by Nettack of (1) three base GNN models (GCN, GraphSAGE, H2GCN), (2) their variants with GDC preprocessing, and (3) their variants with both GDC and Soft Medoid aggregator. We follow the experiment setup in Sec. 5 of our paper, and the experiment results are presented in the following tables, which supplements the results in Table 7 in Appendix E.1.
> >     | Dataset Name             | Cora           |         |               |         | Citeseer       |         |               |         |
> >     | ------------------------ | -------------- | ------- | ------------- | ------- | -------------- | ------- | ------------- | ------- |
> >     | **Model Name**           | **acc_attack** | **std** | **acc_clean** | **std** | **acc_attack** | **std** | **acc_clean** | **std** |
> >     | GCN                      | 1.67%          | 0.00%   | 82.78%        | 4.78%   | 4.44%          | 2.83%   | 69.44%        | 7.74%   |
> >     | GCN + GDC                | 27.22%         | 4.37%   | 82.22%        | 5.67%   | 10.56%         | 2.83%   | 71.11%        | 6.14%   |
> >     | GCN + Medoid + GDC       | 40.00%         | 4.91%   | 77.78%        | 3.93%   | 33.89%         | 2.83%   | 62.22%        | 0.79%   |
> >     | GraphSAGE                | 36.67%         | 2.72%   | 82.22%        | 9.56%   | 31.67%         | 10.89%  | 70.56%        | 6.85%   |
> >     | GraphSAGE + GDC          | 44.44%         | 2.08%   | 82.22%        | 6.71%   | 42.22%         | 6.85%   | 73.33%        | 6.24%   |
> >     | GraphSAGE + Medoid + GDC | 56.67%         | 8.28%   | 78.33%        | 5.44%   | 46.67%         | 3.60%   | 67.78%        | 2.83%   |
> >     | H2GCN                    | 38.89%         | 5.50%   | 82.78%        | 8.31%   | 27.22%         | 1.57%   | 69.44%        | 6.98%   |
> >     | H2GCN + GDC              | 50.00%         | 4.71%   | 81.67%        | 5.93%   | 33.33%         | 1.36%   | 70.56%        | 2.83%   |
> >     | H2GCN + Medoid + GDC     | 59.44%         | 4.37%   | 77.22%        | 4.78%   | 43.33%         | 3.60%   | 67.22%        | 1.57%   |
> >
> >
> >
> >     | Dataset Name         | FB100          |         |               |         | Snap           |         |               |         |
> >     | -------------------- | -------------- | ------- | ------------- | ------- | -------------- | ------- | ------------- | ------- |
> >     | **Model Name**       | **acc_attack** | **std** | **acc_clean** | **std** | **acc_attack** | **std** | **acc_clean** | **std** |
> >     | GCN                  | 0.56%          | 0.79%   | 63.33%        | 2.72%   | 2.22%          | 2.08%   | 33.33%        | 2.72%   |
> >     | GCN + GDC            | 1.11%          | 0.79%   | 58.33%        | 1.36%   | 20.00%         | 9.53%   | 29.44%        | 3.14%   |
> >     | GCN + Medoid + GDC       | 16.67%         | 4.08%   | 51.11%        | 5.67%   | 20.56%         | 5.15%   | 28.33%        | 2.36%   |
> >     | GraphSAGE            | 33.89%         | 3.42%   | 60.00%        | 2.72%   | 16.67%         | 7.07%   | 24.44%        | 4.16%   |
> >     | GraphSAGE + GDC      | 13.89%         | 3.93%   | 57.78%        | 2.08%   | 13.89%         | 6.71%   | 27.78%        | 4.16%   |
> >     | GraphSAGE + Medoid + GDC | 47.22%         | 4.16%   | 59.44%        | 1.57%   | 20.56%         | 3.14%   | 29.44%        | 4.16%   |
> >     | H2GCN                | 27.78%         | 3.42%   | 60.56%        | 1.57%   | 12.78%         | 2.83%   | 30.00%        | 2.72%   |
> >     | H2GCN + GDC          | 18.33%         | 4.08%   | 61.67%        | 2.36%   | 11.67%         | 2.36%   | 28.33%        | 2.72%   |
> >     | H2GCN + Medoid + GDC     | 47.22%         | 1.57%   | 61.67%        | 0.00%   | 22.22%         | 1.57%   | 30.56%        | 0.79%   |

---

> > > ### Author Response · Authors · 2021-11-23
> > > **Reply to Reviewer Vyu4 [3/3]**
> > >
> > > From the above results, we observe that
> > >
> > > - Among all models with both GDC and Soft Medoid aggregator, models based on heterophily-adjusted GNNs have better performance against attacks in all datasets compared to GCN + Medoid + GDC, with up to 19.44% improvement on homophilous graphs and 30.55% improvement on heterophilous graphs. These results further confirm that **our strategy can be readily combined with existing defense mechanisms like Soft Medoid + GDC (or SVD as we showed in the paper) to further boost the robustness of GNNs**.
> > > - GraphSAGE + Medoid + GDC and H2GCN + Medoid + GDC in most cases also show better performance on clean datasets compared to GCN + Medoid + GDC, especially on heterophilous datasets.
> > > - As a side note, when using GCN as the base model, the homophilous connections introduced by GDC can reduce the clean performance of the model on heterophilous graphs. This shows another benefits of incorporating our heterophily-inspired design in the GNN models.
> > >
> > > While we only evaluate these models against targeted attacks (Nettack) in this preliminary experiment, we will complete the experiments on the remaining setups and incorporate the full results in the camera ready version.
> > >
> > > ## Showing clean performance of the models for the untargeted attacks in the main paper
> > >
> > > Thank you for your suggestion; we have updated Table 3 by swapping the std values with the clean performance values. The std values are still provided in the Table 7 and Table 8 in Appendix E.1.
> > >
> > > ## No new method is proposed
> > >
> > > As we mention in the meta-response, our focus is on *understanding and formalizing the relation* between heterophily and adversarial structural attacks, and demonstrating *how the implication of this relation helps* improve the robustness of GNNs. While we do not propose a new defense approach, and some of the designs we identified are used in existing methods, our analysis on the relation between heterophily and attacks is *novel* (which we strengthen in the updated version)*.* Further, its implication on how the heterophily-inspired designs can help boost the robustness of GNNs is also *novel*. We believe these insights are beneficial to the field, and we are pleased for your appreciation of these contributions.

---

> > > > ### Comment · Reviewer_Vyu4 · 2021-11-29
> > > > **Thank you**
> > > >
> > > > Thank you for the response, which addressed most of my concerns. I have adapted my score accordingly. I would like the authors to include the results with GDC preprocessing into the final version of the paper as I feel that this is important information for researchers making use of the findings in this paper.

---

> > > > > ### Author Response · Authors · 2021-11-30
> > > > > **Thanks for your reply!**
> > > > >
> > > > > Thank you for taking the time to read our response! We are glad to hear that we addressed most of your concerns and that you are willing to increase your score. We agree with your thoughts on the importance of the GDC results presented in our responses, and we will make sure to include them into the final version.

---

### Official Review · Reviewer_aMbV · 2021-11-02

**Correctness:** 3
**Technical Novelty And Significance:** 2
**Empirical Novelty And Significance:** 3
**Recommendation:** 6
**Confidence:** 4

**Main Review:**

This work studies the relation between graph heterophily and the robustness of GNNs and theoretically show that effective structural attacks on GNNs for homophilious graphs lead to increased heterophily level, while for heterophilious graphs they alter the homophily level contingent on node degrees under some specific assumptions.

Despite this, the paper is not particularly novel. The theoretical results are derived under some specific assumptions on graphs and GNN layers, which may not be appliable to real-world graphs.
Theorem 3 cannot support the claims that the heterophily-inspired design (i.e. separate aggregators for ego- and neighbor-embeddings) could improve the robustness of GNNs because the GNN layer analyzed in the theorem is significantly different with the heterophily-inspired design.
Furthermore, experiments are conducted on small datasets and the experimental results cannot validate whether the theoretical claims can be generalized to large real-world graphs, which is an important aspect given that the theoretical claims are based on specific assumptions about graphs.



## Strength

1. The paper is well written and is easy to follow.
2. The authors have studied an important open problem in deep graph learning community.


## Weakness

1.  The paper is limited in novelty: while the paper investigates the relation between graph heterophily and the robustness of GNN and show that several heterophily-inspired GNNs work more robust to GCN and GAT, there is no new methodology proposed in the paper.
2.  This paper is constrained to specific target attacks (i.e. CM-type attacks) and linear GNN layers, which may not be applicable to most GNN architectures with nonlinearity and other types of attacks.
In addition, they are derived for specific graphs with fixed degrees for every node and specific node features, which may not generalize to real-world graphs.
3.   Theorem 3 does not support their claims on the effectiveness of the heterophily-inspired design (i.e. separate aggregator for ego- and neighbor embeddings). Theorem 3 shows that the GNN layer $f = ((1 - \alpha)\bar{\mathbf A} + \alpha\mathbf {I})\mathbf{X}\mathbf {W}$ is more robust than the GNN layer $f_s = \bar{\mathbf  A_s}\mathbf {X}\mathbf {W}$ to CM-type attack on a node. The GNN layer f follows the message passing scheme of APPNP (with K=1) and GPRGNN (with K=1, $\gamma_0=\alpha$, $\gamma_1=1-\alpha$), which indicates that Theorem 3 is applicable to APPNP and GPRGNN under some assumptions. However, it is not clear whether it could work for the heterophily-inspired design since the architectures of ANNPN and GPRGNN are significantly different toi the heterophily-inspired design.
4.  It would be nice to include APPNP in the experiments since the architecture of APPNP is very close to the GNN layer analyzed in Theorem 3.
5.  The datasets used in the experiment are small, say, around 2k nodes, and it is not clear whether the results can be extended to large real-world graphs. It would be great to include experiments on larger datasets to validate their theoretical claims,  especially when their claims were made upon specific assumptions. Moreover, the homophily level h of FB100 is 0.531 (> 0.5), which does not meet the definition of heterophilious graphs ($h <<1/ |Y|$) in the paper. Other heterophilious benchmark datasets with lower homophily levels may be more helpful for validating the theoretical claims for heterophilious graphs.
6.   There is no empirical study for validating the claim of Theorem 2 that the type (homophily or heterophily) of attack on a node for heterophilious graph depends on the node degree.

## Minor comments:
1. Table 7 in Appendix E.1 shows that GAT and GCN outperform most heterophily-inspired GNNs (e.g. H2GCN, GPRGNN, FAGCN) on the original (clean) FB100 and Snap datasets. Can you give some insights on why GAT and GCN worked better than heterophily-inspired GNNs on these heterophilious graphs?


**Summary Of The Paper:**

This paper studies the relationship between graph heterophily and robustness of GNNs. The authors claim that effective structural attacks on GNNs for homophilious graphs lead to increased heterophily level, while for heterophilious graphs they alter the homophily level contingent on node degrees. The paper shows that a heterophily-inspired design for GNN, separate aggregators for ego- and neighbor-embeddings, could improve the robustness of several GNN architectures.

**Summary Of The Review:**

This paper studies an interesting open problem in deep graph learning. However, it has limited novelty, and theoretical results do not support some of their claims.

---

> ### Author Response · Authors · 2021-11-23
> **Reply to Reviewer aMbV [1/3]**
>
> We really appreciate your thoughtful and constructive feedback, and we are glad that you find the problem on the relationship between heterophily and robustness interesting and important. In this response, we would like to address your specific questions and comments as follows:
>
> ## "The paper is limited in novelty."
>
> As we mention in the meta-response, rather than proposing a well-rounded new method, the main focus of our work is on ***understanding and formalizing the relation* between heterophily and adversarial structural attacks, and *demonstrating the implication* of this insight on improving the robustness of GNNs against adversarial perturbations.**
>
> - Firstly, we *formally* show that adversarial structural attacks on homophilous graphs mostly lead to increase of the heterophily level. **Though former works have utilized this relation as heuristic in designing GNNs with improved robustness, they do not formally propose this research question, nor provide any formal theoretical analysis.**
>     - Though derived under specific assumptions, our theoretical analysis (Theorem 1) helps reveal *why* effective attacks on commonly seen homophilous graphs need to exploit increased level of heterophily.
>     - Our empirical observations (Table 2 in Section 5.1) confirm our takeaway on more generalized settings for real-world datasets.
> - Moreover, to our knowledge, **we are the first to show on *heterophilous* graphs how adversarial attacks change the level of homophily with formal theoretical and empirical analysis**. With our updated analysis, we reveal that in heterophilous graphs, the effectiveness of heterophilous attacks is dependent on low-degree nodes, which is supported by our empirical analysis on real-world datasets.
> - Finally, we demonstrate how the implication of the relation between heterophily and adversarial attacks can help defend against the attacks by showing the effectiveness of the heterophily-inspired designs in improving the robustness of GNNs. While some of these designs are used in existing methods, **their effectiveness in improving the robustness of GNNs with formal theoretical justifications, and the extensive evaluation on both *empirical* and *certifiable* *robustness* is novel, which has been largely unknown before this work*.*** Though we do not propose a concrete new defense method, we believe our observations can inspire future works in leveraging the relations between heterophily and adversarial attacks.
>
> ## "This paper is constrained to specific target attacks (i.e. CM-type attacks) and linear GNN layers."
>
> In our empirical evaluation, we have validated the takeaways of our theoretical analysis (in Sec. 5.1, which we have strengthened in the updated version) and the effectiveness of the heterophily-inspired design (Sec. 5.2 and 5.3) under generalized settings on real-world datasets, including on both *targeted* (by Nettack) and *untargeted* attacks (by Metattack, which is based on *cross-entropy loss* instead of the CM-type attack loss we discussed), GNNs with *non-linear* activation functions, and both *empirical* and *certifiable* robustness, despite our theoretical analysis is derived under specific assumptions (which we also relaxed in this update).
>
> ## Questions regarding our theoretical results
>
> - "**Theorems are derived under specific assumptions which may not generalize to real-world graphs."**
>
>     We appreciate your feedback; **we have relaxed the assumptions that we made in our theoretical analysis**, the d-regular graph assumption in particular, in the updated version.
>
>     - Our theorems are now able to address the cases where degree $d_a$ of the gambit nodes (i.e., node leveraged by the attackers to affect the prediction of the target node, which we defined in the updated version) are different than the degree $d$ of the target nodes, bringing the theorems much closer to the real-world attack scenarios, where attackers can leverage gambit nodes with arbitrary degrees in typical graphs with power-law degree distributions.
>     - Furthermore, the updated Theorem 2 better reveals on hetereophilous graphs how both the degrees of the target and gambit nodes determine the effective type of attacks: if the degree of *either node* is low, attacks increasing the heterophily are effective; however, if the degrees *both nodes* are high, attacks *decreasing* the heterophily will then be effective.
>     - Though our updated theorems are still derived in a stylized learning setup under specific (relaxed) assumptions, they can explain the behavior of the attacks on both homophilous and heterophilous real-world datasets that go beyond our assumptions (which we discuss more in the next bullet).

---

> > ### Author Response · Authors · 2021-11-23
> > **Reply to Reviewer aMbV [2/3]**
> >
> > - **"No empirical study for validating the claim of Theorem 2"**
> >
> >     As we outline in the meta-response, we have updated our experiments in Section 5.1 to better validate our claims in Theorem 2. Specifically,
> >
> >     - We update the budget of the attacks per node to be 1 (as our theorems focus on unit perturbations), and focus on target nodes that have been correctly classified on clean datasets (since no attacks are needed to misclassify a node that is already incorrectly classified).
> >     - We **investigate the distribution of the degrees of the gambit node leveraged by the attacks**, and discover that the attacks mostly utilize gambit nodes with low-degrees, especially on heterophilous graphs.
> >     - We observe that **all targeted attacks conducted by Nettack on real-world heterophilous datasets follow the takeaways of the updated Theorem 2**. These results support the conclusions of the theorem, and generalize them to real-world attack settings which go beyond our assumptions in theoretical analysis.
> >
> >     While we only conduct the updated experiments on targeted attacks due to limited time of the discussion period, we will complete the experiments on untargeted attacks and incorporate these results in the camera-ready version.
> >
> > - **"Theorem 3 does not support their claims on the effectiveness of the heterophily-inspired design... the GNN layer analyzed in the theorem is significantly different with the heterophily-inspired design."**
> >
> >     The GNN layer we analyzed in Theorem 3 actually *follows* the design we defined: by letting the aggregator of ego-embedding as $\texttt{AGGR1}(\cdot) = \alpha \mathbf{X}$, the aggregator of neighbor-embedding as $\texttt{AGGR2}(\cdot) = (1-\alpha)\mathbf{\bar{A}X}$ and the encoder as $\texttt{ENC}(\mathbf{x}_1, \mathbf{x}_2) = (\mathbf{x}_1+ \mathbf{x}_2)\mathbf{W}$
> >     , the formulation of our design defined in Eq. (1) can be recovered. Theorem 3 shows that it is possible to adjust the weight $\alpha$ (manually or through training) in this *simple* instantiation of the heterophily-adjusted design to achieve better robustness compared to the vanilla GCN without this design, while in practice the instantiation of the design can be more comprehensive (like the models which we use in the benchmark study, which we elaborate in Appendix D.1).
> >
> > - "**Architectures of APPNP and GPR-GNN are significantly different to the heterophily-inspired design."**
> >
> >     Both APPNP and GPRGNN actually *follows* our heterophily-inspired design, as we discussed in the previous bullet. In Appendix D.1, we also elaborated on how the heterophilous design is instantiated in GPR-GNN.
> >
> > ## "Include APPNP in the experiment"
> >
> > Thank you for your suggestion; since APPNP also features our heterophily-inspired design and is close to the GNN layer we analyzed in Theorem 3, we agree it would be a nice addition to include APPNP in our experiments.
> >
> > We have run preliminary experiments to evaluate the empirical robustness of APPNP (with $\alpha=0.9$) against *targeted poison* and *evasion* attacks by Nettack. We follow the experiment setup in Sec. 5 of our paper, and the experiment results are presented in the following tables, which supplements the results in Table 7 in Appendix E.1.
> >
> > |                  | Poison       |         | Evasion      |         | Clean        |         |
> > | ---------------- | ------------ | ------- | ------------ | ------- | ------------ | ------- |
> > | **Dataset Name** | **accuracy** | **std** | **accuracy** | **std** | **accuracy** | **std** |
> > | Cora             | 58.33%       | 3.60%   | 67.78%       | 7.86%   | 72.22%       | 5.50%   |
> > | Citeseer         | 56.11%       | 3.14%   | 65.56%       | 4.16%   | 68.33%       | 4.71%   |
> > | FB100            | 36.67%       | 2.36%   | 49.44%       | 4.78%   | 58.89%       | 3.93%   |
> > | Snap             | 25.00%       | 1.36%   | 25.56%       | 3.42%   | 28.33%       | 2.36%   |
> >
> > We observe that while APPNP has a larger trade off on clean performance on Cora, it is also significantly more robust as a heterophily-adjusted model to the non-heterophily-adjusted GNNs, thus supporting the effectiveness of our design. While we only evaluate APPNP against targeted attacks (Nettack) in this preliminary experiment, we will complete the experiments on the remaining setups and incorporate the full results in the camera ready version.

---

> > > ### Author Response · Authors · 2021-11-23
> > > **Reply to Reviewer aMbV [3/3]**
> > >
> > > ## "The datasets used in the experiment are small, ... not clear whether the results can be extended to large real-world graphs."
> > >
> > > The homophilous datasets we used in the experiment are similar to the sizes of most datasets used in existing works (Geisler et al., 2020; Jin et al., 2020a), with one exception of Pubmed, which has around 20k nodes. While we do not expect that the validity of our takeaways would be affected by the size of the graphs, we have conducted additional experiments to evaluate the empirical robustness of the GNN models against targeted attacks by Nettack on Pubmed. We report the results in the following table, which supplements the results in Table 7 in Appendix E.1; our observations remain the same on these new results. We note that GCN-SVD and ProGNN run out of memory on Pubmed, thus their results are not included here.
> > >
> > > |                | Poison       |         | Evasion      |         | Clean        |         |
> > > | -------------- | ------------ | ------- | ------------ | ------- | ------------ | ------- |
> > > | **Model Name** | **accuracy** | **std** | **accuracy** | **std** | **accuracy** | **std** |
> > > | H2GCN-SVD      | 86.11%       | 3.93%   | 86.11%       | 3.93%   | 87.22%       | 4.37%   |
> > > | GraphSAGE-SVD  | 81.11%       | 4.16%   | 81.11%       | 3.42%   | 84.44%       | 2.08%   |
> > > | H2GCN          | 44.44%       | 5.67%   | 46.67%       | 8.16%   | 87.78%       | 3.14%   |
> > > | GraphSAGE      | 33.33%       | 8.92%   | 34.44%       | 9.06%   | 84.44%       | 3.93%   |
> > > | CPGNN          | 60.00%       | 7.20%   | 60.00%       | 5.93%   | 82.78%       | 5.67%   |
> > > | GPR-GNN        | 13.89%       | 4.78%   | 15.56%       | 6.14%   | 85.56%       | 1.57%   |
> > > | FAGCN          | 27.78%       | 11.00%  | 31.67%       | 13.40%  | 86.67%       | 2.72%   |
> > > | GNNGuard       | 73.89%       | 6.71%   | -            | -       | 82.78%       | 2.83%   |
> > > | GCN            | 2.22%        | 1.57%   | 2.78%        | 2.08%   | 85.56%       | 1.57%   |
> > > | GAT            | 7.22%        | 4.16%   | 6.67%        | 4.08%   | 83.33%       | 1.36%   |
> > > | MLP*           | -            | -       | -            | -       | 86.11%       | 4.37%   |
> > >
> > > ## "FB100 does not meet the definition of heterophilous graphs"
> > >
> > > We noted in the "Datasets & Evaluation Setup" paragraph at the beginning of Sec. 5 that FB100 is a *weakly* heterophilous graph based on our definition, as it has a homophily ratio $h \approx {1}/{|\mathcal{Y}|}$. We agree that a heterophilous datasets with a lower homophily ratio $h$ would be more ideal, but we did not discover other datasets with a lower homophily ratio $h$ while having one-hot encoded feature vectors similar to Cora (in order to run Nettack on the data). We will look more into this in the future.

---

> > > > ### Comment · Reviewer_aMbV · 2021-11-29
> > > > **Thanks for the response!**
> > > >
> > > > I have read the author response! I do appreciate the added experiments and theory by the authors.  Meanwhile, I am still a bit concerned about the scope of applicability of the theory and the novelty of this work.  I am going to increase my score a bit,
> > > > but I would not strongly champion for its acceptance.

---

> > > > > ### Author Response · Authors · 2021-11-30
> > > > > **Thanks for your reply!**
> > > > >
> > > > > Thank you for taking the time to read our response! We are glad to hear that the added experiments and theory address many of your concerns and that you are willing to increase your score.

---

> > > ### Comment · Reviewer_Vyu4 · 2021-11-24
> > > **Question regarding heterophily-inspired design**
> > >
> > > Hi, I am another reviewer reading your response and I have a follow-up question regarding your response to "Theorem 3 does not support their claims on the effectiveness of the heterophily-inspired design... the GNN layer analyzed in the theorem is significantly different with the heterophily-inspired design.".
> > >
> > > You suggest to map the GNN layer of Theorem 3 to the heterophily-inspired design. Could we not also do a similar thing and map GCN (which does **not** follow the heterophily-inspired design) to Theorem 3? I.e.,
> > >
> > > * $\mathrm{AGGR1}(\cdot) = \frac{1}{\tilde{d}_i}\mathbf{X}$ for the ego embedding;
> > > * $\mathrm{AGGR2}(\cdot) = (\hat{\mathbf{A}} - \mathrm{diag}(\hat{\mathbf{A}})) \mathbf{X}$ for the neighbors, hence removing the diagional (self-loops) from the message-passing matrix $\hat{\mathbf{A}}$;
> > > * $\mathrm{ENC}(\mathbf{x}_1, \mathbf{x}_2) = \mathrm{ReLU}((\mathbf{x}_1 + \mathbf{x}_2)\mathbf{W})$?
> > >
> > > It seems to me that this would exactly and wrongly recover GCN as a "heterophily-inspired" GCN. Or am I missing something?

---

> > > > ### Author Response · Authors · 2021-11-24
> > > > **The Essence of the Design**
> > > >
> > > > Thank you for bringing up this great question! While it is possible to write separate formulations for the aggregation weights for ego- and neighbor-embeddings for all GNNs (as you demonstrate in the case of GCN), the emphasis of the design is on having ***separate mechanisms with adjustable weights that can be learned or tuned independently**.* As we note in our response, the focus of the Theorem 3 is on **adjusting the weight $α$**, which is an additional degree of freedom for the ego-embedding, *manually or through training* to achieve better robustness compared to the vanilla GCN. In more complex instantiation of the design (like the models which we discussed in Appendix D.1), the weight $\alpha$ can be replaced by parameters with more degrees of freedom, like a separate weight matrix for linear transformation. In the case of GCN, though we can write the formulation of GCN into two equations as $\mathrm{AGGR1}(\cdot) = \frac{1}{\tilde{d}_i}\mathbf{X}$ and $\mathrm{AGGR2}(\cdot) = (\hat{\mathbf{A}} - \mathrm{diag}(\hat{\mathbf{A}})) \mathbf{X}$, these two equations are from the same aggregation mechanism that calculates together the average of ego- and neighbor-embeddings weighted by the degrees; *it does not have any adjustable weights like $\alpha$ which can be learned or tuned differently for ego- and neighbor-embeddings to make use of the information from the ego-embedding independently.*
> > > >
> > > > With that being said, we think our previous response, which focuses on breaking down the GNN models to match the formulation of our design in Eq. (1), may have created some ambiguity on **the key idea of the design *(separate mechanisms with independently adjustable weights)***, and we hope this response would help clarify that. We will also revise our paper to make the essence of our design more clear in the camera-ready version.

---

### Official Review · Reviewer_jMaA · 2021-11-03

**Correctness:** 2
**Technical Novelty And Significance:** 3
**Empirical Novelty And Significance:** 2
**Recommendation:** 5
**Confidence:** 3

**Main Review:**

In this paper, the authors analyze the relation between GNN robustness against attacks with heterophily and propose a method to improve GNN robustness by separating the aggregators of ego- and neighbor-embeddings. In general, the authors focus on an interesting and novel problem and the paper is of good structure. However, there are some issues the authors need to address.

My primary concern is with the evaluation. First, no baseline methods on improving GNN robustness or defending attacks are provided. The authors only show the performance gap between before- and after-attack on different GNNs that integrate the proposed technique. However, it is unclear how much the results are significant without comparing the baseline methods. I suggest the authors either add several baselines to compare or justify why no baseline is applicable. Meanwhile, I suggest the authors to compare the GNN models that integrate the proposed technique with the vanilla versions to examine the results without attacking so that the readers can understand how the general performance can be affected by separating the ego-aggregator.

The novelty of the proposed method also needs to be justified. Using stacked ego-embeddings for prediction has been applied before to solve issues like oversmoothing. I suggest the authors to further justify why the proposed approach is novel and significant.

**Summary Of The Paper:**

In this paper, the authors analyze the relation between GNN robustness against attacks with heterophily and propose a method to improve GNN robustness by separating the aggregators of ego- and neighbor-embeddings.

**Summary Of The Review:**

Strength:
+ Focusing on an interesting problem
+ The paper is of good structure

Weakness:
- Incompelete evaluation
- The novelty needs to be further justified

---

> ### Author Response · Authors · 2021-11-23
> **Reply to Reviewer jMaA**
>
> We appreciate your review on our work, and we would like to address your specific comments as follows:
>
> ## "Incomplete evaluation"
>
> - **"No baseline methods on improving GNN robustness or defending attacks are provided."**
>
>     We have conducted a thorough and well-designed experimental study (as noted by Reviewer Vyu4) which includes as baselines 3 state-of-the-art "vaccinated" GNN models (ProGNN, GNNGuard and GCN-SVD) proposed by previous works that incorporate designs and architectures with improved robustness in mind. We have made this lineup clear in the "GNN Models" paragraph at the beginning of Section 5, and we have extensively compared the empirical robustness of these "vaccinated" baselines to the heterophily-adjusted models with our design in Section 5.2.
>
> - **"Compare heterophily-adjusted GNNs with the vanilla versions to examine the results without attacking."**
>
>     We have discussed the comparison on clean performance between heterophily-adjusted GNNs and models without it (GCN and GAT) in Section 5.2 in the previous version: heterophily-adjusted GNNs have similar performance on clean datasets to models without this design. The detailed results on clean performance of GNNs are previously presented in the appendix; in the updated version, we have incorporated them in Table 3 of the main paper to make them more prominent following the suggestion of Reviewer Vyu4.
>
>
> ## Justifications of novelty and significance
>
> As we mention in the meta-response, rather than proposing a well-rounded new method, the main focus of our work is on ***understanding and formalizing the relation* between heterophily and adversarial structural attacks, and *demonstrating the implication* of this insight on improving the robustness of GNNs against adversarial perturbations.**
>
> - Firstly, we *formally* show that adversarial structural attacks on homophilous graphs mostly lead to increase of the heterophily level. **Though former works have utilized this relation as heuristic in designing GNNs with improved robustness, they do not formally propose this research question, nor provide any formal theoretical analysis.**
>     - Though derived under specific assumptions, our theoretical analysis (Theorem 1) helps reveal *why* effective attacks on commonly seen homophilous graphs need to exploit increased level of heterophily.
>     - Our empirical observations (Table 2 in Section 5.1) confirm our takeaway on more generalized settings for real-world datasets.
> - Moreover, to our knowledge, **we are the first to show on *heterophilous* graphs how adversarial attacks change the level of homophily with formal theoretical and empirical analysis**. With our updated analysis, we reveal that in heterophilous graphs, the effectiveness of heterophilous attacks is dependent on leveraging low-degree nodes, which is supported by our empirical analysis on real-world datasets.
> - Finally, we demonstrate how the implication of the relation between heterophily and adversarial attacks can help defend against the attacks by showing the effectiveness of the heterophily-inspired designs in improving both empirical and certifiable robustness of GNNs. While some of these designs are used in existing methods, **their effectiveness in improving the robustness of GNNs with formal theoretical justifications, and the extensive evaluation on both *empirical* and *certifiable* *robustness* is novel, which has been largely unknown before this work*.*** Though we do not propose a concrete new defense method, we believe our observations can inspire future works in leveraging the relations between heterophily and adversarial attacks.

---

> > ### Author Response · Authors · 2021-11-30
> > **Gentle Reminder for Checking out the Rebuttal Responses**
> >
> > Since the discussion period is concluding soon, can you please take a look at our rebuttal responses and let us know if we have adequately addressed the points that you raised in your original reviews and if you have any further questions? Thank you in advance for taking the time to check out our updates and responses!

---

### Author Response · Authors · 2021-11-23
**Meta Response: Updates to the Paper, Recap on the Novelty, and Assumptions in Theoretical Analysis**

In this meta-response, we first outline the updates to our paper, and then clarify the technical contributions of our work and discuss the concerns regarding the limitations of the assumptions we made in our theoretical analysis. We also address specific comments of each reviewer through separate replies.

# Updates to the paper

- We have made several **updates to our theoretical analysis in Section 3**, more specifically,
    - **We relaxed the assumptions we made in the theorems**, in particular the d-regular graph assumption, as suggested by several reviewers. The updated theorems are much closer to the real-world attack scenarios, where the graphs usually follow power-law degree distributions, and the attackers can leverage nodes with arbitrary degrees (mostly low-degree nodes as our updated experiments revealed) as gambit to attack the target nodes.
    - Furthermore, **the updated Theorem 2 better reveals on hetereophilous graphs how  degrees of both the target and gambit nodes determine the effective type of attacks**: if the degree of *either node* is low, attacks increasing heterophily are effective; however, if the degrees of *both nodes* are high, attacks *decreasing* heterophily will then be effective.
- We also improved our experiments in Section 5.1 to better validate our claims in Theorem 2 to address the concerns of Reviewer aMbV. Specifically,
    - We update the budget of the attacks per node to be 1 (as our theorems focus on unit perturbations), and focus on target nodes that have been correctly classified on clean datasets (since no attacks are needed to misclassify a node that is already incorrectly classified).
    - We **investigate the distribution of the degrees of the gambit node leveraged by the attacks**, and discover that the attacks mostly utilize gambit nodes with low-degrees, especially on heterophilous graphs.
    - We observe that **all targeted attacks conducted by Nettack on real-world heterophilous datasets follow the takeaways of the updated Theorem 2**. These results support the conclusions of the theorem, and generalize them to real-world attack settings which go beyond our assumptions in theoretical analysis.

# Recap: Technical Contributions & Novelty

Rather than proposing a well-rounded new method, the main focus of our work is on ***understanding and formalizing the relation* between heterophily and adversarial structural attacks, and *demonstrating the implication* of this insight on improving the robustness of GNNs against adversarial perturbations.**

- Firstly, we *formally* show that adversarial structural attacks on homophilous graphs mostly lead to increase of the heterophily level. **Though former works have utilized this relation as heuristic in designing GNNs with improved robustness, they do not formally propose this research question, nor provide any formal theoretical analysis.**
    - Though derived under specific assumptions, our theoretical analysis (Theorem 1) helps reveal *why* effective attacks on commonly seen homophilous graphs need to exploit increased level of heterophily.
    - Our empirical observations (Table 2 in Section 5.1) confirm our takeaway on more generalized settings for real-world datasets.
- Moreover, to our knowledge, **we are the first to show on *heterophilous* graphs how adversarial attacks change the level of homophily with formal theoretical and empirical analysis**. With our updated analysis, we reveal that in heterophilous graphs, the effectiveness of heterophilous attacks is dependent on leveraging low-degree nodes, which is supported by our empirical analysis on real-world datasets.
- Finally, we demonstrate how the implication of the relation between heterophily and adversarial attacks can help defend against the attacks by showing the effectiveness of the heterophily-inspired designs in improving both empirical and certifiable robustness of GNNs. While some of these designs are used in existing methods, **their effectiveness in improving the robustness of GNNs with formal theoretical justifications, and the extensive evaluation on both *empirical* and *certifiable robustness* is novel.** We believe these observations can inspire future works in leveraging the relations between heterophily and adversarial attacks.

# Assumptions Used in Theoretical Analysis

We agree that there are limitations for our theoretical takeaways due to the assumptions made in the theorems; **we have relaxed these assumptions in the updated version**, as we outlined in the above discussion. We also want to note that **some of the assumptions we made in the analysis, such as linearized GNN, have also been utilized by previous works** (Zügner et al., 2018; Zügner & Günnemann, 2019a). Despite the theorems being derived under specific assumptions, **our experiments in Sec. 5.1 have confirmed their generalizability on real-world graphs that go beyond these assumptions**, where the behavior of the attacks can be well explained by our theorems.

---

### Decision · Program_Chairs · 2022-01-20

**Decision:**

Reject

**Comment:**

This work studies the relation between graph heterophily and the robustness of GNNs and theoretically shows that effective structural attacks on GNNs for homophilous graphs lead to increased heterophily level, while for heterophils graphs they alter the homophily level contingent on node degrees under some specific assumptions.

Overall, the findings in the paper are interesting and can be useful for other researchers trying to improve GNNs' robustness on homophilic and heterophilic datasets. After the discussion and rebuttal, the main concerns are:
- while the paper has shown some interesting observations, no new methodology was proposed based on these findings.
- The authors have attempted to relax assumptions and justified their setup on experiments, however the explanations are still limited. For example, Theorem 1 does not allow attention mechanism, different choices of aggregator, skip-connection, and more GNN layers.